

# A high-resolution growth series of *Tyrannosaurus rex* obtained from multiple lines of evidence

Thomas D. Carr

Department of Biology, Carthage College, Kenosha, WI, USA

## ABSTRACT

**Background:** During the growth of complex multicellular organisms, chronological age, size and morphology change together in a hierarchical and coordinated pattern. Among extinct species, the growth of *Tyrannosaurus rex* has received repeated attention through quantitative analyses of relative maturity and chronological age. Its growth series shows an extreme transformation from shallow skulls in juveniles to deep skulls in adults along with a reduction in tooth count, and its growth curve shows that *T. rex* had a high growth rate in contrast to its closest relatives. However, separately, these sets of data provide an incomplete picture of the congruence between age, size, and relative maturity in this exemplar species. The goal of this work is to analyze these data sets together using cladistic analysis to produce a single hypothesis of growth that includes all of the relevant data.
**Methods:** The three axes of growth were analyzed together using cladistic analysis, based on a data set of 1,850 morphological characters and 44 specimens. The analysis was run in TNT v.1.5 under a New Technology search followed by a Traditional search. Correlation tests were run in IBM SPSS Statistics v. 24.0.0.0.
**Results:** An initial analysis that included all of the specimens recovered 50 multiple most parsimonious ontograms a series of analyses identified 13 wildcard specimens. An analysis run without the wildcard specimens recovered a single most parsimonious tree (i.e., ontogram) of 3,053 steps. The ontogram is composed of 21 growth stages, and all but the first and third are supported by unambiguously optimized synontomorphies. *T. rex* ontogeny can be divided into five discrete growth categories that are diagnosed by chronological age, morphology, and, in part, size (uninformative among adults). The topology shows that the transition from shallow to deep skull shape occurred between 13 and 15 years of age, and the size of the immediate relatives of *T. rex* was exceeded between its 15th and 18th years. Although size and maturity are congruent among juveniles and subadults, congruence is not seen among adults; for example, one of the least mature adults (RSM 2523.8) is also the largest and most massive example of the species. The extreme number of changes at the transition between juveniles and subadults shows that the ontogeny of *T. rex* exhibits secondary metamorphosis, analogous to the abrupt ontogenetic changes that are seen at sexual maturity among teleosts. These results provide a point of comparison for testing the congruence between maturity and chronological age, size, and mass, as well as integrating previous work on functional morphology into a rigorous ontogenetic framework. Comparison of

Corresponding author
Thomas D. Carr, tcarr@carthage.edu

the growth series of *T. rex* with those of outgroup taxa clarifies the ontogenetic trends that were inherited from the common ancestor of Archosauriformes.

## INTRODUCTION

Even among large tyrannosaurids, *Tyrannosaurus rex* is an outlier in terms of its gigantic absolute size (*Persons, Currie & Erickson, 2019*; *Snively et al., 2019*) and extremely high bite force (*Bates & Falkingham, 2012*, *2018*; *Cost et al., 2019*; *Gignac & Erickson, 2017*; *Henderson, 2002*), which is clear evidence that ontogeny in this species was carried far beyond the limits seen in its immediate relatives (*Carr, 1999*; *Erickson et al., 2004*). In recent decades, the details of growth in *T. rex* have been greatly expanded from studies of morphology (*Carr, 1999*; *Carr & Williamson, 2004*; *Witmer & Ridgley, 2009*, *2010*), histology (*Erickson et al., 2004*; *Horner & Padian, 2004*; *Woodward et al., 2020*), functional morphology (*Henderson, 2002*; *Henderson & Snively, 2004*; *Snively & Russell, 2003*; *Snively, Henderson & Phillips, 2006*; *Snively et al., 2019*; *Therrien, Henderson & Ruff, 2005*), and mass estimation (*Bates et al., 2009*; *Campione et al., 2014*; *Hutchinson et al., 2011*; *Persons, Currie & Erickson, 2019*). Despite this present heyday of wide-ranging research that has swiftly established *T. rex* as an exemplar fossil species (*Brusatte et al., 2010*), a comprehensive growth series that combines size, chronological age, and maturity (i.e., size-independent characters) (*Brinkman, 1988*) is currently lacking as well as criteria from all three axes of growth for defining its primary growth stages (*Brochu, 1996*; *Carr, 1999*). At present, the congruence between size, age, and maturity are unknown for *T. rex*, and, in that context, how the details of its growth differed from that of other tyrannosaurids.

The use of quantitative cladistic methodology is the most significant advance in studies of ontogeny in extinct species because it can treat size, age, and maturity as independent variables in a single analysis (*Brochu, 1996*). This approach has been used to recover growth series in extant (*Brochu, 1996*; *Tumarkin-Deratzian, Vann & Dodson, 2006*) and extinct (*Carr, 2010*; *Carr & Williamson, 2004*; *Carr et al., 2017*; *Ezcurra & Butler, 2015*; *Frederickson & Tumarkin-Deratzian, 2014*; *Longrich & Field, 2012*) archosaurs, including the tyrannosaurids *T. rex* (*Carr & Williamson, 2004*), *Albertosaurus sarcophagus* (*Carr, 2010*), and *Daspletosaurus horneri* (*Carr et al., 2017*). Although the first cladistic growth series that was published for *T. rex* was based on a small data set limited to 84 craniodental characters in five specimens, it recovered the ontogenetic progression from a long and low skull to a tall skull that is extensively buttressed to resist the loads from a growth-related increase in bite force (*Carr & Williamson, 2004*). As used in this introduction, the terms for growth categories of *T. rex* are based on *Carr & Williamson (2004)*, which recovered the synontomorphies (diagnostic characters) of the juvenile, subadult, and adult growth categories. However, based on the low sample size, this analysis

did not (1) capture the nuances of the continuum of growth, (2) include postcranial characters, (3) include size and mass data, and (4) was not calibrated to the chronological age of individual specimens.

The histology-based study of *Erickson et al. (2004)* defined five growth categories, namely adolescent, juvenile, sub-adult, young adult, and senescent adult. The definitions of these categories were based on the position of specimens along a logistic growth curve, but they were not defined morphologically aside from the number of Lines of Arrested Growth (LAG) and mass estimates. Therefore, independent criteria of maturity are lacking from this study and so growth categories cannot be diagnosed for specimens that lack a mass estimate and LAG number.

In order to overcome those shortcomings, and to answer the call for nonarbitrary criteria for growth stages in nonavian dinosaurs (*Hone, Farke & Wedel, 2016*), 1,850 craniomandibular and postcranial characters, along with size and age data, were analyzed in 44 specimens that span the ontogenetic spectrum from small juvenile (e.g., LACM 28471) to senescent adult (e.g., FMNH PR2081); eight specimens are histologically aged (*Erickson et al., 2004*; *Horner & Padian, 2004*; *Woodward et al., 2020*). The resulting ontogram was mapped onto the growth curve of *T. rex* that was published by *Erickson et al. (2004)* to (1) identify the timing of ontogenetic changes, (2) define growth categories based on all of the available evidence; and (3) set previous work into a comparative framework.

## Terminology

An *ontogram* is the ontogenetic equivalent of cladogram, a branching diagram that shows the nested sets of progressively exclusive growth stages, which is analogous to the phylogenetic hierarchy of clades. Individual *specimens* are positioned at the tips of branches and so are analogs of taxa. A *synontomorphy* is an optimized character that supports a growth stage (i.e., a node) and is the equivalent of a synapomorphy.

A *growth stage* is the ontogenetic equivalent of a set that includes a node, its preceding internode, and its corresponding branch; the term refers to the position of nodes along the ontogram. Growth stages are numbered from the node from which the least mature specimen extends in the ontogram (i.e., closest to the root) to the most mature specimen (i.e., farthest from the root), which is indicated by an arrowhead. In contrast, a *growth category* corresponds to a group that is made up of several growth stages or just one growth stage. Examples of growth categories used in the results section include small juvenile, large juvenile, subadult, young adult, adult, and senescent adult. These categories, aside from the senescent adult, are analogs of phylogenetic grades.

A *growth rank* is used in the correlation tests where a subset of the total number of growth stages is used; for example, not all specimens have maxillae so, in a comparison of maxillary tooth counts, only a subset of the total number of growth stages was used. Therefore, growth ranks will be fewer in number than growth stages. For example, if, among 19 growth stages, there are only nine specimens of nine different growth stages with maxillary tooth counts, there are nine relevant growth ranks available for comparison in a correlation test. *Maturity* in the correlation comparisons refers to either the entire *x*-axis, where maturity increases away from the origin (from immature to mature), or to

the position of a given specimen (immature or mature, relative to the origin and other specimens) among the growth ranks. Maturity also refers to the position of a specimen along the ontogram.

A *corrected rank* (=*midrank*) was noted on the occasions when data points were tied for the same value (i.e., have equivalent ranks) that were converted into midranks following a conventional and straightforward procedure where all initial scores were ranked and then the mean rank (i.e., midrank) was calculated for each set of tied scores; the midranks replace the initial ranks for use in the correlation tests (*Whitlock & Schluter, 2015*). Without this procedure, tied ranks will affect the null distribution of a sample, increase the value of correlation tests, and alter the significance level, and so it is necessary to convert them to midranks (i.e., corrected ranks) to obtain meaningful significance values (*Amerise & Tarsitano, 2015*).

## Goals

The specific goals of this study were to: (1) recover the growth series of *T. rex* using cladistic analysis based on an expanded dataset (relative to *Carr & Williamson, 2004*) that includes chronological age, size, and size-independent cranial and postcranial characters; (2) obtain a synthesis of age, size and maturity data by aligning the cladistic results with a previously published growth curve for *T. rex* (*Erickson et al., 2004*); (3) redefine growth stages based on all three axes of growth (i.e., age, size, maturity); (4) evaluate previously published hypotheses of variation and growth changes in *T. rex* based on the results obtained here; (5) use Spearman rank correlation to quantitatively test hypotheses of congruence between bite force, size, mass, chronological age, tooth count, and abiotic factors with maturity; (6) test the hypothesis that ontogeny recapitulates phylogeny; and (7) test the hypothesis of sexual dimorphism in *T. rex* (e.g., *Larson, 2008*).

## Sequence polymorphism

The presence of sequence polymorphism, or the occurrence of multiple growth patterns in a single taxon, is not explicitly obvious using cladistic ontogeny, the approach used here. The method of Ontogenetic Sequence Analysis (OSA; *Colbert & Rowe, 2008*) is used to identify characters that exhibit sequence polymorphism. The presence of multiple pathways of ontogenetic character change has been reported elsewhere in Amniota, including mammals (*Colbert & Rowe, 2008*) and Archosauria (*Griffin & Nesbitt, 2016a, 2016b*; *Griffin, 2018*). Ergo, *T. rex* is almost certainly not an exception to this general pattern. However, this labor-intensive line of investigation was not pursued here, and is deferred for a later study.

## Assumptions

For the purposes of this study, it was assumed that the assemblage of *T. rex*, which spans Laramidia for a duration of less than 1.0 million years (*Fowler, 2017*), was a single nonanagenetic population. The phylogenetic position of *T. rex* shows that it is nested among taxa endemic to Asia (*T. bataar*, *Zhuchengtyrannus magnus*) and presumably it dispersed from there to Laramidia no earlier than 67 million years ago

(cf. *Brusatte & Carr, 2016*). Unlike preceding Laramidian tyrannosaurid taxa (*Loewen et al., 2013*), *T. rex* was not localized to a single depositional basin during this time of seaway regression and so this widespread taxon is best regarded as a continuous reproductive network that lacks the conditions (i.e., basins) for directional selection over the last million years of the Mesozoic. To test this assumption, an a posteriori comparison between stratigraphic position and relative maturity was made for specimens whose position has been published in the literature.

## MATERIALS AND METHODS

### Data acquisition

A character matrix of 1,850 hypothetical ontogenetically variable characters for 44 specimens was compiled in MacClade (*Maddison & Maddison, 2005*) from first-hand observations of fossils, whereas histological counts of LAGs and some size measurements were obtained from the literature (*Erickson et al., 2004*; *Horner & Padian, 2004*). The character list, table of character states for each specimen, and character matrix are in Data S1–S3, respectively. All fossils included in this study are accessioned in museum or university collections of the United States that are accredited by the American Alliance of Museums (AAM) or are federal repositories, or both. The non-US institutions—NHMUK, ROM, RSM, RTMP—are internationally recognized public trusts. Mass estimates were obtained from several sources (*Bates et al., 2009*; *Hutchinson et al., 2011*; *Campione et al., 2014*; *Snively et al., 2019*; *Persons, Currie & Erickson, 2019*), but were only used in a posteriori analyses. The characters include binary and multistate transformation series, and the coding approach of *Brazeau (2011)* was followed for nested but independent characters.

### Characters

An effort was made to produce an all-inclusive character matrix that drew from as many ontogenetically variable size-independent anatomical domains as possible from the entire skeleton (*Brinkman, 1988*), including: shape, pneumatization, suture closure and form, muscle scar relief, limb proportions, subcutaneous surface texture, fenestrae, skull frame, neurovascular foramina and sulci, ornamentation, endocranial space, etc. (Data S1 and S2). The unusually complete juvenile specimen BMRP 2002.4.1 served as the template for the character matrix, and so the bones missing from it, or characters not drawn from its bones, do not appear in the character matrix, which include the ribs, gastralia, furculum, carpus, manus, and proximal and distal tarsals. Specimens that had multiple states for a given character (e.g., "juvenile" state on the left side, "adult" state on the right) were coded for the mature state.

### Character polarity

The process of polarizing ontogenetic characters is analogous to that used for phylogenetic characters. Whereas an outgroup taxon is used to establish the plesiomorphic condition in phylogeny, the least mature specimen, or a set of least mature specimens, is used as the point of contrast between the immature state of a character from its mature state

(*Brochu, 1996*). The distinction is based primarily on the relative development of features (*Brinkman, 1988*), such as the progression from a bone with a smooth dorsal surface to a bone with a lump on the homologous surface (*Brochu, 1996*).

The juvenile morphotype for tyrannosaurids in general (*Carr, 1999*) and *T. rex* in particular (*Carr, 1999*; *Carr & Williamson, 2004*) is well established. In those studies, the specimens LACM 28471 and CMNH 7541, both referable to *T. rex* (*Carr & Williamson, 2004*), are the most complete juvenile exemplars of the craniomandibular skeleton for comparison with other *T. rex* specimens. These specimens are small (skull length less than 60 cm) and display the nascent state of ontogenetically variable characters (*Carr, 1999*; *Carr & Williamson, 2004*). In addition, histological work has shown that LACM 28471 is young—two years old—in terms in chronological age (*Erickson et al., 2004*). As an example of the procedure, the two specimens share the possession of smooth nasal bones in contrast to the coarse condition that is seen in larger, presumably more mature, specimens. Ergo, the smooth condition is coded with a zero, analogous to the plesiomorphic state of a phylogenetic character, whereas the coarse condition is coded with a one, which is analogous to the apomorphic state. Once a character matrix of independent characters is compiled in this way, the presence of an ontogenetic hierarchy is tested by character congruence, just as the hypothesis of a phylogenetic hierarchy is tested in a conventional cladistic analysis of phylogenetic data. A priori it is not known if the sample of specimens, with their different constellations of hypothetical ontogenetic character states, contain a hierarchical ontogenetic signal: if a signal is present in the data, then an ontogram will be recovered; if signal is absent, then a polytomy will result from the proliferation of multiple competing hypotheses.

The hypothesis that a single tyrannosaurid taxon, namely *T. rex*, is present in the Hell Creek Formation and its lateral equivalents is followed here (*Carr, 1999*; *Carr & Williamson, 2004*; *Woodward et al., 2020*). As such, several juvenile specimens that are represented by an isolated tooth (e.g., DDM 1863.1) or bone (e.g., RSM 2347.1) were included in the analysis even if *T. rex* autapomorphies were absent from those specimens, but otherwise they compared closely with LACM 28471 and CMNH 7541. Likewise, isolated bones from adults (e.g., CM 9401) that did exhibit *T. rex* autapomorphies were included in the analysis.

## Character matrix

An initial character matrix of 1,851 characters and 45 specimens was compiled in MacClade; upon close scrutiny, one transformation series was found to lack hierarchical structure (i.e., all specimens shared the same code) that was discarded, which reduced the number of characters to 1,850. Also, a tibia (MOR 3028) was coded for only one character and so it was excluded from the analysis, reducing the number of specimens to 44.

The character matrix that was analyzed includes two sections: the first 214 characters were drawn from the phylogenetic literature (*Rauhut, 2003*; *Carrano, Benson & Sampson, 2012*; *Carr et al., 2017*; Data S1–S3), and the rest were based on comparisons made between *T. rex* specimens, including a subset that was drawn from the ontogenetic

literature (*Carr, 1999*; *Carr & Williamson, 2004*). Among the phylogenetic characters, juvenile specimens tended to be coded with the plesiomorphic character states, whereas mature specimens were coded with the apomorphic character states. In view of this pattern, the analysis was used to test the hypothesis that ontogeny recapitulates phylogeny. A total of 211 multistate characters were ordered (Data S1).

## Dorsotemporal fossa

The dorsotemporal fossa is an important osteological feature and its anatomical interpretation affects character definition and construction. In a recent publication (*Holliday et al., 2019*) it was hypothesized that the rostral part of the dorsotemporal fossa that covers the frontal bone in tyrannosaurids was occupied by vascular tissue—not adductor musculature. Five lines of evidence can be brought against this hypothesis. First, the entire surface of the fossa has a coarse, muscle scar-like texture that is continuous between the parietal, frontal, and postorbital; on occasion the rostral edge of the fossa is elaborated into a coarse ridge (e.g., OMNH 10131-1) and the sagittal crest of the frontal is likewise coarse from muscle scarring. As such, the entire surface is an extensive, bowl-like muscle scar that is similar to the large, discrete muscle scars that are seen elsewhere in the skeleton, such as the semicircular scar on the dorsolateral surface of the ischium or the oval-shaped posterior scars of the femur in tyrannosaurids. Therefore, the claim that "the (dorso)temporal fossa does not bear any…osteological feature that clearly identif[ies] it as a muscular attachment" is not true (*Holliday et al., 2019*: 9).

Second, the authors point out that the muscles hypothesized to originate from the dorsotemporal fossa (superficial pseudotemporal, external deep mandibular adductor) must wrap rostroventrally around the caudal edge of the laterosphenoid buttress to insert onto the mandibular ramus, forming an acute angle between the origin and the insertion of the muscle (*Holliday et al., 2019*). The authors claim that this acute angle "would render [the muscles] functionally equivocal if not entirely useless" (*Holliday et al., 2019*: 10). However, a hairpin turn in a functional cephalic muscle is seen elsewhere in Archosauria, namely the ventral pterygoid muscle that originates from the palate, which in crocodylians "conspicuously wraps around (the dorsal pterygoid muscle) and the retroarticular process (of the mandibular ramus) to attach to the caudolateral surface of the angular" (*Holliday & Witmer, 2007*: 465; *Gignac & Erickson, 2017*: Fig. 2) and in several clades of living birds it "attaches to the lateral surface of the mandible similar to the condition found in crocodylians" (*Holliday & Witmer, 2007*: 467). Therefore, there is precedent for an important cephalic muscle that makes a hairpin turn around a bone without the mediating presence of a sesamoid or trochlea (cf. *Holliday et al., 2019*). In a similar fashion, the superior oblique of the human eye takes an acute turn through a soft tissue trochlea before it inserts onto the eyeball (*Agur & Lee, 1991*). It is possible that the turn of the adductor musculature in tyrannosaurids is not as extreme as hypothesized: a digital model of *T. rex* adductor muscles that was used to precisely estimate bite forces in the taxon shows neither a sharp nor acute turn from the fossa to the mandibular ramus (*Gignac & Erickson, 2017*: fig. 2B). Even if the digital model is inaccurate, a sharp bend can also be taken by a muscle in the postcranium; for example, in mammals the

popliteus originates on the lateral surface of the femur from which it wraps posteromedially onto the posterior surface of the tibia (e.g., *Felis domesticus*: *De Iuliis & Pulerà, 2019*; *Homo sapiens*: *Van de Graaff, 1998*).

Third, manipulation of a cast of a tyrannosaurid skull shows that the surangular shelf and the coronoid process are situated below the caudal region of the subtemporal fenestra of the skull; that is, below the channel bounded rostrally by the frontal, postorbital, and laterosphenoid, and caudally by the parietal, squamosal, and quadrate. Regardless, the adductor scar in tyrannosaurids extends further rostrally on the dorsolateral surface of the surangular, to a position below the caudal end of the orbital fenestra; the location of the scar shows that it was produced by a muscle or muscles that extended rostroventrally whether they originated from the dorsotemporal fossa or not. Dissections do show that in crocodylians and birds that it is the superficial pseudotemporal and the deep external mandibular adductor muscles (*Holliday & Witmer, 2007*: fig. 10B–D) that are the rostralmost to insert onto the mandible, a condition inherited by tyrannosaurids from their common archosaurian ancestor. Therefore, the identification of the superficial pseudotemporal and deep external mandibular adductor muscles as the rostralmost jaw adductors requires a turn around the shelf formed by the frontal and postorbital.

Fourth, the authors observe that "the caudal edge of the frontoparietal fossa (or rostral edge of the dorsotemporal fossa) is angled sharply vertically creating a physical obstacle for a muscle belly to cross, rather than the excavated, concave fossa one would expect to find where a muscle belly was passing" (*Holliday et al., 2019*: 9). This observation is correct, but it is important to point out that the fossa formed by the frontal and postorbital is bowl-like, bounded medially by the prominent sagittal crest and laterally by a vertical wall formed by the postorbital. Also, in some amniotes, such as mammals, the fossa is convex, so a concave surface is not required to anchor the jaw-closing muscles (e.g., *Canis familiaris*). The caudal edge of the frontopostorbital shelf (=laterosphenoid buttress) is merely an abrupt plane change that allows muscles to extend unobstructed ventrally to their insertions. Analogous abrupt edges are seen in the temporal fossae of mammals and turtles above the mandibular fossa. Therefore, an abrupt edge does not constitute an obstacle, rather, it provides a means for a muscle to extend from the origin to reach the insertion.

Fifth, the authors point out that the "frontoparietal fossa in some larger tyrannosaurids…are perforated by numerous foramina and erosional pits in the skull roof" (*Holliday et al., 2019*: 11) as evidence for a vascular structure in the fossa instead of muscle. Neurovascular foramina do penetrate the dorsotemporal fossa of tyrannosaurids, a condition that is seen in other amniotes, such as mammals, where the temporal fossa is unambiguously a large muscle origin. Also, the erosional pits are a part of the overall mottled texture of the dorsotemporal fossa of tyrannosaurids, analogous to the wrinkled surface of the fossa seen in other amniotes; again, mammals provide convenient examples (e.g., *C. familiaris*). Therefore, for these reasons the vascular tissue hypothesis is rejected here, and the entire fossa on the frontal is instead regarded as the origin for the adductor musculature. This distinction of causal process (i.e., remodeling associated with a vascular structure or a muscle origin) a priori affects how the characters of the

dorsotemporal fossa of the frontal are conceptualized or coded and, a posteriori, affects hypotheses when accounting for how the differences seen between different growth stages are integrated with the rest of the adductor surface of origin.

Finally, it is worth pointing out that the image showing the adductor muscle insertions onto the medial surface of the postdentary moiety of a specimen of *T. rex* (MOR 008) in *Holliday (2009*: fig. 4K*)* is upside down, which the labels for the insertions do not take into account. This misorientation has resulted in a complete mislabeling of the moiety; also, it far too rostrally positions the insertion of the deep external adductor mandibular muscle, which is marked by an inflection point at the rostral end of the muscle scar on the dorsolateral surface of the surangular (the ventral edge of the bone in the photograph). The corrected, caudalward position would reduce the angle of the hairpin turn made by the muscle from the cranium to the mandible.

## Size

In order to test the hypothesis that ontogeny is congruent with phylogeny, size characters were divided into discrete states to match their phylogenetic homologs (*Brusatte & Carr, 2016*; *Carr et al., 2017*). Where possible, skull length (premaxilla to quadrate) was used as the measurement of absolute size; in the absence of complete skulls, the length of the ilium, or, failing that, the femur was used. The length of the ilium and femur have been shown to closely approximate the length of the skull (*Currie, 2003*), and so they are used here as proxies when a skull length is unavailable.

In addition to that, relative size was used only when a specimen is represented by a single bone or a partial skull or skeleton. For example, the isolated lacrimal FMNH PR2411 is smaller than that of CMNH 7541, a juvenile, and so it is reasonable to assume that when complete the skull, of which the single bone was once a part, was smaller than the larger skull. However, an absolute size cannot be given since allometric trends among small juveniles of *T. rex* are currently unknown.

## Analysis

Following *Brochu (1996)*, an artificial embryo was included to optimize the transformation series on the topology; the codes for the artificial embryo were based on the least mature character states seen in juvenile specimens. The character matrix (S3) was analyzed in TnT v. 1.5 (*Goloboff, Farris & Nixon, 2003*; *Goloboff & Catalano, 2016*) under a driven New Technology search using the default parameters for ratchet, tree drift, tree fusion, and sectorial search, and 10 replicates were run to find a minimum length ontogram. The ontograms obtained from these results were then run under a Traditional Search. In the first analysis that included all 44 specimens, 50 ontograms of 3,099 steps were recovered, and a strict consensus ontogram recovered only three nodes (Fig. 1A).

A systematic approach was taken to identify wildcard specimens by analyzing specimens in order of decreasing completeness (Table 1); that is, the three most complete specimens (BMRP 2002.4.1, FMMH PR2081, MOR 1125) were analyzed together, which recovered a single ontogram. In the next analysis, the tree buffer was cleared and next least complete specimen (LACM 150167) was added to the analysis and so on. Specimens
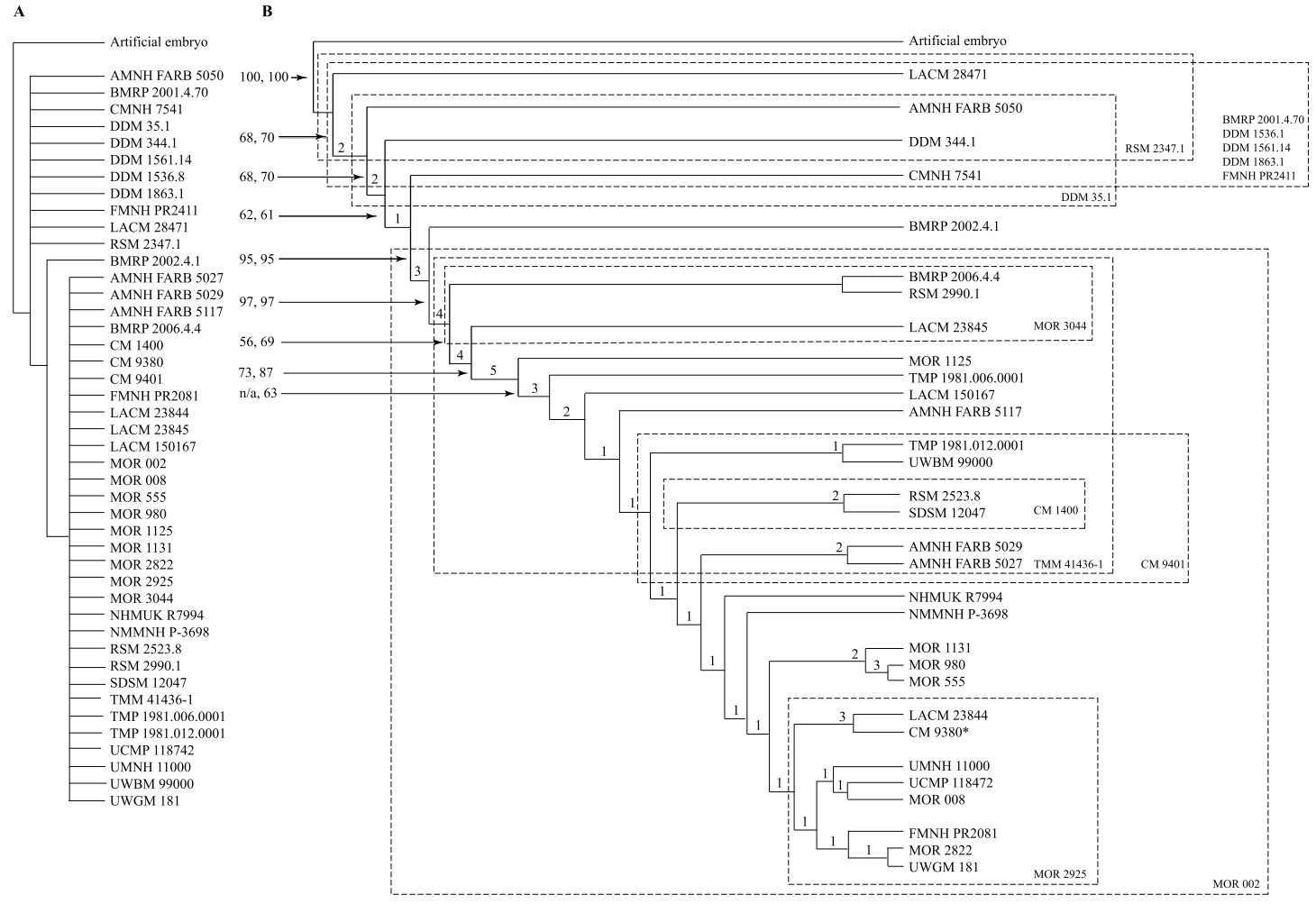

**Figure 1 Results of the cladistic analysis of 1,850 characters among 44 specimens of *Tyrannosaurus rex*.** (A) Strict consensus of 50 MPTs showing the recovery of three primary growth stages separated by the specimen BMRP 2002.4.1. (B) The single ontogram recovered after the exclusion of wildcard specimens, reducing the number of OTUs to 31. Numbers to the left of the internodes are bootstrap and jackknife values, respectively; numbers to the right are Bremer decay indices. Asterisk indicates the type specimen. Ellipses enclose the regions of polytomies produced by the wildcard specimens, which are listed in the lower right hand corner of the corresponding ellipse. Note that the ellipses are limited to one side or the other relative to BMRP 2002.4.1, which corresponds to the topology of the strict consensus ontogram.

that resulted in multiple ontograms (i.e., wildcards) were identified and excluded from subsequent analyses (Table 1). Several specimens initially resulted in multiple equally parsimonious ontograms but later resulted in a single ontogram after additional specimens were included. Two rounds of this process identified 13 wildcard specimens (Table 1) that ranged in completeness from 6.1% to 0.2% of characters coded (i.e., over 90% missing information). Specimens that were coded for less than 1.5% of the characters resulted in multiple topologies, indicating the lower limit of information content on topological resolution in this data set (Table 1).

In the end, the remaining 31 specimens resulted in a single most parsimonious ontogram of 3,053 steps, which was then tested under a Traditional search using TBR branch swapping in case a more parsimonious tree island was missed by the New

**Table 1 Summary of completeness of the *Tyrannosaurus rex* specimens included in this analysis with identification of the wildcard specimens.** Summary of the completeness of the specimens of *T. rex* included in this study, given as the percentage of missing characters and the percentage of characters present out of a total of 1,850. The list is given in descending order, where the least complete specimen is at the top of the table and the most complete is at the bottom. Grey fill indicates the 13 wildcard specimens.

| Specimen | % Missing characters | % Present characters |
|---|---|---|
| BMRP 2001.4.70 | 99.8 | 0.2 |
| DDM 1863.1 | 99.7 | 0.3 |
| DDM 1562.14 | 99.6 | 0.4 |
| MOR 002 | 99.4 | 0.6 |
| DDM 1536.8 | 98.9 | 1.1 |
| DDM 35.1 | 98.6 | 1.4 |
| TMP 1981.012.0001 | 98.2 | 1.8 |
| AMNH FARB 5050 | 98.2 | 1.8 |
| CM 9401 | 98.1 | 1.9 |
| UMNH 11000 | 97.8 | 2.2 |
| UCMP 118742 | 97.6 | 2.4 |
| RSM 2347.1 | 97.3 | 2.7 |
| TMM 41436-1 | 97.1 | 2.9 |
| MOR 2925 | 96.5 | 3.5 |
| DDM 344.1 | 96.3 | 3.7 |
| FMNH PR2411 | 94.6 | 5.4 |
| RSM 2990.1 | 94.6 | 5.4 |
| MOR 3044 | 94.4 | 5.6 |
| NMMNH P-3698 | 94.2 | 5.8 |
| CM 1400 | 93.9 | 6.1 |
| AMNH FARB 5029 | 93.4 | 6.6 |
| BMRP 2006.4.4 | 93.2 | 6.8 |
| MOR 1131 | 92.9 | 7.1 |
| NHMUK R7994 | 92.6 | 7.4 |
| UWGM 181 | 91.5 | 8.5 |
| LACM 28471 | 89.6 | 10.4 |
| AMNH FARB 5117 | 88.2 | 11.8 |
| SDSM 12047 | 87.7 | 12.3 |
| MOR 2822 | 83.3 | 16.7 |
| LACM 23845 | 83.2 | 16.8 |
| TMP 1981.006.0001 | 82.4 | 17.6 |
| CM 9380 | 75.7 | 24.3 |
| CMNH 7541 | 71.8 | 28.2 |
| LACM 23844 | 62.4 | 37.6 |
| AMNH FARB 5027 | 60.8 | 39.2 |
| RSM 2523.8 | 59.5 | 40.5 |
| MOR 008 | 57.6 | 42.4 |
| MOR 555 | 52.7 | 47.3 |

| | Table 1 (continued). | |
| --- | --- | --- |
| Specimen | % Missing characters | % Present characters |
| UWBM 99000 | 49.3 | 50.7 |
| LACM 150167 | 44.1 | 55.9 |
| MOR 980 | 42.4 | 57.6 |
| MOR 1125 | 42.2 | 57.8 |
| FMNH PR2081 | 15.7 | 84.3 |
| BMRP 2002.4.1 | 13.3 | 86.7 |

Technology analysis (Fig. 1B). This analysis returned a single most parsimonious ontogram of the same topology. Descriptive tree statistics and the apomorphy list were obtained by running the analysis under a heuristic search in PAUP (*Swofford, 2002*) under ACCTRAN optimization with the ontogram length and ontogram topology (assembled in MacClade; *Maddison & Maddison, 2005*) constrained.

## Artificial adult

An attempt to identify the most mature specimen of the sample was made by adding an artificial adult to the character matrix using the following procedure: the ontogram obtained from the TnT analysis was reconstructed in MacClade (*Maddison & Maddison, 2005*) and the character trace tool was used to display the optimization of each transformation series on the topology. Since the ontogram follows a gradient from least mature (toward the root) to most mature (away from the root), the node furthest from the root served as a point of reference. The furthest node splits into two subgroups (UMNH 11000 + UCMP 118472 + MOR 008 on the one hand, and FMNH PR2081 + MOR 2822 + UWGM 181 on the other; Fig. 1B), and, a priori, it could not be determined which group is more mature than the other. Therefore, the artificial adult was coded based on the character state(s) optimized at the internode supporting that group. For example, if the internode was optimized as "1", then the artificial adult was coded accordingly; if it was optimized ambiguously as "0" and "1", the artificial adult was coded with both states.

Once the codings for the artificial adult were completed, the analysis was again run and the resulting sister specimen of the artificial adult was identified as the most mature specimen in the sample. That specimen would then function as the terminal exemplar of the growth series and its autontomorphies (i.e., individual variation) would be considered to represent the last changes in growth.

## Statistical tests

Statistical tests were completed by using the licensed software package IBM SPSS Statistics version 24.0.0.0. (*IBM Corp., 2016*). For each Spearman rank correlation comparison, the growth series ranks were compared with ranks converted from each data set; the variables were treated as ordinal, and a two-way Spearman test was run. For data sets where congruence was seen among juveniles and subadults, but not among adults, a separate test was run for the mature specimens that excluded the immature specimens.

## Chronological age data

Four sources from the literature were used to obtain chronological age for eight specimens (*Erickson et al., 2004*; *Erickson, 2005*; *Horner & Padian, 2004*; *Woodward et al., 2020*). The overall age estimates of *Horner & Padian (2004)* were used. The chronological age of BMRP 2002.4.1 is based on *Woodward et al. (2020)*, which reported a higher estimated age of 13 years than the earlier published estimate of 11 years (*Erickson, 2005*); the *Woodward et al. (2020)* estimate was based on a wider sampling of the skeleton (e.g., femur and tibia) than that of *Erickson (2005)*, which sampled the fibula. The chronological age of 14 years for MOR 555 (*Horner & Padian, 2004*), an unambiguous adult, was excluded from the comparisons made here because this underestimate was based on a damaged bone (J. Horner, 2010, personal communication).

## RESULTS

### Cladistic analyses

The reduced cladistic analysis of 31 specimens, following the New Technology and Traditional searches, recovered one most parsimonious tree (i.e., ontogram) of 3,053 steps, with an ensemble Consistency Index (CI) excluding uninformative characters of 0.65, an ensemble Homoplasy Index (HI) of 0.35, an ensemble Retention Index (RI) of 0.72, and an ensemble Rescaled Consistency Index (RCI) of 0.50. The ontogram recovered 21 growth stages; in the adult region of the ontogram several branches contain multiple specimens; these are most simply interpreted as specimens of the same maturity (Fig. 2). Bremer, jackknife, and bootstrap values are shown in Fig. 1B.

The ontogram is composed of 21 growth stages, including the group of most mature specimens, and all but the first and third are supported by unambiguously optimized synontomorphies (Data S4). Despite the addition of 26 specimens and 1,766 characters, these results are congruent with those obtained by *Carr & Williamson (2004)*, where LACM 28471, CMNH 7541, LACM 23845, AMNH FARB 5027, and LACM 23844 were again recovered at progressively mature growth stages (Figs. 1B and 2).

The a posteriori analysis that included an artificial adult resulted in seven 3,071-step MPTs that, in a strict consensus ontogram did not group with a single, presumably most mature, specimen. Therefore, a different approach was taken, specifically the greatest distance from the root. Of that group, FMNH PR2081 possessed the greatest number of autontomorphies, 19 character changes, in contrast to the five others; ergo, that specimen was regarded as the most mature of the group, having undergone the greatest amount of change, and so represents the twenty-first growth stage (i.e., the terminus of the growth series). In contrast, the massive specimen RSM 2523.8, previously regarded as the most mature individual *T. rex* (*Persons, Currie & Erickson, 2019*), was recovered as one of the least mature adults.

The two purported female specimens (BMRP 2006.4.4, MOR 1125), a sex identification based on the presence of femoral medullary bone (*Schweitzer, Wittmeyer & Horner, 2005*; *Woodward et al., 2020*; this assessment has been challenged by *O'Connor et al., 2018*), were recovered as a subadult and a young adult, respectively (Fig. 2). Sexual dimorphism

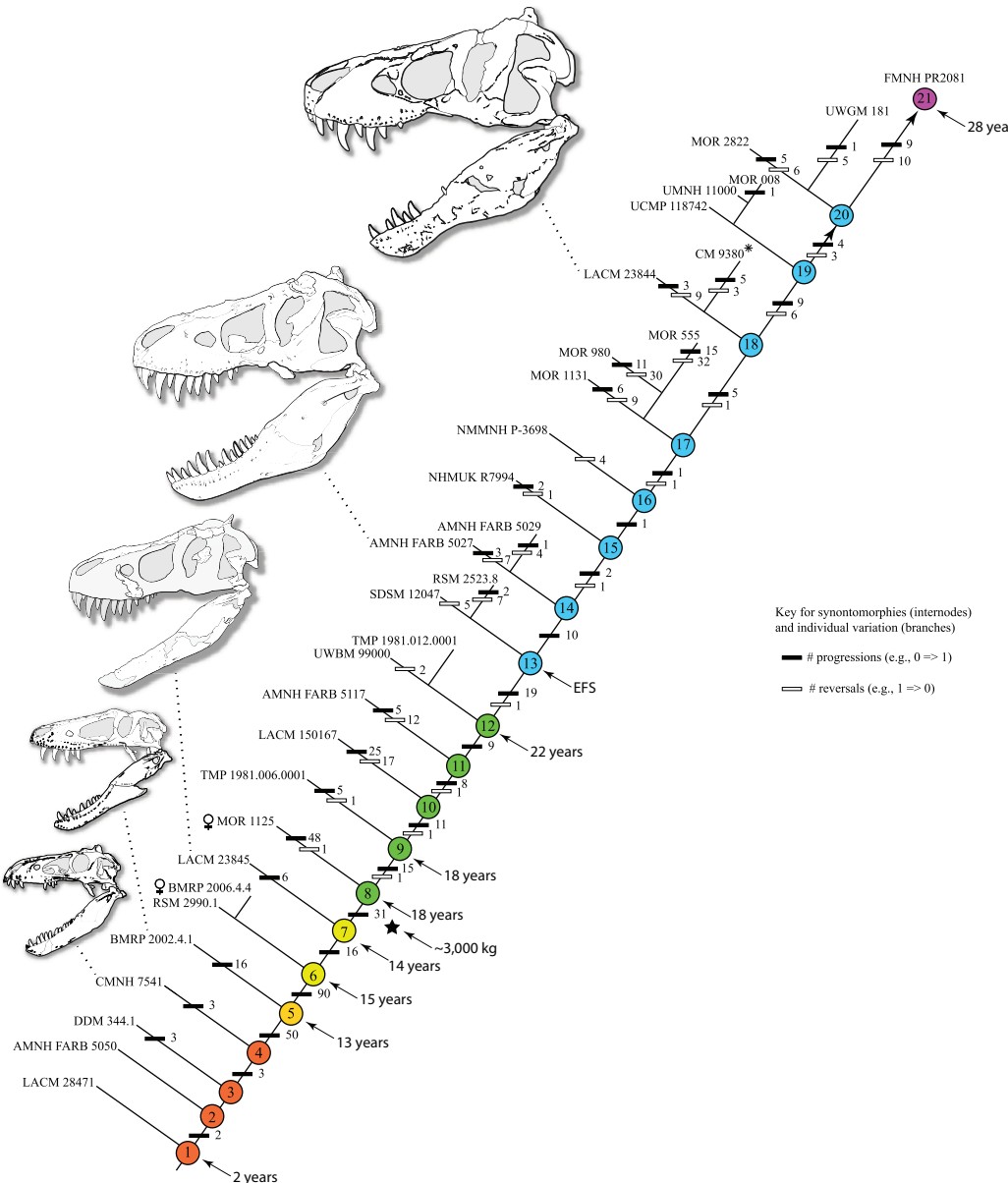

**Figure 2 Ontogram of *Tyrannosaurus rex* showing growth stages, synontomorphies, individual variation, individual specimens, and chronological ages.** Arrowhead points to the most mature specimen and the direction of the entire ontogenetic axis; that is, the least mature specimen is at the lower left whereas the most mature specimen is at the upper right. Asterisk indicates the type specimen. Individual variation occurs as progressions until young adulthood, where reversals are first seen. The maximum amount of change occurs at growth stages 5 and 6, which corresponds to the transition from a long and low skull and jaws to a deep and stout skull frame; this event, marked by the concentration of an extreme number of changes, is evidence that the ontogeny of *T. rex* is metamorphic (sensu *Rose & Reiss, 1993*). Each circle represents a numbered growth stage; these numbers do not correspond to those seen in Fig. 12. The star at growth stage 7 marks the ~3,000 kg threshold that separates *T. rex* from its closest, but smaller, relatives. Color key: red, small juveniles; orange, large juveniles; yellow, subadults; green, young adults; blue, adults; violet, senescent adults. See text for definition of growth categories. Skulls are to scale; AMNH FARB 5027 is scaled to a premaxilla to quadrate length of 1.3 m.

**Table 2 An assessment of evidence for sexual dimorphism in *Tyrannosaurus rex* based on character states that are unambiguously optimized as individual variation.** The female specimen MOR 1125 shares homologous character transformations with five other specimens, but these shared changes are not common to all of the specimens, which would be expected if they represented diagnostic character states unique to one sex. Based on this lack of pattern it is inferred that character states unique to females are absent and, by extension, the skeleton of *T. rex* is not sexually dimorphic.

| MOR 1125 | CMNH 7541 | LACM 150167 | MOR 555 | BMRP 2002.4.1 | AMNH FARB 5117 |
|---|---|---|---|---|---|
| 330. Maxilla, subnrl frmn, dpth, 0 => 1 | | 330, 0=>1 | | | |
| 419. Maxilla, js for jgl, rstrl extnt, 0 => 1 | | | 419, 0=>1 | | |
| 449. Lacrimal, drsl rms, sbctns srfc, hght, 0 => 1 | | | | 449, 0=>1 | |
| 618. Postorbital, drsl mrgn, ornnttn, 1 => 2 | | 618, 0=>1 | | | |
| 728. Quadratojugal, sq pr, rstrl srfc, 1 => 2 | | | 728, 1=>2 | | |
| 963. Parietal, sggtl crst, cdl end, extnt, 0 ==> 1 | 963, 0 ==> 1 | | | | |
| 973. Parietal, prtsoc pllr, prsnc, 1 => 2 | | | | | 973, 1=>2 |
| 1022. Basioccipital, occ con, cdl mrgn, 0 => 1 | | | | | 1022, 0=>1 |

was presumed here to be expressed in one of two ways: either the ontogram would split into separate male and female branches, or sexually diagnostic characters might be optimized as individual variation.

In the first instance, a split of the ontogram into separate branches is not seen. In the second, the individual variation of MOR 1125 was compared with that of other specimens. Sexually informative variation, in this case female variation, was assumed to be expressed as multiple homologous character changes seen in MOR 1125 and a repeated set of specimens (i.e., the identical set of specimens for each shared homologous character change). The comparison found eight homologous character changes shared between MOR 1125 and five other specimens; however, the character changes are not shared with a uniform set of specimens (Table 2). Therefore, there is no skeletodental evidence for sexual dimorphism in the data set (i.e., males and females are skeletally identical aside from the presence of medullary bone); otherwise, the ontogram would have divided into separate male and female branches or homologous character changes would have singled out MOR 1125 and an associated set of specimens. This test is not dependent upon MOR 1125 being a female (*O'Connor et al., 2018*); if in actuality it is a male, then the set of shared characters would be evidence of that sex.

## Wildcard specimens

Thirteen wildcard specimens were identified; in each case, the analyses that included them resulted in multiple ontograms that collapsed into one or several polytomies; the regions of collapse are enclosed by ellipses in Fig. 1B. Upon comparison of the polytomies, the wildcard specimens could be divided into two groups, where resolution was lost either rootward or distalward relative to BMRP 2002.4.1, the most complete specimen in the dataset; that is, that specimen was not part of either polytomy. The partial adult specimen, CM 1400, is shown in a polytomy with two other specimens at the end of a branch (Fig. 1B); however, structure was lost elsewhere in the ontogram when this specimen was included in the analysis, and so it was excluded from the backbone topology (Figs. 1B and 2).

Despite the fact that the wildcards are coded for less than 7% of the characters, some occupy a relatively precise location of the ontogram (e.g., CM 1400, CM 9401, DDM 35.1, MOR 3044, MOR 2925, RSM 2347.1), whereas others collapse entire regions (e.g., BMRP 2001.4.70, DDM 1536.1, DDM 1562.14, DDM 1863.1, FMNH PR2411, MOR 002, TMM 41436-1). Of these specimens, many are single bones (BMRP 2001.4.70, DDM 35.1, DDM 1536.8, FMNH PR2411, MOR 2925, MOR 3044, RSM 2347.1, TMM 41436-1) or teeth (DDM 1562.14, DDM 1863.1), whereas only two are a partial skull (CM 1400) or skull and skeleton (MOR 002).

## Overview of frequencies of growth changes
### Influence of specimen completeness
The effect of specimen completeness upon the number of synontomorphies at each node was quantified through a Spearman rank correlation test. Examination of the raw data as histograms with a fitted normal curve showed they are positively skewed, and a Shapiro–Wilk test of normality found that they were not normally distributed (completeness, $p = 0.001$; synontomorphies, $p = 0.000$) and so the data were normalized by converting them to ranks (Table 3; Fig. 3). A Shapiro-Wilk test of normality found that the ranked completeness and ranked number of synontomorphies were normally distributed ($p = 0.217, 0.134$, respectively). A Spearman rank correlation test resulted in a nonsignificant ($p = 0.423$) correlation coefficient ($r_S = 0.149$), indicating that the peaks of synontomorphies at each node are not dependent upon the completeness of the specimens that extend from each branch.

### Synontomorphy trends
The overall distribution of growth changes is unimodal (Fig. 4): the highest number occurs early, peaking at the subadult growth category, which then precipitously drops, aside from several low peaks in the adult growth stages. In total, there are five peaks, and four of them mark the onset of growth categories. The peak at growth stage 6 marks the beginning of the subadult growth category, where an increase in the height of the skull frame and inflation of the bones that enclose the antorbital sinus are seen, as well as changes to the pectoral girdle and limb, and pes. The peak at growth stage 8 marks the onset of the young adult growth category. The peak at growth stage 13 marks the onset of adulthood, where the External Fundamental System (EFS; a narrow band of lines of arrested growth that indicate near cessation of appositional growth) is first seen as well as changes to the antorbital sinus system and the origin of the adductor musculature. The peak at growth stage 19 marks extensive changes to the skull and postcranial skeleton, but by itself does not define a new growth category. Finally, the peak of changes at growth stage 21 correspond to the extensive changes in the skeleton of FMNH PR2081, the most mature and second-best sampled specimen in the data set.

Given their abundance in the character matrix, nonphylogenetic synontomorphies are more numerous at each node than phylogenetic synontomorphies, but they broadly follow the same pattern of peaks and valleys (Fig. 5), aside from growth stages 6, 8, 10, and 21, where phylogenetic changes tend to decrease in frequency whereas nonphylogenetic

**Table 3 Comparison of the completeness of *Tyrannosaurus rex* specimens included in the backbone ontogram and the number of synontomorphies at each node.** Raw data are in normal typeface, ranks used in the correlation test are in boldface.

| Specimen | Specimen completeness | Specimen completeness rank | Specimen completeness midranks | # Synontomorphies at node | #Synontomorphies rank | # Synontomorphies midranks |
|---|---|---|---|---|---|---|
| BMRP 2002.4.1 | 86.7 | **1** | **1** | 51 | **3** | **3** |
| FMNH PR2081 | 84.3 | **2** | **2** | 7 | **18** | **19** |
| MOR 1125 | 57.8 | **3** | **3** | 31 | **4** | **4** |
| MOR 980 | 57.6 | **4** | **4** | 2 | **25** | **26.5** |
| LACM 150167 | 55.9 | **5** | **5** | 12 | **12** | **12** |
| UWBM 99000 | 50.7 | **6** | **6** | 9 | **15** | **16** |
| MOR 555 | 47.3 | **7** | **7** | 2 | **26** | **26.5** |
| MOR 008 | 42.4 | **8** | **8** | 15 | **8** | **9.5** |
| RSM 2523.8 | 40.5 | **9** | **9** | 20 | **5** | **5.5** |
| AMNH FARB 5027 | 39.2 | **10** | **10** | 10 | **13** | **13.5** |
| LACM 23844 | 37.6 | **11** | **11** | 6 | **21** | **21.5** |
| CMNH 7541 | 28.2 | **12** | **12** | 3 | **23** | **23.5** |
| CM 9380 | 24.3 | **13** | **13** | 6 | **22** | **21.5** |
| TMP 1981.006.0001 | 17.6 | **14** | **14** | 16 | **7** | **7** |
| LACM 23845 | 16.8 | **15** | **15** | 15 | **9** | **9.5** |
| MOR 2822 | 16.7 | **16** | **16** | 7 | **19** | **19** |
| SDSM 12047 | 12.3 | **17** | **17** | 20 | **6** | **5.5** |
| AMNH FARB 5117 | 11.8 | **18** | **18** | 9 | **16** | **16** |
| LACM 28471 | 10.4 | **19** | **19** | 0 | **30** | **30.5** |
| UWGM 181 | 8.5 | **20** | **20** | 7 | **20** | **19** |
| NHMUK R7994 | 7.4 | **21** | **21** | 3 | **24** | **23.5** |
| MOR 1131 | 7.1 | **22** | **22** | 2 | **27** | **26.5** |
| BMRP 2006.6.4 | 6.8 | **23** | **23** | 89 | **1** | **1.5** |
| AMNH FARB 5029 | 6.6 | **24** | **24** | 10 | **14** | **13.5** |
| NMMNH P-3698 | 5.8 | **25** | **25** | 1 | **29** | **29** |
| RSM 2990.1 | 5.4 | **26** | **26** | 89 | **2** | **1.5** |
| DDM 344.1 | 3.7 | **27** | **27** | 0 | **31** | **30.5** |
| UCMP 118742 | 2.4 | **28** | **28** | 15 | **10** | **9.5** |
| UMNH 11000 | 2.2 | **29** | **29** | 15 | **11** | **9.5** |
| AMNH FARB 5050 | 1.8 | **30** | **30.5** | 2 | **28** | **26.5** |
| TMP 1981.012.0001 | 1.8 | **31** | **30.5** | 9 | **17** | **16** |

changes increase. This difference in frequency distribution shows that the number of phylogenetic changes is not controlled by the number of characters scored. Phylogenetic changes are frequent (i.e., more than five) early in growth (stages 2–9), whereas they are less common among adults (stages 10, 12, 13, 18–21); if ontogeny is congruent with phylogeny then this gross pattern should be expected since fewer phylogenetic synontomorphies should occur among adults, which presumably would only express characters at the level of species.
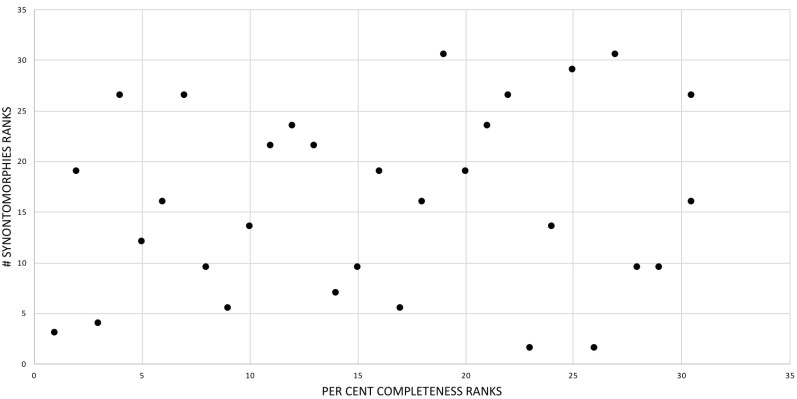

**Figure 3 Scatterplot showing the noncongruence in *Tyrannosaurus rex* between the completeness of specimens (i.e., number of characters scored) and the number of synontomorphies at each corresponding node.** Per cent completeness (decreasing away from the origin) and the number of synontomorphies supporting the corresponding node (decreasing away from the origin) have been converted to ranks. A Spearman correlation test on these data results in a nonsignificant correlation coefficient; ergo, the number of synontomorphies at an internode is not an artifact of specimen completeness.

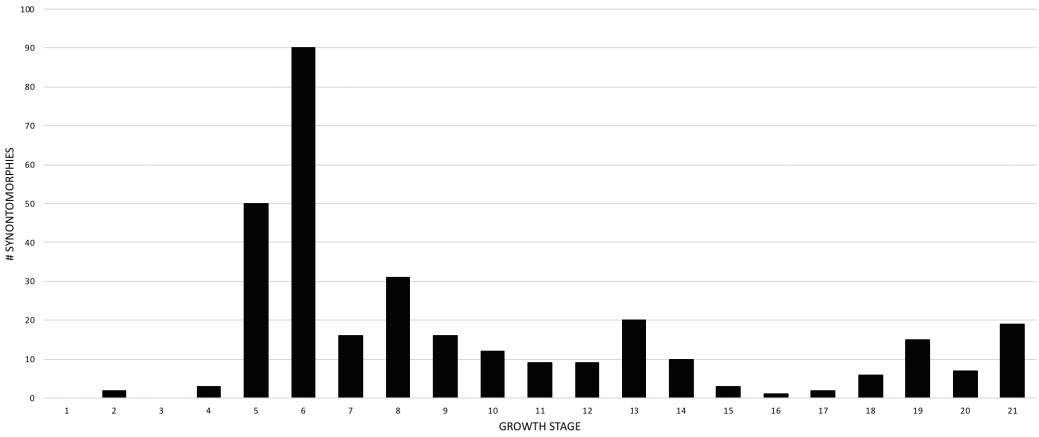

**Figure 4 Frequency distribution of unambiguously optimized synontomorphies during the growth of *Tyrannosaurus rex*.** Growth stages (corresponding to the numbered nodes of the ontogram in Fig. 2) are along the *x*-axis and the number of changes are along the *y*-axis. The greatest number of changes are seen in the transition from large juvenile to subadult, or, from growth stage 5–6; the high concentration of change between these growth categories is evidence that *T. rex* ontogeny is metamorphic (sensu *Rose & Reiss, 1993*).

A comparison of the frequency of cranial and postcranial changes shows that cranial synontomorphies are the most frequent (Fig. 6). Both sets of changes follow the same general pattern; cranial changes are dominant in terms of number and the pattern of the frequency distribution, whereas postcranial changes do not always occur. Cranial and postcranial changes are most frequent at growth stage 6, and thereafter they show less than 22 changes per growth stage. This indicates that once the adult morphotype (i.e., tall skull, inflated antorbital sinuses) is achieved, the rate of change, as shown by the number of changes per growth stage, greatly decreases. During adulthood, cranial changes generally outnumber postcranial changes, which tend to cease altogether. Cranial changes are
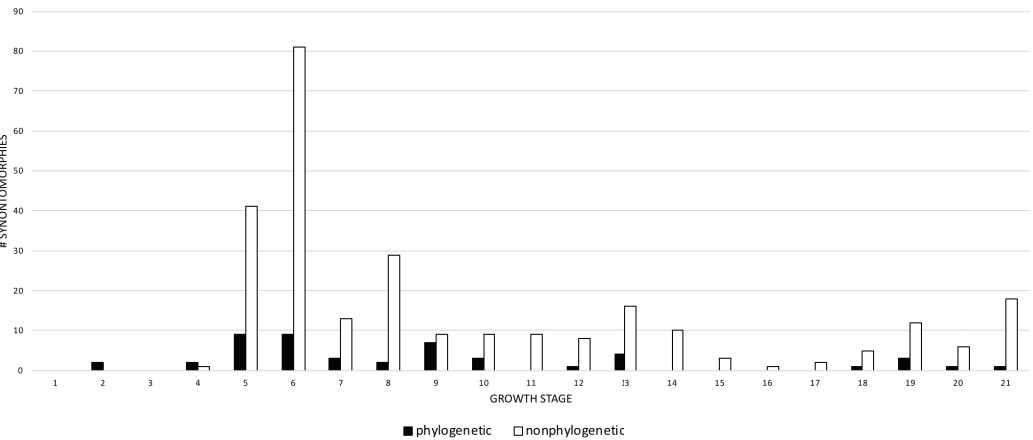

**Figure 5 Comparison of the frequency distributions of phylogenetic and nonphylogenetic synontomorphies in the ontogeny of *Tyrannosaurus rex*.** Growth stage is along the *x*-axis (corresponding to the numbered nodes of the ontogram in Fig. 2) and number of synontomorphies is along the *y*-axis. Phylogenetic characters are in solid bars; nonphylogenetic characters are in hollow bars. The frequency distributions of both sets of data follow the same general pattern, aside from the flatter distribution of the phylogenetic synontomorphies relative to the nonphylogenetic synontomorphies and the reversed pattern seen at growth stages 7 and 8. Both types of changes occur throughout the lifespan of *T. rex*, indicating that ontogeny is not strictly congruent with phylogeny.

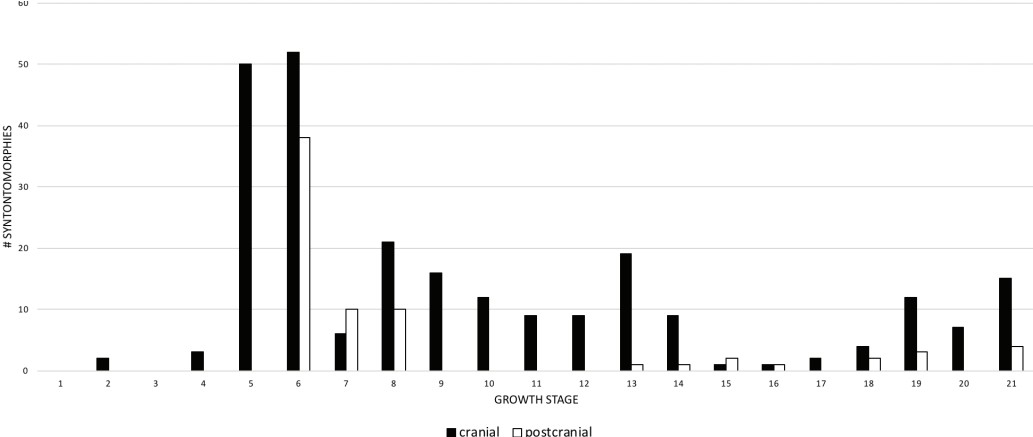

**Figure 6 Comparison of the frequency distributions of cranial and postcranial changes in the growth series of *Tyrannosaurus rex*.** The growth stages are along the *x*-axis (corresponding to the numbered nodes of the ontogram in Fig. 2) and the *y*-axis corresponds to the number of synontomorphies. Cranial changes are shown in solid bars; postcranial changes are shown in hollow bars. Cranial and postcranial changes tend to follow the same overall pattern although postcranial changes are exceeded by cranial changes, except at growth stages 7, 15, and 16. The relatively late occurrence of postcranial changes (at growth stage 6) is an artifact of the absence of postcranial material among the least mature specimens in the sample.

nearly continuous and have five peaks, whereas only three are seen postcranially that are often preceded and followed by one or more growth stages of no change. The absence of postcranial changes in growth stage 5 is an artifact of the least mature specimens lacking postcranial material. The abundance of postcranial changes at growth stage 7 is an

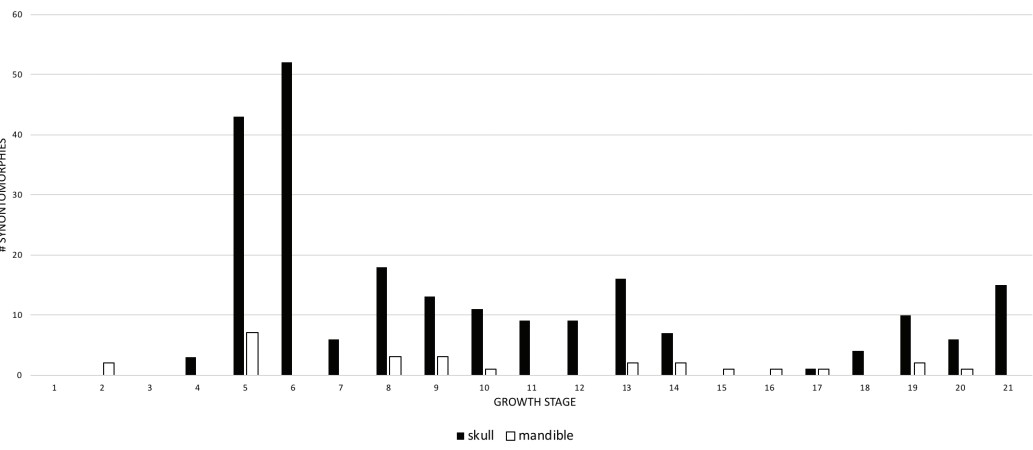

**Figure 7 Comparison of the frequency distribution of synontomorphies of the cranium with that of the mandibular ramus in the ontogeny of *Tyrannosaurus rex*.** Growth stages are along the x-axis (corresponding to the numbered nodes of the ontogram in Fig. 2) and the y-axis corresponds to the number of synontomorphies. Skull changes are shown with solid bars; mandible changes are shown with hollow bars. Although a greater number of changes is seen in the cranium than in the mandibular ramus, the lower jaw completes its early phase of changes (stage 5) before the cranium (stage 6). Thereafter, the pattern of mandibular changes is generally congruent with the cranium.

artifact of the exemplar specimen having an incomplete skull and a relatively complete hindlimb. In contrast, the absence of postcranial change seen from growth stage 9–12 reflects a signal of quiescence given that several specimens in that interval do include postcranial bones.

When the frequencies of changes to the cranium and mandible are compared, both follow the same overall pattern (Fig. 7), but the cranial changes generally outnumber those of the mandible. Mandibular changes occur in a series of three low peaks (i.e., less than 8 synontomorphies per growth stage) throughout ontogeny; the highest number of changes (7) occurs early in the 5th growth stage, but this trend does not continue into the sustained and extremely high number of changes seen in the cranium at the 6th stage. This difference results from the absence of mandibular bones in the sample of subadults at the fifth stage. In contrast, the exemplar of the seventh growth stage, LACM 23845, is represented by an incomplete mandibular ramus, but, notably, it does not result in mandibular synontomorphies at that growth stage; presumably those changes occurred earlier, at growth stage 5. The absence of changes at the 7th growth stage indicates that the completion of the progression of changes in the mandibular ramus, which produces the dorsoventrally deep skull frame, precedes that of the cranium. In adulthood, the number of changes in the cranium exceeds that of the mandible or, on occasion, mandibular changes exceed cranial changes at growth stages where the cephalic skeleton is represented solely by mandibular bones (NHMUK R7994) or very few cranial bones (NMMNH P-3698).

Changes to the skull and jaws were examined by anatomical domain, which includes discrete regions of the skull such as bony structures associated with the antorbital air sac system or aggregates of functional structures, such as joint surfaces. For ease of

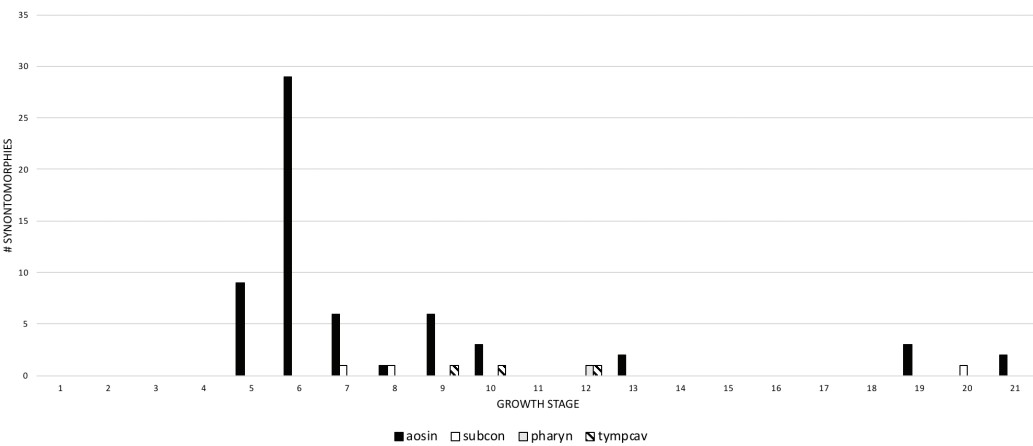

**Figure 8 The frequency distribution of synontomorphies by cephalic pneumatic system in the growth series of *Tyrannosaurus rex*.** Growth stages are along the *x*-axis (corresponding to the numbered nodes of the ontogram in Fig. 2) and the *y*-axis corresponds to the number of synontomorphies. Changes to the antorbital sinus system are dominant over others and are sustained though growth, in contrast to the other systems that occur in adulthood and are transient in occurrence. aosin, antorbital sinus system; pharyn, pharyngeal sinus system; subcon, subcondylar sinus system; tympcav, tympanic cavity.                                                   

comparison, the domains were separated into two groups: pneumatic systems on the one hand, and apneumatic features on the other. The pneumatic systems include the antorbital sinus system (invades the snout), tympanic sinus system (invades the lateral surface of the braincase), median pharyngeal sinus system (invades the sphenoid rostrum and basicranium), and the cervical air sac system (invades the occiput). In descending order, the most frequent pneumatic changes are associated with the antorbital, cervical (=subcondylar) and tympanic sinus systems, and the median pharyngeal sinus system (Fig. 8). Changes associated with the antorbital sinus occur throughout ontogeny, whereas cervical changes begin in adulthood and continue almost to senescence; in contrast, median pharyngeal changes are limited to growth stage 12, at the end of the young adult growth category. Finally, tympanic changes are limited to growth stages 9–12, during young adulthood.

Apneumatic domains include joint surfaces, muscle scars, the subcutaneous surface, neurovasculature, the occiput, dentition, and the skull frame. The skull frame sees the most changes throughout the growth series (Fig. 9). Change to all other domains are low in frequency (i.e., fewer than 10 changes per growth stage), but most of them also change throughout growth, including joint surfaces (growth stage 4–21), the dorsotemporal fossa (growth stage 4–21), neurovasculature (growth stage 2–21), muscle scars (growth stage 5–21), and the subcutaneous surface (growth stage 5–21). A limited pattern is seen in the dentition where changes occur from the 5th to 17th growth stages. In contrast, nonmuscular and apneumatic changes to the occiput are limited to the young adult growth category (growth stage 9).

In the postcranial skeleton, the greatest number of changes (more than five) are seen in the transition from the subadult to the young adult growth categories (Fig. 10) that include, in descending order, the pes, fibula, scapula, coracoid, and humerus. Thereafter,

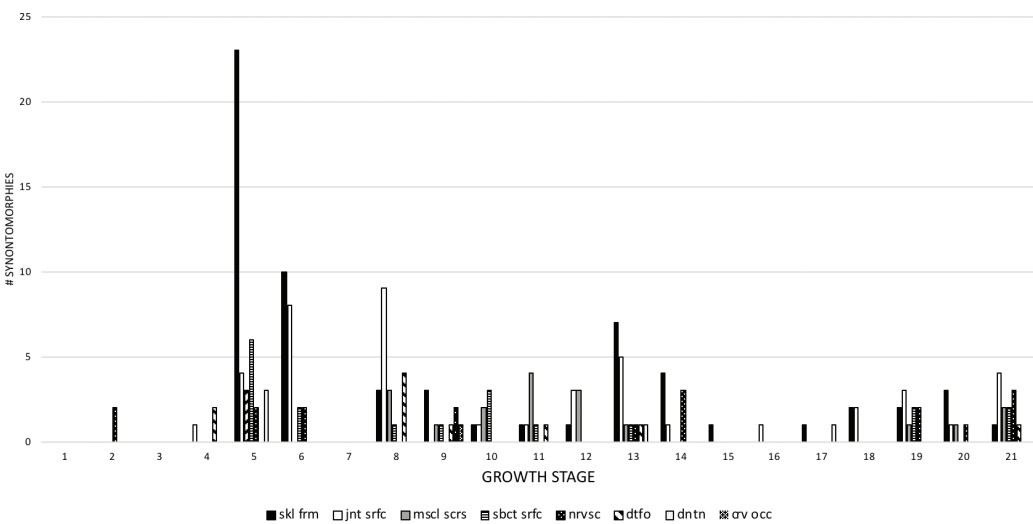

**Figure 9 The frequency distribution of synontomorphies by apneumatic anatomical domain in the growth series of *Tyrannosaurus rex*.** Growth stages are along the *x*-axis (corresponding to the numbered nodes of the ontogram in Fig. 2) and the *y*-axis corresponds to the number of synontomorphies. Changes to the skull frame are dominant over others and all are sustained throughout growth, aside from the dentition and cervical occiput. crv occ, cervical occiput; dntn, dentition; dtfo, dorsotemporal fossa; jnt srfc, joint surfaces; mscl scrs, muscle scars; nrvsc, neurovasculature; sbct srfc, subcutaneous surface; skl frm, skull frame.

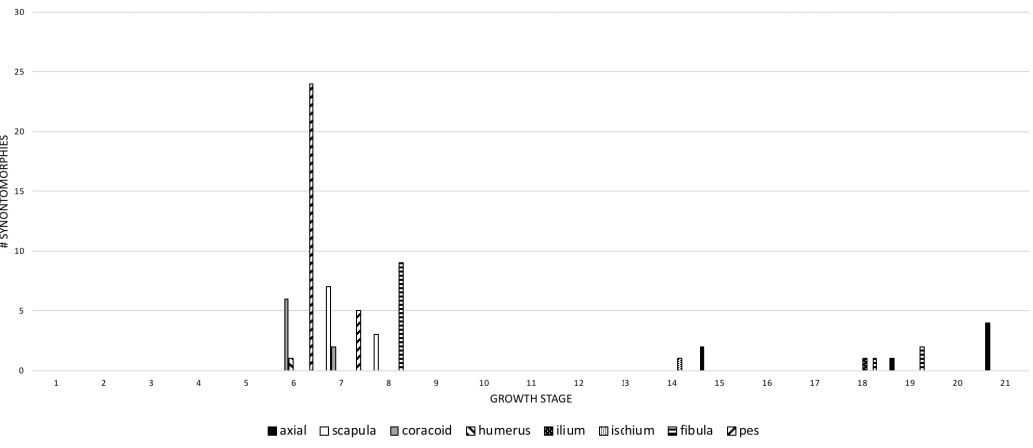

**Figure 10 The frequency distribution of postcranial synontomorphies in the growth series of *Tyrannosaurus rex*.** Growth stages are along the *x*-axis (corresponding to the numbered nodes of the ontogram in Fig. 2) and the *y*-axis corresponds to the number of synontomorphies. Changes to the appendicular skeleton dominate in the transition between juvenile and subadult, whereas changes to the pelvic girdle and axial skeleton occur late in adulthood.

postcranial changes effectively cease until three clusters of changes in adulthood, at the 14th and 15th stages, the 18th and 19th stages, and a final set at the 21st growth stage.

Early changes (juvenile to young adult categories) happen to the pectoral girdle and pes (6th and 7th growth stages), the humerus (6th growth stage), and the fibula (8th growth stage). Later changes (adult to senescent adult categories) are seen in the axial column (15th, 19th, and 21st growth stages), pelvic girdle (14th and 18th growth stages), and the

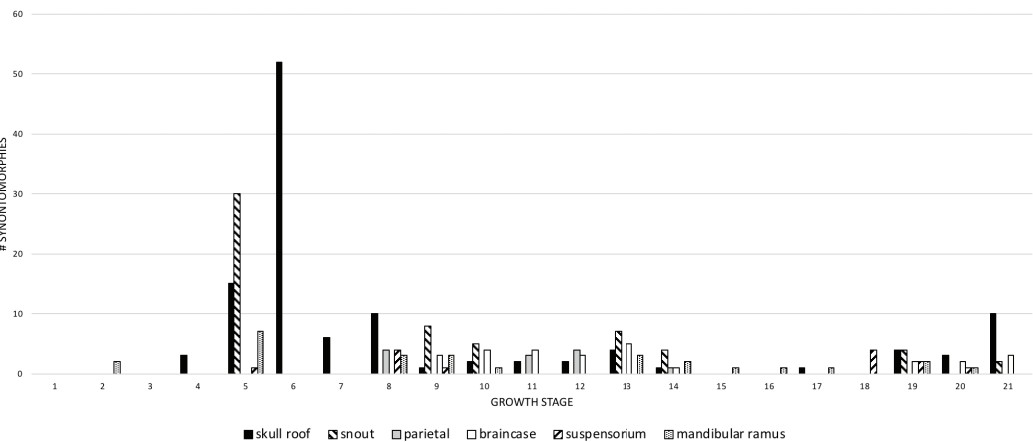

**Figure 11 The frequency distribution of changes to the craniomandibular functional modules (sensu *Werneburg et al., 2019*) in the growth series of *Tyrannosaurus rex*.** Growth stages are along the *x*-axis (corresponding to the numbered nodes of the ontogram in Fig. 2) and the *y*-axis corresponds to the number of synontomorphies. The onset of the changes to the skull roof, snout, mandibular ramus, and suspensorium modules occur early in growth, whereas the onset of changes to the parietal and braincase occur in adulthood. Changes continue throughout growth in all domains, aside from those to the parietal that cease at growth stage 14.

**Table 4 Summary of synontomorphies and individual variation in *Tyrannosaurus rex* organized by growth category and functional modules of the skull and jaws.** Summary of the number of unambiguously optimized changes recovered for the craniomandibular skeleton of *Tyrannosaurus rex*, organized by functional modules (sensu *Werneburg et al., 2019*). Individual variation is shown in parentheses.

| Growth category | Skull roof | Snout + palate | Parietal | Suspensorium | Braincase | Mandible | Total |
|---|---|---|---|---|---|---|---|
| Small juvenile | 3 (3) | 0 (1) | 0 (2) | 0 (0) | 0 (0) | 2 (0) | 5 (6) |
| Large juvenile | 15 (6) | 30 (5) | 0 (0) | 1 (0) | 0 (0) | 7 (3) | 53 (14) |
| Subadult | 58 (2) | 0 (0) | 0 (2) | 0 (0) | 0 (0) | 0 (1) | 58 (5) |
| Young adult | 17 (38) | 13 (23) | 11 (10) | 5 (8) | 14 (8) | 7 (11) | 67 (98) |
| Adult | 14 (51) | 15 (28) | 1 (9) | 7 (15) | 8 (21) | 11 (27) | 56 (151) |
| Senescent adult | 10 (n/a) | 2 (n/a) | 0 (n/a) | 2 (n/a) | 1 (n/a) | 0 (n/a) | 15 (n/a) |
| Total | 117 (100) | 60 (57) | 12 (23) | 13 (23) | 25 (29) | 27 (42) | 254 (274) |

fibula (18th and 19th growth stages). This pattern suggests a relatively instantaneous transition in the pectoral girdle and limb, and the pes; in contrast, changes to the axial column, and to the pelvic girdle and limb, are relatively sustained throughout growth.

### Craniomandibular modules

Recent work has found that the skull of *T. rex* is organized into six modules of functional integration (*Werneburg et al., 2019*). The growth series obtained here shows that the modules experience different amounts of ontogenetic change (Fig. 11; Table 4): in descending order, the dorsum of the snout, circumorbital bones, and frontals; the sides of the snout and palate; the mandibular ramus; the braincase; the suspensorium; and the parietal bone. The modular pattern of the skull is considered evidence for a flexible framework (this hypothesis conflicts with inferences based on Finite Element Analysis (FEA) modeling; *Cost et al., 2019*); in terms of growth, the suspensorium, parietal,

and braincase are the modules that change the least (less than 10 changes per growth stage), which is consistent with their stable keystone-like function against which the relatively flexible palate and facial skeleton can passively move and the lower jaws freely rotate.

The greatest change across modules (more than 10) occurs in the 5th and 6th growth stages; this corresponds to the transition from small to large juveniles (Fig. 11). The transition from the long and low skulls of juveniles to the tall skulls of subadults imposes 52 changes, where the snout dorsum module is changed from the delicate and thin morphotype of juveniles to the greatly expanded and inflated form of subadults.

Changes to the skull roof and mandible are seen throughout ontogeny; changes to the snout and supensorium occur later, from the 5th growth stage onwards. Other modules do not change until young adulthood, including the parietal and braincase; changes to the parietal are limited to young adults and adults, whereas changes to the braincase continue to the last growth stage. These differences in the timing of change to modules is consistent with the hypothesis of modularity, otherwise they would change together if they were integrated. During the interval of greatest change, in the transition from juveniles to subadults (growth stages 5 and 6), the modules that change the most are the skull roof, snout, and mandible.

### *Ontogram mapped onto growth curve*

Mapping the ontogram onto a previously published growth curve for *T. rex* (*Erickson et al., 2004*) permits refinement of the diagnosis of the higher-level growth categories of *Carr (1999)* and *Erickson et al. (2004)*, namely juvenile, subadult, young adult, adult, and senescent adult (Fig. 12). This approach provides a framework for comparison with previous studies on functional morphology in *T. rex* and the evolution of its ontogeny. This template serves as a heuristic device and is not intended to estimate growth rates; for example, the position of BMRP 2006.4.4 on the steepest part of the growth curve conflicts with the histological evidence that the individual was at a moderate growth rate before death (*Woodward et al., 2020*). At best, the curve is an approximation of the true growth rate. The chronological age of BMRP 2002.4.1 is based on the revised age of *Woodward et al. (2020)*; the age of LACM 23845 (14 years; *Erickson et al., 2004*) is considered here to be an underestimate since it conflicts with that of the less mature BMRP 2006.4.4 (15 years; *Woodward et al., 2020*).

## Growth categories

Alignment between the growth series, chronological age, cortical histology, number of growth changes, growth rate, and, to a lesser degree, size and mass, were used to define five growth categories, once the ontogram was mapped onto the growth curve of *Erickson et al. (2004)*, a procedure that was constrained by seven histologically aged specimens (Fig. 12). Although mass is a relevant variable in these comparisons, given the variety of methods used by different workers, the published mass estimates conflict with each other (Table 5); for that reason, the decision for each mention of mass as a relevant diagnostic character is given below. Also, although the masses are given to the nearest
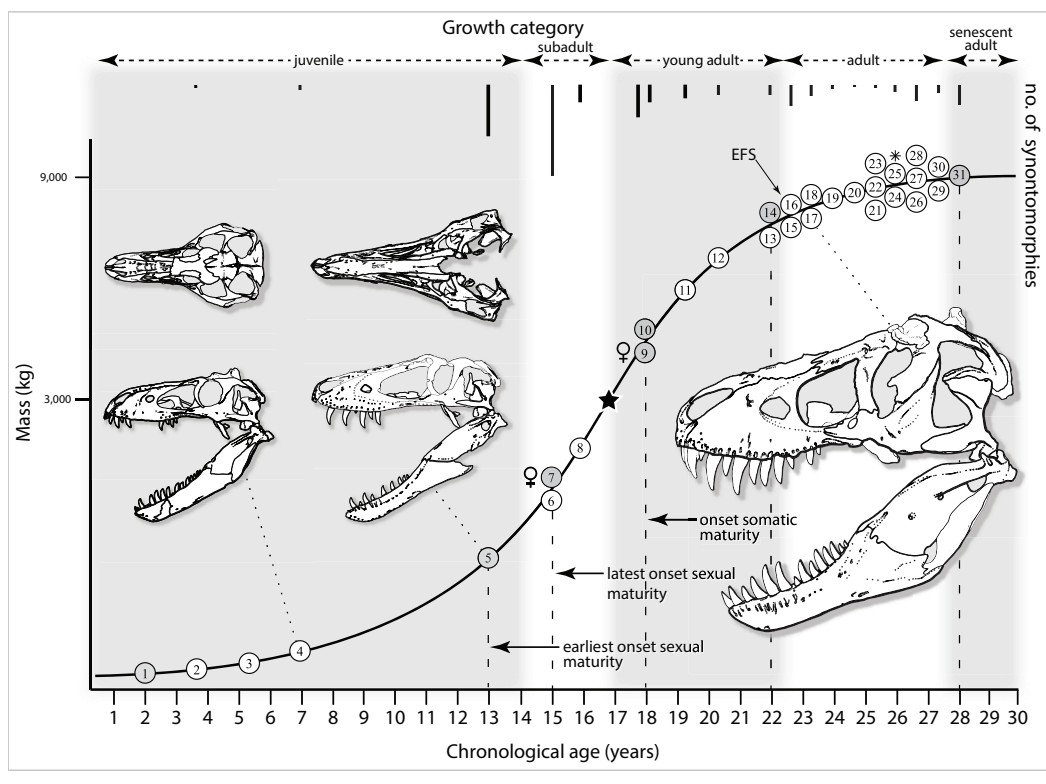

**Figure 12** *Tyrannosaurus rex* ontogram, chronological age, and mass mapped onto the growth curve of *Erickson et al. (2004)*. Ranges of growth categories are indicated across the top. Each circle represents an individual specimen; the vertical columns of circles indicate multiple specimens in a single growth stage; the horizontal position of the white circles does not imply a specific chronological age. For ease of interpretation, and to accommodate missing mass estimates, the position of the circles do not correspond with the scale of the *y*-axis. The gray circles indicate histologically aged specimens that are connected to the *x*-axis by dashed lines for ease of interpretation. The star represents the ~3,000 kg threshold that separates *T. rex* from its closest, but smaller, relatives. Alternating gray and white bars in the background block out the growth categories and their blurred edges reflect the imprecision of their boundaries. The height of each black vertical bar corresponds to the number of synontomorphies in each growth stage, which ranges from 1 to 90 characters. Asterisk indicates the type specimen of *T. rex*. Female symbols indicate BMRP 2006.6.4 and MOR 1125, the only unambiguous female specimens in the data set. Skull illustrations are to scale, with that of the adult set to 1.3 m. From left to right the skulls are: CMNH 7541, BMRP 2002.4.1, and AMNH FARB 5027. The large juvenile BMRP 2002.4.1 is in dorsal view to show the early occurrence of the narrow snout and wide temporal region that characterizes *T. rex* to the exclusion of other tyrannosaurids, which is ontogenetically incongruent with its late-occurring (i.e., autapomorphic) phylogenetic homolog; see text for discussion. Major life history events are indicated, including the onset of sexual maturity and somatic maturity, and the earliest occurrence of histologic adulthood. Suspensorium in CMNH 7541 is reconstructed after BMRP 2002.4.1. EFS, External Fundamental System. Specimens: 1, LACM 28471; 2, AMNH FARB 5050; 3, DDM 344.1; 4, CMNH 7541; 5, BMRP 2002.4.1; 6, RSM 2990.1; 7, BMRP 2006.4.4; 8, LACM 23845; 9, MOR 1125; 10, TMP 1981.006.0001; 11, LACM 150167; 12, AMNH FARB 5117; 13, UWBM 99000; 14, TMP 1981.012.0001; 15, RSM 2523.8; 16, SDSM 12047; 17, AMNH FARB 5027; 18, AMNH FARB 5029; 19, NHMUK R7994; 20, NMMNH P-3698; 21, MOR 1131; 22, MOR 980; 23, MOR 555; 24, LACM 23844; 25, CM 9380; 26, UCMP 118742; 27, MOR 008; 28, UMNH 11000; 29, MOR 2822; 30, UWGM 181; 31, FMNH PR2081.

kilogram or, in four instances, to the tenth of a kilogram, it is not intended to give the impression of precision, rather the exact estimates are here simply reproduced unaltered from the original sources. To reflect the natural imprecision, mass estimates given in

**Table 5 Summary of mass estimates for *Tyrannosaurus rex*.** Summary of published mass estimates (kg) for individual specimens of *Tyrannosaurus rex* with rank order in each column in parentheses; sum rank order given in right-hand column. The rank of AMNH FARB 5027 could not be identified given its presence in only one analysis and the great difference in estimates between the analyses. Column numbers: 1, *Erickson et al. (2004)*; 2, *Henderson & Snively (2004)*; 3, *Bates et al. (2009)*; 4, *Hutchinson et al. (2011)*; 5, *Campione et al. (2014)*; 6, *Snively et al. (2019)*; 7, *Persons, Currie & Erickson (2019)*.

| Specimen | 1 | 2 | 3 | 4 | 5 | 6 | 7 | Rank |
|---|---|---|---|---|---|---|---|---|
| LACM 28471 | 29.9 (1) | | | | | | | 1 |
| BMRP 2002.4.1 | | | | 954 (1) | | 660.2 (1) | | 2 |
| LACM 23845 | 1,807 (2) | | | | | | | 3 |
| TMP 1981.006.0001 | 3,230 (3) | | | | | | 4,469 (1) | 4 |
| TMP 1981.012.0001 | 5,040 (4) | | | | | | | 5 |
| MOR 980 | | | | | | | 5,112 (2) | 6 |
| MOR 1125 | | | | | | | 6,100 (3) | 7 |
| AMNH FARB 5027 | | | | | | 6,986.6 (2) | | ? |
| MOR 555 | | | 6,071.8 (1) | 8,272 (2) | 6,216 (1) | | 6,264 (4) | 8 |
| CM 9380 | | | | 9,081 (3) | 6,688 (2) | | 6,740 (5) | 9 |
| FMNH PR2081 | 5,654 (5) | 10,200 (1) | | 13,996 (4) | 7,377 (3) | 9,130.9 (3) | 8,462 (6) | 10 |
| RSM 2523.8 | | | | | 8,004 (4) | | 8,870 (7) | 11 |

the next section are rounded up in the discussion of diagnostic features of growth categories.

## Juveniles

Juveniles (=adolescent category of *Erickson et al., 2004*) have a skull length less than 80 cm, long and low skulls with a length to height ratio of 3.0 or greater, are equal to or less than 13 years old, and lack an EFS. This growth category sees 58 unambiguously optimized growth changes, of which 24 have a CI of 1.0 (Table 6). In the skull, most changes are seen in the snout module (sensu *Werneburg et al., 2019*). This category corresponds to growth stages 1–5 of the ontogram (Fig. 2) and to the initial lag phase of the *T. rex* growth curve of *Erickson et al. (2004*; Fig. 12). The juvenile stage extends from hatching to the 13th year of life, before the long and low skull proportions are lost at the subadult category.

A single mass estimate of 39.9 kg has been published for the small juvenile LACM 28471 by *Erickson et al. (2004)*. Two mass estimates have been published for the large juvenile BMRP 2002.4.1: a range of 639–1,269 kg, and a mean of 954 kg was published by *Hutchinson et al. (2011)* and an estimate of 660.2 kg was published by *Snively et al. (2019)*. The maximum estimate is used here as a point of reference; therefore, juveniles are here considered to have a mass equal to or less than ~1,300 kg.

The juvenile growth stage is diagnosed by the possession of a dorsoventrally shallow skull and jaws, the increase in height of which defines the subadult stage. To precisely reflect the primary changes that occur even during this initial phase of life, this growth category is divided into two categories in the following diagnosis, namely small juveniles and large juveniles. A total of five growth changes occur in the small juvenile growth stage, which affect the skull roof and mandible modules of *Werneburg et al. (2019)*.

**Table 6 Diagnostic characters of the juvenile growth category of *Tyrannosaurus rex*.** These characters are unambiguously optimized on the topology and have a consistency index (CI) of 1.0 indicating that they have changed only once on the ontogram and so can be considered diagnostic of the growth category. PHYLO, corresponding homologous phylogenetic character (see Data S1).

| Character | Growth change |
|---|---|
| 4. Orbital fenestra, shape (PHYLO 7) | 0 ==> 1 |
| 27. Maxilla, maxillary flange, size (PHYLO 47) | 0 ==> 1 |
| 57. Jugal, postorbital process, orientation (PHYLO 101) | 0 ==> 1 |
| 66. Postorbital, jugal, ramus, subocular process, presence (PHYLO 116) | 0 ==> 1 |
| 89. Frontal, postorbital buttress, distinctiveness (PHYLO 159) | 0 ==> 1 |
| 123. Dentary, Meckelian groove, depth (PHYLO 236) | 0 ==> 1 |
| 215. Bony naris, dorsal margin | 0 ==> 1 |
| 226. Internal antorbital fenestra, jugal contribution | 0 ==> 1 |
| 238. Postorbital bar, jugopostorbital suture | 0 ==> 1 |
| 239. Postorbital bar, orientation | 0 ==> 1 |
| 240. Postorbital bar, plane change | 0 ==> 1 |
| 254. Premaxilla, body, rostral margin, angulation position | 0 ==> 1 |
| 293. Nasal, cross-section of rostral third, shape | 0 ==> 1 |
| 296. Nasal, midline bumps, number | 1 ==> 2 |
| 299. Nasal, dorsal foramina, number rows | 1 ==> 2 |
| 315. Nasal, frontal ramus, dorsum, cross section | 0 ==> 1 |
| 358. Maxilla, subcutaneous surface, orientation to ventral margin | 0 ==> 1 |
| 415. Maxilla, palatal process, extent of bony choana | 0 ==> 1 |
| 556. Jugal, body, degree of texture | 0 ==> 1 |
| 621. Postorbital, body, medial rugosities | 0 ==> 1 |
| 1069. Dentary, bone, shallowest point | 0 ==> 1 |
| 1170. Prearticular, caudal ramus, dorsal & ventral margins, orientation | 0 ==> 1 |
| 1173. Prearticular, rostral ramus, form | 0 ==> 1 |
| 1176. Dentition, frst maxllary tooth, form | 0 ==> 1 |

In contrast, 53 changes occur in the large juvenile growth stage that are more widespread across the skull, including modules of the skull roof, snout, suspensorium, and mandible. The large juvenile category marks the earliest onset of sexual maturity (see below).

## Small juveniles
### Joint surfaces
In the fourth growth stage, there is a clear distinction between the postorbital buttress and the caudal shelf of the frontal bone.

### Neurovasculature
In the second growth stage, the alveolar row of foramina that penetrates the lateral surface of the dentary acquires a sulcus that unites all of the openings. In medial view, the Meckelian sulcus becomes a deeply inset groove.

### Dorsotemporal fossa

In the fourth growth stage, the sagittal crest of the frontal becomes tall and the fossa extends rostrodorsally at a steep angle relative to the level forehead (subcutaneous surface ahead of the fossa).

## Large juveniles
### Skull frame

In the fifth growth stage, several correlates of an increase in skull height are seen, including the dorsal position of the angulation in the rostral margin of the premaxilla, the rostrodorsal orientation of the nasal process of the premaxilla, an oval orbital fenestra, vertical orientation of the postorbital bar with the presence of a depression and strut on the lateral surface of the postorbital process of the jugal, the inset and braced ventral end of the jugal process of the postorbital, and presence of the subocular process of the postorbital.

Some changes in skull architecture almost certainly reflect a response to high bite forces (*Henderson, 2002*; *Bates & Falkingham, 2012*, *2018*; *Gignac & Erickson, 2017*; *Cost et al., 2019*) and their corresponding high loads upon the skull and teeth. These include a convex cross section of the rostral third of the nasal and its frontal ramus, the tall height of the antorbital fossa below the internal antorbital fenestra, the straight rostral margin of the rostroventral ala of the lacrimal, and the concave dorsal margin of the vomer.

Other changes are almost certainly unrelated to the increase in skull height, including the straight dorsal margin of the bony naris, the presence of a contribution of the jugal to the margin of the internal antorbital fenestra, enhancement of the plane change between the temporal region and the orbitosnout region, the medioventral orientation of the lateral surface of the maxilla to the alveolar margin of the bone, the intermediate position of the bony choana, the rostralward shift of the ventral margin of the joint surface for the quadratojugal ramus on the jugal, the straight shape of the dentary in ventral view, and caudalward position of the shallowest point of the dentary.

### Joint surfaces

In the fifth growth stage an alteration for a stable skull frame is seen, namely a distinct maxillary flange that produces a deep slot for the nasal. The plane change between snout and orbitotemporal region is reflected in the sinuous jugopostorbital suture.

### Subcutaneous surface

In the fifth growth stage, an increase in the relief of the subcutaneous surface of the face is seen, where alveolar skirts are present at the premaxilla, the body of the jugal is rugose, indicating the appearance of armor-like skin (*Carr et al., 2017*), and subcutaneous rugosities extend onto the medial surface of the postorbital.

### Cephalic ornamentation

In the fifth growth stage, some changes in cephalic ornamentation are independent of the texture of the subcutaneous surface, including the decrease in size of the cornual process of the jugal and the increase in the number of midline bumps on the nasals.

### Paranasal pneumaticity

In the fifth growth stage, internal inflation of the maxillary sinus is seen externally, shown by the convex rostrodorsal surface of the bone. Some changes to the antorbital fossa might correspond to internal changes of the dentition, including elimination of the rostral end of the crease that defines the ventral margin of the external antorbital fenestra.

In the maxilla, the rostroventral corner of the external antorbital fossa is indistinct, but it is difficult to determine whether this is caused by a difference in pneumaticity or of the subcutaneous surface; the antorbital fossa is deeply inset relative to the lateral surface of the bone, especially rostrodorsally; and the rostral pneumatic recess of the palatine becomes longer than the caudal recess.

### Neurovasculature

In the fifth growth stage, the number of foramina rows in the nasal increases to three and the rostrodorsal foramen of the tract of the subnarial foramen makes its appearance.

### Adductor musculature

In the fifth growth stage, the insertion for the ventral pterygoid muscle on the surangular and dentary is enhanced, the dorsal margin of the postorbital is everted medially, and in rostral and medial views its dorsal margin is convex.

Several bony changes almost certainly reflect an increase in the forcefulness of the adductor musculature, including rostrally converging dorsal and ventral margins of the caudal ramus of the prearticular. Changes in the adductor musculature impose alterations to bones that are not directly related to increased bite force; for example, the rostral ramus of the prearticular is paddle-shaped, which in life medially increased the bony enclosure around the inserted musculature.

### Joint capsules

In the fifth growth stage, the horizontal ridge on the ventral quadrate process of the quadratojugal is absent, and the lateral scar below the glenoid of the surangular becomes rugose.

### Dentition

In the fifth growth stage, the maxillary tooth count increases to 16 and the dentary tooth count increases to 17; and the first maxillary tooth becomes subincisiform. The teeth and antorbital fossa complex are useful points of reference for capturing changes in tooth size. For example, in the fifth growth stage less than six teeth are ahead of the external antorbital fossa, which almost certainly reflects enlargement of the teeth if, and only if, the rostral margin of the fenestra is a stable landmark throughout ontogeny. Geometric morphometric quantification of skull shape in *T. bataar* shows that the rostral end of the fossa shifts caudally during growth (*Foth, Hendrick & Ezcurra, 2016*). If the same trend occurs in *T. rex*, and if there was no increase in the size of teeth then an increase in the number of teeth ahead of the fossa would be seen. Therefore, tooth size increases despite the caudalward shift of the fossa.

## Subadults

Subadults (= juvenile category of *Erickson et al., 2004*) have a skull length from 80 cm to 1.1 m, tall skulls with a length to height ratio less than 3.0 (deduced from the straight form of the ventral ramus of the lacrimal in LACM 23845 and RSM 2990.1, in contrast to the strongly curved form seen in juveniles such as CMNH 7541 and BMRP 2002.4.1), are from 15 to 17 years old, and lack an EFS. The maximum number of changes in the entire growth series, 106, are seen in this growth category; 69 of the synontomorphies have a CI of 1.0 (Table 7). In the skull the changes are limited to the skull roof module. The maximum number of postcranial changes, 48, occur in this growth stage and affect the limb girdles and appendages. This category corresponds to growth stages 6 and 7 of the ontogram (Fig. 2) and to the first half of the exponential phase of the *T. rex* growth curve (*Erickson et al., 2004*; Fig. 12). The presence of medullary bone in BMRP 2006.4.4 (*Woodward et al., 2020*) marks the latest onset of sexual maturity (Fig. 12).

A mass estimate of 1,807 kg was published for LACM 23845 by *Erickson et al. (2004)*; no other estimates have been published for this specimen. Therefore, a range of mass greater than ~1,300 kg and equal to or less than ~1,810 kg (the mass seen in young adults) is considered here as diagnostic of subadults.

The transition from the long and low skull and narrow teeth of juveniles to the tall and powerful skull and thick teeth of subadults occurred shortly before the halfway point of life (assuming a lifespan of at least 28 years; *Erickson et al., 2004*), within a narrow two-year interval between the growth categories, at the start of the exponential phase of growth (sensu *Erickson et al., 2004*). This rapid change to the skull frame and dentition marks the onset of the increase in bite force; an order-of-magnitude increase in bite force between juvenile and adult *T. rex* was found by *Bates & Falkingham (2012)*. This increase is almost certainly congruent with the niche partitioning that is hypothesized to ecologically separate juveniles from adults (*Snively, Henderson & Phillips, 2006*). Therefore, this extreme transition in skull shape and function occurs before the onset of somatic maturity (i.e., adult size; sensu *Erickson et al., 2004*) (Fig. 12). Also, the presence of medullary bone in BMRP 2006.4.4 shows that *T. rex* is typically reptilian given that sexual maturity precedes somatic maturity (*Erickson et al., 2007*; *Lee & Werning, 2008*).

### Skull architecture

In the sixth growth stage, several changes to the lacrimal reflect an increase in bite force, including the 7-shaped bone, a rostrocaudally long ventral ramus, a dorsoventrally deep rostrodorsal process, a long rostroventral process that separates the nasal from the maxilla, a sliver-like maxillary process, the medially extended and convex medial margin of its dorsal ramus, the lengthened region caudal to the lacrimal pneumatic recess, and abruptly rostroventrally extending rostrodorsal margin of the rostral ramus.

Other architectural changes are seen in the lacrimal that are less obviously the result of high loads: the distal (=ventral) end of the orbitonasal ridge is mediolaterally wide but tapers ventrally; the ridge is not backswept; the orbitonasal ridge grades into the ventral ramus ahead of it; the caudal edge of the ridge is positioned far ahead of the caudal margin of the ventral ramus; the ridge extends abruptly rostroventrally dorsally, above the

Table 7 Diagnostic characters of the subadult growth category of *Tyrannosaurus rex*. These characters are unambiguously optimized on the topology and have a consistency index (CI) of 1.0 indicating that they have changed only once on the ontogram and so can be considered diagnostic of the growth category. PHYLO, corresponding homologous phylogenetic character (see Data S1).

| Character | Growth change |
| --- | --- |
| 37. Lacrimal, bone, shape (PHYLO 65) | 0 ==> 1 |
| 38. Lacrimal, cornual process, apex, shape (PHYLO 66) | 1 ==> 2 |
| 40. Lacrimal, height above lacrimal pneumatic recess (PHYLO 69) | 0 ==> 1 |
| 41. Lacrimal, rostral ramus, inflation (PHYLO 75) | 0 ==> 1 |
| 41. Lacrimal, rostral ramus, inflation (PHYLO 75) | 1 ==> 2 |
| 43. Lacrimal, number of recesses ahead of lacrimal pneumatic recess (PHYLO 78) | 0 ==> 1 |
| 44. Lacrimal, medial pneumatic recess, presence (PHYLO 79) | 0 ==> 1 |
| 45. Lacrimal, supraorbital ramus, length (PHYLO 82) | 0 ==> 1 |
| 46. Lacrimal, supraorbital process, inflation (PHYLO 82) | 0 ==> 1 |
| 219. Snout, dorsum, width caudal end | 0 ==> 1 |
| 433. Lacrimal, dorsal ramus, medial edge, form | 0 ==> 1 |
| 434. Lacrimal, medial fossa, rostral ridge, presence | 0 ==> 1 |
| 436. Lacrimal, dorsal ramus, conchal surface, dorsal ridge, depth of caudal part | 0 ==> 1 |
| 447. Lacrimal, dorsal ramus, region dorsal & rostrodorsal to lacrimal recess, form | 0 ==> 1 |
| 454. Lacrimal, cornual process, lateral extent | 0 ==> 1 |
| 455. Lacrimal, lacrimal pneumatic recess, foramen in ventral margin, presence | 0 ==> 1 |
| 461. Lacrimal, region caudal to lacrimal pneumatic recess, length & depth | 0 ==> 1 |
| 462. Lacrimal, length lacrimal pneumatic recess relative to region behind it | 0 ==> 1 |
| 465. Lacrimal, rostral ramus, subcutaneous surface height | 0 ==> 1 |
| 466. Lacrimal, junction antorbital fossa & subcutaneous surface ahead of recess | 0 ==> 1 |
| 468. Lacrimal, region between lacrimal pneumatic recess septum & distal recess | 0 ==> 1 |
| 470. Lacrimal, pneumatic recess, proximity to maxilla | 0 ==> 1 |
| 474. Lacrimal, rostral ramus, medial joint surfaces, height | 0 ==> 1 |
| 476. Lacrimal, medial tab, dorsal margin | 0 ==> 1 |
| 481. Lacrimal, maxillary process, height | 0 ==> 1 |
| 482. Lacrimal, caudolateral shelf, form | 1 ==> 2 |
| 489. Lacrimal, frontal process, groove on dorsal surface, presence | 0 ==> 1 |
| 517. Lacrimal, ventral ramus, proximal to distal ends | 0 ==> 1 |
| 524. Lacrimal, ventral ramus, subcutaneous surface, caudal extension | 0 ==> 1 |
| 1546. Scapulocoracoid, acromial region, lateral surface, form | 0 ==> 1 |
| 1548. Scapula, glenoid, width joint surface relative to shaft | 0 ==> 1 |
| 1549. Scapula, glenoid, anterolateral corner, form | 0 ==> 1 |
| 1550. Scapula, lip of glenoid, emergence form ventral edge, form | 0 ==> 1 |
| 1551. Scapula, acromion, medial surface, form | 0 ==> 1 |
| 1561. Scapula, muscle scar along dorsal edge, distinctiveness | 0 ==> 1 |
| 1568. Scapula, shaft, ventral surface adjacent to glenoid, orientation | 0 ==> 1 |
| 1574. Coracoid, glenoid, orientation | 0 ==> 1 |
| 1576. Coracoid, glenoid, separation from coracoid process, distinctiveness | 0 ==> 1 |
| 1577. Coracoid, low ridge that extends anteriorly form biceps tubercle, presence | 0 ==> 1 |

(Continued)

| Character | Growth change |
|---|---|
| 1581. Coracoid, ventral half of bone, form | 0 ==> 1 |
| 1583. Coracoid process, length, depth & curvature | 0 ==> 1 |
| 1584. Coracoid, medial fossa between coracoid foramen & anterior edge of bone | 0 ==> 1 |
| 1707. D II, PH 2, proximal joint surface, ventral margin, form | 0 ==> 1 |
| 1711. D II, PH 2, lateral distal condyle, orientation, lateral view | 0 ==> 1 |
| 1712. D II, PH 2, medial distal condyle, elevation | 0 ==> 1 |
| 1714. D II, PH 2, proximodorsal flange, width | 0 ==> 1 |
| 1715. D II, PH 2, medial distal condyle, medial margin, orientation | 0 ==> 1 |
| 1719. D II, PH 2, medial distal condyle, ventral extent | 0 ==> 1 |
| 1748. D III, PH 1, proximal joint surface, form | 0 ==> 1 |
| 1754. D III, PH 1, lateral caudolateral ligament pit, frm | 0 ==> 1 |
| 1779. D IV, PH 1, supracondylar pit, form | 0 ==> 1 |
| 1782. D IV, PH 1, ventromedial condyle, angle relative to shaft | 0 ==> 1 |
| 1784. D IV, PH 1, proximal joint surface, ventral notch, mediolateral position | 0 ==> 1 |
| 1787. D IV, PH 1, distal joint surface, proportions | 0 ==> 1 |
| 1790. D IV, PH 1, scar on shaft ahead of proximal end, prominence | 0 ==> 1 |
| 1792. D IV, PH 1, dorsolateral surface, form | 0 ==> 1 |
| 1793. D IV, PH 1, shaft, ridge lateral to supracondylar pit, form | 0 ==> 1 |
| 1806. D IV, PH 3, form | 0 ==> 1 |
| 1809. D IV, PH 3, ratio of mediolateral width to dorsoventral height | 0 ==> 1 |
| 1810. D IV, PH 3, proximal joint surface, medial margin, form | 0 ==> 1 |
| 1812. D IV, PH 3, medial collateral ligament point, depth | 0 ==> 1 |
| 1814. D IV, PH 3, distal condyles, divergence | 0 ==> 1 |
| 1815. D IV, PH 3, flexor muscle scar, form | 0 ==> 1 |
| 1817. D IV PH 4, dorsum, form | 0 ==> 1 |
| 1818. D IV PH 4, dorsum, mediolateral orientation | 0 ==> 1 |
| 1820. D IV PH 4, distal joint surface, form | 0 ==> 1 |
| 1824. D IV PH 4, distal joint surface, dorsal margin, indentation, presence | 0 ==> 1 |
| 1826. D IV PH 4, distal condylar region, posterior margin, undercut | 0 ==> 1 |
| 1848. Growth rings, number | 2 ==> 3 |

rostroventral ala; the rostrodorsal margin of the ventral ramus is straight or concave; and the rostroventral margin of the ventral ramus is straight. In the seventh growth stage, the rostroventral margin of ventral ramus of the lacrimal is concave.

### Joint surfaces

In the sixth growth stage, several changes to the lacrimal are seen, including the joint surface for the prefrontal on the dorsal ramus of the lacrimal is a dorsoventrally deep and laterally shallow groove that caudally twists to face mediodorsally, the joint surface for the nasal on the rostral ramus covers its ventral third, the dorsal half of the medial joint surfaces is dorsoventrally shallow, the medial joint surface (behind the medial tube) is

concave in vertical section, a groove on the dorsal surface of the frontal process is present, and a medially extending process behind the joint surface for the prefrontal is present.

### Subcutaneous surface

In the sixth growth stage, the subcutaneous surface of the lacrimal wraps onto the caudal surface of the ventral ramus and the coarse patch caudodorsal to the lacrimal pneumatic recess is absent.

### Cephalic ornamentation

In the sixth growth stage, the low ridge caudodorsal to the lacrimal pneumatic recess is lost.

### Paranasal pneumaticity

In the sixth growth stage, the invasive antorbital air sac produces the medial pneumatic recess of the lacrimal. Some pneumatic changes are almost certainly an epiphenomenon of the overall dorsoventral and mediolateral expansion of the skull, including the distal pneumatic recess of the lacrimal that is positioned close to maxilla. Some pneumatic changes more directly reflect the resorptive tendency of pneumatic epithelia, including the absorption of the caudoventral margin of the medial pneumatic recess.

Inflation of the dorsal ramus of the lacrimal has several simultaneous effects upon the bone, including obscuring the apex of the cornual process and the process in general; increase in the height of the cornual process; loss of the ridge that rostrally bounds the conchal surface; increase of the depth of the ridge above the conchal surface; the convex caudodorsal region of the shallow conchal surface; the convex lateral surface around the lacrimal pneumatic recess; the laterally bulging region of the cornual process of the lacrimal; increase in the height of the subcutaneous surface above the antorbital fossa to twice the height of the rostral ramus of the lacrimal; reduction of the size of the lacrimal pneumatic recess to less than half the length of the region behind it; inflation of the region below the lacrimal pneumatic recess, and the septum ahead of it, which merges the junction of the antorbital fossa and subcutaneous surface ahead of the lacrimal pneumatic recess; the shallowly concave region between the distal recess and the septum ahead of the lacrimal pneumatic recess; the lengthened septum between the accessory recesses; the shallowly concave distal pneumatic recess of the lacrimal; the widened supraorbital process and reduction of its length and increase in its height; and the caudal end of the snout is widened by the inflation of the lacrimals.

In the seventh growth stage, inflation has obliterated the cornual process of the lacrimal. A single accessory recess, the distal recess, is present ahead of the lacrimal pneumatic recess, which is reduced to a foramen.

### Neurovasculature

In the sixth growth stage, the foramen in the ridge that bounds the lacrimal pneumatic recess is absent, and the dorsal margin of the medial tube of the lacrimal extends rostrodorsally.

### Appendicular skeleton

In the seventh growth stage several changes are seen in the pectoral girdle and limb. In the scapula, the acromial region is most deeply concave on the scapula, the glenoid fossa has an intermediate position between lateral and ventral, the glenoid fossa is narrower than the shaft, the anterolateral corner of glenoid widens toward the coracoid, the medial surface of the acromion is convex, the muscle scar along the dorsal edge of the shaft is indistinct, and the ridge on lateral surface of the shaft is absent. The coracoid also shows changes, including a posteroventral orientation of the glenoid, an abrupt separation of the glenoid from the coracoid process, loss of the ridge that extends anteriorly from the biceps tubercle, the ventral half of the bone is convex, and the medial fossa ahead of the coracoid foramen stops short of the anterior edge of the bone. Finally, the deltoid scar at the distal end of the deltopectoral crest of the humerus is deeply inset.

In the seventh growth stage, the lip of the glenoid of the scapula extends from the shaft at an abrupt angle, the shaft is not distinctly narrow between the acromial region and the shaft, and the ventral surface of the shaft next to the glenoid faces ventrally. Changes in the coracoid include the anterolateral orientation of the coracoid foramen and the stout form of the coracoid process.

The changes to the pelvic limb happen in the pes. In the sixth growth stage, the ventral margin of the proximal joint surface of D II PH 2 is concave, the anterior end of the lateral distal condyle is flattened, the medial distal condyle extends anterodorsally, the proximodorsal flange is wide, the medial distal condyle extends anterolaterally, and it extends posteriorly below the level of the posterior margin of the medial collateral ligament pit. In D III PH 1, the proximal joint surface is dissected by sulci and the lateral collateral ligament pit is lens-shaped. In D IV PH 1, the ventromedial condyle extends at a steep angle relative to the shaft, the ventral notch of the proximal joint surface is positioned at the midline, the distal joint surface is much wider than tall, the scar on the lateral surface of the shaft ahead of the proximal end is distinct, the dorsolateral surface of the shaft is convex, and the ridge lateral to the supracondylar pit is short. In D IV PH 3, the phalanx is short, the proximal joint surface is distinctly wider than tall, the lateral collateral ligament pit is deep, and the flexor muscle scar is coarse. In D IV PH 4, the dorsum of the bone is wide, the mediolateral orientation of the dorsum is distinct, the proximal end of the bone is wider than the distal end, the distal joint surface is distinctly concave, the entire posterior margin of the distal condylar region is undercut by a groove that separates it from the ventral surface of the shaft, and the indentation along the dorsal margin of the distal joint surface is wide.

In the seventh growth stage, changes are seen in D IV PH 1, where the supracondylar pit is a deep crease that sharply elevates the distal condyles. In D IV PH 3, supracondylar pit is deep, the medial margin of the proximal joint surface is distinctly convex, the medial collateral ligament pit is large, and the distal condyles shallowly diverge from each other.

### Young adults

Young adults (=subadult of *Erickson et al., 2004*) have a skull length from ~1.16 to 1.4 m, tall skulls with a length to height ratio of 2.6–2.3, are from 18 to 22 years old, lack an EFS,

**Table 8 Diagnostic characters of the young adult growth category of *Tyrannosaurus rex*.** These characters are unambiguously optimized on the topology and have a consistency index (CI) of 1.0 indicating that they have changed only once on the ontogram and so can be considered diagnostic of the growth category. PHYLO, corresponding homologous phylogenetic character (see Data S1).

| Character | Growth change |
| --- | --- |
| 18. Maxilla, interfenestral strut, length (PHYLO 31) | 0 ==> 1 |
| 24. Maxilla, alveolar process, subcutaneous surface, fossae (PHYLO 44) | 0 ==> 1 |
| 109. Otoccipital, paroccipital process, caudal surface, form (PHYLO 200) | 0 ==> 1 |
| 253. Premaxilla, lateral margin, form | 0 ==> 1 |
| 376. Maxilla, antorbital fossa, mediolateral depth | 0 ==> 1 |
| 523. Lacrimal, subcutaneous surface, texture | 0 ==> 1 |
| 651. Postorbital, jugal ramus, width, caudal | 0 ==> 1 |
| 717. Squamosal, dorsotemporal fossa, lateral half, texture | 0 ==> 1 |
| 737. Quadratojugal, squamosal process, ridge, width | 0 ==> 1 |
| 741. Quadratojugal, dorsal joint surface for quadratojugal, texture & depth | 0 ==> 1 |
| 742. Quadratojugal, dorsal joint surface for quadrate, rostral margin, form | 0 ==> 1 |
| 882. Frontals, proportions apposed bones | 0 ==> 1 |
| 913. Frontal, joint surface for the prefrontal, rostrocaudal position | 0 ==> 1 |
| 931. Frontal, orbital surface, medial margin, orientation | 1 ==> 2 |
| 1014. Otoccipital, metotic strut, oval scar complex, participation | 0 ==> 1 |
| 1017. Subcondylar recesses, distance from each other | 0 ==> 1 |
| 1043. Basisphenoid, oval scar, texture | 1 ==> 2 |
| 1048. Basisphenoid, spout, presence | 0 ==> 1 |
| 1052. Basisphenoid, flange form ventral end of preotic pendant | 0 ==> 1 |
| 1055. Supraoccipital, bar across lateral processes, prominence | 0 ==> 1 |
| 1082. Dentary, chin, subcutaneous surface, texture | 0 ==> 1 |
| 1111. Dentary, ventral bar, depth below Meckelian fossa | 0 ==> 1 |
| 1168. Angular, 2ndry ridge, form | 0 ==> 1 |
| 1647. Fibula, ratio of midheight length to total height | 0 ==> 1 |
| 1657. Fibula, proximal end, dorsal margin, form | 0 ==> 1 |
| 1672. Fibula, anterior surface proximal surface to bipartite scar | 1 ==> 2 |
| 1675. Fibula, medial fossa, differentiation | 0 ==> 1 |
| 16.83. Fibula, ventral end below bipartite scar, form | 0 ==> 1 |
| 1848. Growth rings, number | 3 ==> 4 |

and, as seen in subadults, they initially have a high number of synontomorphies, which thereafter drops in frequency (Figs. 2 and 4). In total, this growth stage has 77 growth changes; 29 have a CI of 1.0 (Table 8). In the skull, the greatest number of changes is seen in the skull roof and braincase modules. In contrast to subadults, growth changes are seen across every module of the skull. Postcranial changes are limited to the hindlimb. This category corresponds to growth stages 8–12 of the ontogram (Fig. 2), occurring in the last half of the exponential phase of the *T. rex* growth curve (*Erickson et al., 2004*; Fig. 12).

Mass estimates for the young adult TMP 1981.006.0001 range from 3,230 kg (*Erickson et al., 2004*) to 4,469 kg (*Persons, Currie & Erickson, 2019*). Another young adult,

MOR 1125, has been estimated at 6,100 kg (*Persons, Currie & Erickson, 2019*). This doubling in mass from 1,810 kg subadults to 3,000 kg young adults occurred halfway through life, in the 3-year interval (15–18 years) in the middle of the exponential phase of growth (*Erickson et al., 2004*); this ontogenetic increase in mass corresponds to the phylogenetic threshold in mass, where *T. rex* exceeds the 3,000 kg limit of its closest relatives, namely *Daspletosaurus torosus* and *T. bataar* (*Snively et al., 2019*). Therefore, this character state in *T. rex* is hypermorphic in that it suprasses the ancestral condition (cf. *Erickson et al., 2004*).

### Skull architecture

In the seventh growth stage, the rostral margin of the rostroventral lamina of the lacrimal is convex, the jugal ramus of the postorbital is rostrocaudally wide and parallel-sided, the apposed frontals are wider than long, the caudal end of the joint surface for the prefrontal on the frontal is positioned below the joint surface for the postorbital, the orbital surface of the frontal is horizontally oriented, the midline strut of the nuchal crest that is above the supraoccipital is prominent, the ventral bar of the dentary that is below the Meckelian groove is dorsoventrally deep, and the secondary ridge on the medial surface of the angular is distinct.

In the eighth growth stage, the dorsal and ventral margins of the ascending ramus of the maxilla converge as they extend caudally, the point of rostralmost incursion of the internal antorbital fenestra is positioned within less than five teeth of the caudal end of the tooth row, the subocular process of the postorbital is as long or longer than the frontal process, and the chin of the dentary is positioned ahead of the fourth alveolus.

In the ninth growth stage, the caudal end of the antorbital fossa has a uniform height below the internal antorbital fenestra and it is obliterated (i.e., fades out) ahead of the jugal ramus, and the lateral margin of the premaxilla is straight or convex. In the tenth growth stage, the concavity in the caudal surface of the nuchal crest fades below the dorsal margin of the crest, and the ventral surface of the neck of the occipital condyle has the form of a deep pit between pillars. In the eleventh growth stage, the dorsal surface of the neck of the occipital condyle is penetrated by a shallow pit.

### Joint surfaces

In the seventh growth stage, the orbital notch in the frontal bone is pinched between the joint surfaces for the lacrimal and postorbital; the dorsal joint surface for the quadrate of the quadratojugal is wide and coarse, its rostral margin is concave, and its caudal half is developed into a wide ridge; the frontoparietal junction is wider than the sagittal crest rostral and caudal to it; and the joint surface for the dentary on the splenial is reinforced by a wide longitudinal ridge.

In the ninth growth stage, the joint surface for the quadratojugal on the jugal is positioned far caudal to the cornual process. In the tenth growth stage, the joint surface for the squamosal on the postorbital is coarsened by deep and numerous longitudinal ridges, the caudal shelf of the postorbital extends subtly ventrolaterally, and a ridge extends along the laterosphenoidoprootic suture. In the eleventh growth stage, the rostral margin

of the parietal extends mediolaterally (i.e., it is not wedge-shaped), the postorbital buttress extends caudal to the midlength of the dorsotemporal fossa, and the lateral extent of the parietofrontal suture extends from mediolaterally to caudolaterally.

### Subcutaneous surface

In the seventh growth stage, the subcutaneous surface of the ventral ramus of the lacrimal is coarse with papillae concentrated toward its rostral margin. In the eighth growth stage, lateral fossae are present on the alveolar process of the maxilla. In the ninth growth stage, the circumfossa ridge of the maxilla is low or absent, and the texture of the subcutaneous region of the chin is highly rugose. In the tenth growth stage, the groove that extends across the prefrontolacrimal process is deeply incised.

### Cephalic ornamentation

In the ninth growth stage, the cornual process of the postorbital exceeds the height of the bone.

### Paranasal pneumaticity

In the eighth growth stage, the maxillary fenestra closely approaches the ventral margin of the external antorbital fenestra, the interfenestral strut is rostrocaudally narrow, the caudodorsal margin of the maxillary fenestra is straight or concave, the ventral margin of the maxillary fenestra is obliterated and incorporated into the antorbital fossa, and the mediolateral depth of the antorbital fossa is deep. In the ninth growth stage, the dorsal surface of the lacrimal extends dorsomedially, and the ventral edge of the subcutaneous surface at the septum ahead of the lacrimal pneumatic recess extends rostroventrally or grades into the antorbital fossa without a distinct orientation.

### Endoccipital pneumaticity

In the eighth growth stage, the caudal surface of the paroccipital process is deeply concave and the caudodorsal surface is strongly convex. In the ninth growth stage, the bar that extends across the level of the lateral processes of the supraoccipital is prominent.

### Basicranial pneumaticity

In the eighth growth stage, the subcondylar fossa is obliterated by inflation. In the ninth growth stage, the pneumatic foramina of the subcondylar fossa are positioned far apart from each other (cf. *Witmer & Ridgley, 2009*). In the eleventh growth stage, the rim of the caudoventral end of the subsellar recess is spout-like, and the ventral end of the preotic pendant is a low ridge.

### Dorsotemporal fossa

In the seventh growth stage, the sagittal crest of the frontal is equal to or greater than 37% the length of the bone, and the sagittal foramen is positioned far caudal to the rostral margin of the fossa (indicating rostralward encroachment of the musculature).

### Muscle scars

In the seventh growth stage, the ridge along the squamosal process of the quadratojugal is mediolaterally wide in rostral view, and the nuchal crest is rugose but the texture does not

reach the midline of the crest. In the eighth growth stage, the lateral half of the nuchal surface of the squamosal is coarsely textured, and the oval scar of the basisphenoid is oriented medioventrally. In the ninth growth stage, the oval scar faces lateroventrally and it is coarse in texture.

In the tenth growth stage, the rugose texture of the nuchal crest of the parietal reaches the midline of the crest, the ventral end of the metotic strut is scoured by the oval scar complex, and the basal tuber is coarse in texture. In the eleventh growth stage, the pits mediodorsal to the supraoccipital extend deeply into the nuchal crest, the dorsal rugosities of the nuchal crest extend onto the rostral surface of the crest, and the ventrolateral edge of the base of the rostrolateral process of the parietal is crossed by a subtle, ventrolaterally extending ridge.

### Dentition

In the eighth growth stage, the first mesial alveolus of the dentary is substantially smaller than the alveoli at the middle of the tooth row (i.e., the second alveolus is enlarged), and the number of dentary alveoli is 13.

### Appendicular skeleton

In the seventh growth stage, the fibula is stout, the proximal end of the bone is differentiated into distinct anterior and posterior knob-like convexities that produces a deeply concave dorsal margin, the anterior surface of the fibula distinctly widens toward its distal (=ventral) end, the medial fossa of the fibula is sharply differentiated into anterior and posterior fossae, the posterior extent of the medial fossa is a wide ridge that separates the fossa from the posterior edge of the bone, and the anterior surface of the bone above the iliofibularis tubercle (= bipartite scar) is wide and flat before becoming dorsally a narrow but blunt ridge.

## Adults

Adults (=young adult of *Erickson et al., 2004*) have a skull length of 1.3–1.4 m, a skull length to height ratio of 2.6–2.3, are greater than 22 years old, have an EFS, and, as is seen in subadults and young adults, initially have a high number of growth changes, which rapidly drops. Sixty-four synontomorphies occur in this category, and 13 of them have a CI of 1.0 (Table 9). In the skull, the greatest number of changes, 15 and 13, occurs in the snout and skull roof modules, respectively, and changes are seen in all modules (Table 4). Eight changes occur in the postcranium, where five occur in the pelvic girdle and limb, and three are seen in the axial skeleton. The adult stage corresponds to growth stages 13–20 of the ontogram (Fig. 2) and to the first part of the stationary phase of the *T. rex* growth curve (*Erickson et al., 2004*; Fig. 12). An ontogenetic progression among adults, based on size, was first hypothesized by *Paul (1988)*, "AMNH FARB 50270 and other big *T. rex* specimens may or may not be subadults. This is possible because the biggest specimen (UCMP 118742)…is 29 percent longer than 5027" (1988: 344–345). Indeed, the results here recover this exact hypothesis of relative maturity (Figs. 1 and 2).

Several mass estimates are available for adults (Table 5). The mass for adults is 10,200 kg in Henderson & Snively (2004), 6,071.8 kg in *Bates et al. (2009)*, 5,777–10,768 kg in

**Table 9 Diagnostic characters of the adult and senescent growth categories of *Tyrannosaurus rex*.** These characters are unambiguously optimized on the topology and have a consistency index (CI) of 1.0 indicating that they have changed only once on the ontogram and so can be considered diagnostic of the growth category. The last row corresponds to the senescent adult character. PHYLO, corresponding homologous phylogenetic character (see Data S1).

| Character | Growth change |
|---|---|
| 147. Cervical vertebrae, centrum, hypapophysis, presence (PHYLO 280) | 0 ==> 1 |
| 166. Ilium, pubic peduncle, ventral margin, orientation (PHYLO 333) | 0 ==> 1 |
| 546. Jugal, medial maxillary process, rostral end, depth inset | 1 ==> 2 |
| 590. Jugal, ventral quadratojugal process, dorsolateral surface | 1 ==> 2 |
| 671. Squamosal, quadratojugal process, ventral margin, concavity | 0 ==> 1 |
| 694. Squamosal, joint surface for otoccipital, form | 0 ==> 1 |
| 735. Quadratojugal, jugal ramus, ventral margin, form | 1 ==> 2 |
| 1051. Basisphenoid, lateral margin, form | 0 ==> 1 |
| 1223. Axis, axial intercentrum, joint surface, ratio height to mediolateral width | 0 ==> 1 |
| 1256. Axis, spinous process, dorsolateral process, dorsoventral height | 0 ==> 1 |
| 1613. Ischium, semicircular scars, number | 0 ==> 1 |
| 1668. Fibula, distal end, anterior margin, form | 0 ==> 1 |
| 1849. EFS, presence | 0 ==> 1 |
| 1220. Axis, axial intercentrum, anterior joint surface, ventral margin, separation | 1 ==> 2 |

*Hutchinson et al. (2011)*, 4,660–10,007 kg in *Campione et al. (2014)*, 6,986.6 kg in *Snively et al. (2019)*, and 5,112–8,870 kg in *Persons, Currie & Erickson (2019)*. The minimum estimated masses of adults are greater than 4,500 kg (*Hutchinson et al., 2011*; *Campione et al., 2014*; *Persons, Currie & Erickson, 2019*), which overlaps with that of young adults (Table 5). In contrast, the mean mass estimates for adults exceed those for young adults, indicating that a mass greater than 5,100 kg might be diagnostic for adults, whereas the range of mass in young adults is from ~3,000 kg to ~5,100 kg. Given the variety of techniques used for estimating mass in *T. rex*, the differences in the results between them, and the incomplete sampling of specimens, mass is here only tentatively considered informative in distinguishing between the young adult and adult categories.

### Skull architecture

In the thirteenth growth stage, the lateral depression of the jugal is shallow and enhanced by ridges, the surangular shelf extends lateroventrally, the angulation in the ventral margin of the maxilla is distinct, the ventral jugal process of the maxilla is a massive convex strut, the rostrodorsal margin of the rostral ramus of the lacrimal extends sharply rostroventrally, the rostral end of the medial maxillary process of the jugal above the joint surface for the maxilla is sharply inset caudally, the base of the medial process of the squamosal is separated by a notch from the margin of the dorsotemporal fenestra, the caudodorsomedial edge of the shaft of the quadrate is distinctly concave, the rostral margin of the vomeropterygoid process of the palatine is positioned ahead of the rostral palatine recess, and the lateral margin of the basisphenoid is deeply embayed.
In the fourteenth growth stage, the narrow caudal end of the ascending ramus of the maxilla above the internal antorbital fenestra twists along its course, the tip of the rostrodorsal process of the lacrimal extends rostrally past the midlength of the internal antorbital fenestra, the lateral depression of the postorbital process of the jugal reaches the level of the rostral end of the postorbital contact, the parietal is elaborated into a ridge above each of the tines of the dorsal process of the supraoccipital, the Meckelian groove of the dentary is positioned at the midheight of the bone rostrally and above the midheight caudally, and the caudal margin of the dentary is deeply concave.

In the fifteenth growth stage, the convex dorsal margin of the dentary extends to alveolus 7. In the seventeenth growth stage, the dorsal margin of the postorbital is swollen such that the medial eversion is completely obscured. In the eighteenth growth stage, the medial edge of the caudal margin of the quadratojugal is positioned caudal to the lateral edge of the bone. In the nineteenth growth stage, the rostralmost incursion of the internal antorbital fenestra is at the level of the fifth tooth from the caudal end of the tooth row, the rostroventral margin of the ventral ramus of the lacrimal is straight, the lateral ridge of the ventral postorbital process of the squamosal extends more ventrally than laterally, the dorsal margin of the jugal ramus of the quadratojugal extends to the medial edge of the shaft, and the ventral margin of the ramus if ventrally convex.

In the twentieth growth stage, the quadrate fossa of the quadratojugal is shallow in depth, the ventral ramus of the lacrimal below the medial pneumatic recess is convex in vertical section, the nasal ramus of the frontal is wide and truncated, a ventrally-extending flange is absent from the ventral margin of the basioccipital, and the Meckelian groove is positioned below the midheight of the dentary rostrally and above midheight caudally.

### Joint surfaces

In the thirteenth growth stage, the dorsolateral surface of the ventral quadratojugal process of the jugal is dominated by a single large ridge, the dorsal margin of the joint surface for the quadrate of the ventral quadrate process of the quadratojugal extends rostrodorsally, the joint surface for the pterygoid on the caudal process of the ectopterygoid is deeply excavated, and the joint surface for the splenial on the ventral bar of the dentary has a peg-in-socket form.

In the fourteenth growth stage, the condylar surface of the occipital condyle has a distinct marginal rim. In the sixteenth growth stage, the joint surface for the splenial on the lingual bar of the dentary is inset. In the eighteenth growth stage, the concavity in the ventral margin of the quadratojugal process of the squamosal is absent, and the joint surface for the otoccipital on the squamosal is convex.

In the nineteenth growth stage, the distal end of the jugal ramus of the postorbital is wedge-shaped to fit into a complementary groove in the jugal, the joint surface for the medial frontal process of the nasal is long and reaches caudally past the rostral end of the joint surface for the prefrontal, the rostral margin of the Meckelian fossa extends ventrolaterally to the medial surface of the bone, and the prearticuloangular buttress is

present. In the twentieth growth stage, the joint surface for the palatine is present medially and rostrally on the rostroventral ala of the lacrimal.

### Paranasal pneumaticity

In the thirteenth growth stage, the distal recess of the rostral ramus of the lacrimal faces more ventrally than laterally. In the nineteenth growth stage, a strut extends across the medial wall of the secondary fossa of the jugal.

### Basicranial pneumaticity

In the twentieth growth stage, the subcondylar foramen of the basioccipital emits a deeply inset channel from its lower margin.

### Neurovasculature

In the thirteenth growth stage, the medial surface of the ventral ramus of the lacrimal is scoured by deep sulci. In the fourteenth growth stage, the subnarial foramen produces a deep notch in the rostral margin of the maxilla, and lateromedially penetrating foramina in the antorbital fossa of the maxilla are present. In the eighteenth growth stage, the caudal surface of the jugal process of the ectopterygoid is not perforated by a foramen, and lateromedially penetrating foramina are absent from the antorbital fossa of the maxilla.

### Dorsotemporal fossa

In the thirteenth growth stage, the rostral margin of the dorsotemporal fossa of the frontal is elaborated into a ridge or crest.

### Subcutaneous surface

In the thirteenth growth stage, the texture of the caudal process of the squamosal is coarsely rugose.

### Cephalic ornamentation

In the nineteenth growth stage, the epipostorbital is fused to the underlying bone.

### Dentition

In the thirteenth growth stage, the number of dentary teeth is 14. In the seventeenth growth stage, the rostral margins of the interdental plates are notched (almost certainly an epiphenomenon of tooth enlargement).

### Axial skeleton

In the fifteenth growth stage, the joint surface of the axial intercentrum is dorsoventrally deep, and the spinous process of the axis is dorsoventrally massive. In the nineteenth growth stage, the hypophysis is present on the cervical centra.

### Appendicular skeleton

In the fourteenth growth stage, the ischium has two semicircular scars. In the eighteenth growth stage, the ventral margin of the public peduncle of the ilium extends horizontally and the anterior surface of the fibula below the bipartite scar is narrow. In the nineteenth growth stage, the distal end of the shaft of the fibula flares abruptly to the
ventral joint surface that produces a deep crease in the shaft, and the anterior margin of the distal end of the fibula bulges anteriorly.

## Senescent adults

Finally, senescent adults have a skull length of at least 1.4 m, are at least 28 years old, and have an EFS. In the skull, the greatest number of changes occurs in the skull roof module, and the rest of the synontomorphies are limited to the snout, suspensorium, and the braincase modules, and postcranial changes occur in the axis (Table 4). Of the 19 synontomorphies that support this growth stage, only one has a CI of 1.0 (Table 9). This category corresponds to growth stage 21 of the ontogram (Fig. 2) and to the end of the stationary phase of the *T. rex* growth curve (*Erickson et al., 2004*; Fig. 12); there is a single exemplar specimen, FMNH PR2081.

The estimated mass for FMNH PR208 is greater than those obtained for other adults in *Hutchinson et al. (2011)* and *Snively et al. (2019)* (Table 5). However, its mass is exceeded by an adult in the comprehensive samples of *Campione et al. (2014)* and *Persons, Currie & Erickson (2019)*; therefore, the mass of senescent adults does not exceed the range seen in adults and so it cannot be used to diagnose this growth stage.

### Skull architecture
The antorbital fossa below the internal antorbital fenestra is extremely shallow, low ridges extend into the quadrate cotyle of the squamosal, and the orbital notch is a deep cleft in the orbital surface of the frontal.

### Joint surfaces
The joint surface for the palatine on the rostroventral ala of the lacrimal is present medially only, the crease between the postorbital buttress and shelf of the frontal is present, and the joint surface for the prefrontal in the orbital surface of the frontal is positioned close to the orbital notch.

### Subcutaneous surface
The dorsolateral surface of the lacrimal is rugose.

### Cephalic ornamentation
The cornual process of the postorbital does not interrupt the rostrolateral corner of the dorsotemporal fossa.

### Paranasal pneumaticity
The caudalmost extent of the trough medial to the tooth root bulges is limited to the maxillary antrum.

### Neurovasculature
The medial surface of the ventral ramus of the lacrimal is smooth, the foramen that penetrates the orbital surface of the jugal ramus of the postorbital is far medial to the lateral margin of the bone, and the ventral surface of the basituberal web is not penetrated by a foramen.

### Muscle scars

The suborbital ligament scar is convex or bulbous, the lateral surface between the jugal process and ventral quadrate process of the quadratojugal is scoured by a deep fossa that is secured by a ridge, and the mediolaterally extending ridge ahead of the parietofrontal suture is absent.

### Axial skeleton

The anterior surface of the axial intercentrum is deeply concave and bounded ventrally by a prominent rim, a deep groove separates the ventral edge of the anterior joint surface of the intercentrum from the ventral surface of the bone, the anterior margin of the intercentrum is concave and the posterior margin is convex, and the concave ventral surface of the intercentrum is limited to the ventral and lateroventral surfaces of the bone by the swollen parapophysis.

## Craniomandibular variation

The results obtained here provide an opportunity to review the variation reported by *Molnar (1990)* in his extensive descriptive monograph of the skull and mandibular rami of *T. rex*, and recast in the context of ontogeny.

### Facial skeleton

The foramen in the base of the antorbital fossa of the maxilla in LACM 23844 is individual variation. The difference in shape of the maxillary fenestra seen between adults is almost certainly an artifact of variability in the form of its caudodorsal margin, ventral margin, and rostral margin. Comparison of the shapes of the fenestra (trapezoidal, oval, triangular) described by *Molnar (1990)* with the ontogram do not show a sequential pattern. The absence of "discernable sculpture" on the maxilla of TMP 1981.006.0001 (*Molnar, 1990*: 142), taken here to mean the absence of fossae on the horizontal ramus, represents individual variation.

The difference in the angle and position of the flexure of the jugal ramus of the maxilla might correspond to relative maturity, where it changes from rostral and distinct (e.g., AMNH FARB 5027, SDSM 12047) to caudal and indistinct (e.g., LACM 23844). Likewise, the change from a large third maxillary antrum chamber (e.g., LACM 23844; *Molnar, 1990*: Fig. 2) to a large second chamber (e.g., UCMP 118742) corresponds to an increase in relative maturity. The foramen in the caudodorsal surface of the interfenestral strut in CM 9380 is individual variation. The dorsoventral position of the palatal process of the maxilla is broadly congruent with relative maturity, where it is ventral in position in relatively immature adults (e.g., AMNH FARB 5027, SDSM 12047) and dorsal in relatively mature adults (UCMP 118742). However, these conditions are seen in specimens that share growth stage 17 (ventral in CM 9380, dorsal in LACM 23844). This character was not included in the character matrix; it may turn out to separate the specimens at growth stage 17 from each other, if, and only if, the character states can be verified and the analysis re-run with them included. The absence of interdental pits from the maxilla are individual variation unique to SDSM 12047.

Although *Molnar (1990)* regarded absence of sculpture on the subcutaneous surface of the maxilla, lacrimal, and nasal in TMP 1981.006.0001 as evidence for immaturity, based on the results here it is individual variation, as well as the coarse texture of the nasal that is seen in MOR 008. The difference between flattened premaxillary processes of the nasal in AMNH FARB 5027 and the rod-like processes in LACM 23844 and MOR 008 are consistent with increasing relative maturity.

*Molnar's (1990)* comments on the relative development of the cornual process of the postorbital deserve some comment. The process is a composite structure that includes a rugosity that is an intrinsic part of the bone to which an overlying osteoderm, termed here the epipostorbital, is attached. The epipostorbital is sometimes difficult to distinguish from the underlying postorbital, but it is clearly distinguished by its distinctly flat ventral surface along which a distinct, tuberculate, and eave-like flange extends (*Molnar, 1990*). The epipostorbital is not preserved in LACM 23844, which accounts for its difference from the other specimens. The progression in development regarded by *Molnar (1990)*, from SDSM 12047 to AMNH FARB 5027 to MOR 008, does follow the progression of maturity found here. However, the reduced condition of the epipostorbital among the most mature specimens (growth stages 19, 20), indicates a reversal of the pattern. A growth-related reduction in ornamentation is seen in other dinosaurs (e.g., *Triceratops*, *Pachycephalosaurus*; *Horner & Goodwin, 2006*; *Horner & Goodwin, 2009*).

### Palate and quadrate

The rostral process of the vomer is large in the less mature AMNH FARB 5027, whereas it is small in the more mature LACM 23844. The pits seen in the caudal process of LACM 23844 are regarded here as damage or individual variation, since they are not seen in any other *T. rex* specimens (AMNH FARB 5027, BMRP 2002.4.1, MOR 008, UWBM 99000). The foramen in the ventral surface of the palatine in AMNH FARB 5027 is regarded here as either individual variation or a lesion, since this feature is not seen in any other specimen of *T. rex*. The difference in texture of the caudal process (=anteroventral part of the pterygoid limb (*Molnar, 1990*)) of the ectopterygoid is not congruent with the growth series found here, which follows a sequence from SDSM 12047 (not rough) to LACM 23844 (rough) to MOR 008 (not rough). The thickness of the quadrate process of the pterygoid is congruent with the growth series, which is thick in the subadult category (LACM 23845), whereas it is thin in adults (e.g., LACM 23844, MOR 008, SDSM 12047). A possible ontogenetic progression is seen on the medial surface of the orbital process of the quadrate, which is coarse in the less mature SDSM 12047, whereas it is smooth in the more mature LACM 23844.

### Braincase

*Molnar (1990)* reported that the joint surface for the laterosphenoid on the postorbital is variable, where it is convex (AMNH FARB 5117), shallowly concave (TMP 1981.012.0001), or deeply concave (LACM 23844). However, examination of AMNH FARB 5117 for this study found that the joint surface is concave. It is possible that the difference between shallow and deep states is ontogenetically informative, but the rim that

surrounds the joint surface is often damaged and missing, preventing a precise assessment of depth.

### Mandibular ramus

Molnar suggested that the intercoronoid is fused to the dentary in most specimens of *T. rex*, but that condition has not been seen during the course of this study; in all cases, the bones are separate. *Molnar (1990)* is correct in stating that the right angular of MOR 008 is coarsened by lesions, and so this is not a growth-related change. The fusion of the prearticular reported by *Molnar (1990)* in MOR 008 might be a growth-related co-ossification event. The thickened surangular shelf seen in LACM 23844, in contrast to the thinner condition reported in AMNH FARB 5027 (*Molnar, 1990*), might be another growth-related change.

Molnar (1990) noted the presence of perforations in the lateral plate of the surangular in *T. rex*; these purportedly pathological features (*Wolff et al., 2009*) deserve comment. Three perforations occur in the same location in different specimens. The first penetrates the bone rostroventral to the caudal surangular foramen (csf) at the dorsal margin of the angular; this is seen in two young adults (MOR 1125, UWBM 99000) and in one adult (MOR 980). The second penetrates the region ahead of the csf below the surangular shelf; this is seen in one young adult (UWBM 99000), three adults (MOR 008, MOR 980, RSM 2523.8), and in the senescent adult (FMNH PR2081). The third penetrates the rostroventral quadrant of the bone ahead of the external mandibular fenestra, which is seen in one young adult (UWBM 99000), an adult (MOR 008), and in the senescent adult (FMNH PR2081). Given the spotty distribution of these openings, and, indeed, their lesion-like appearance, they were not included in the cladistic analysis of growth; regardless, the regularity of their positions indicates that they might not be lesions.

### Suture closure

Molnar documented numerous cranial (postorbitojugal, quadratoquadratojugal, squamosoquadrate) and mandibular (angulosurangular, prearticulosurangular) suture closures in the adult MOR 008, which he regarded as ontogenetic in nature. Given the relatively immature maturity of the specimen, and the fact that closure of these sutures are not seen in other *T. rex*, and they are regarded here as individual variation; in at least one case, a clear lesion is seen at the point of fusion (e.g., right angulosurangular joint). Also, it is not obvious that the right quadratoquadratojugal suture is closed in the specimen.

### Ontogenetic variation among adults

*Molnar (1990)* regarded the texture of the subcutaneous surface of facial bones, inflation of cranial sinuses, joint fusions, development of the cornual process of the postorbital, as reflecting relative maturity. Also, he proposed that the variable shape of the maxillary fenestra and the joint surface for the laterosphenoid of the postorbital represent individual variation. The results here broadly agree with these assessments. He was less certain of the significance of the form of the premaxillary process of the nasal, difference in height of the palatal process of the maxilla and the form of the rostral plate of the vomer

**Table 10 Data used in the correlation test of bite force and maturity in *Tyrannosaurus rex*.** The data for BMRP 2002.4.1are from *Bates & Falkingham (2012)* and the data for the others are from *Gignac & Erickson (2017)*. Data used in the correlation test are in boldface.

| Specimen | Growth rank | Revised growth rank | Bite force | Bite force rank |
|---|---|---|---|---|
| BMRP 2002.4.1 | 4 | **1** | 2,400–3,850 N | **1** |
| TMP 1981.006.0001 | 8 | **2** | 12,197–21,799 N | **2** |
| MOR 980 | 16 | **3** | 14,201–30,487 N | **4** |
| LACM 23844 | 17 | **4** | 16,352–31,284 N | **5** |
| MOR 008 | 18 | **5** | 13,736–28,101 N | **3** |
| FMNH PR2081 | 20 | **6** | 17,769–34,522 N | **6** |

(*Molnar, 1990*). The present study does not resolve the latter set of features, but they will be taken up in a future iteration of this work.

## Secondary metamorphosis

The abrupt, 2-year transition in *T. rex* from the sleek craniomandibular skeleton of juveniles to the deep and stout form of subadults (a change that is terminal) is an example of secondary metamorphosis, an extreme transformation of morphology that is associated with sexual maturity (*Rose & Reiss, 1993*). Every bone and anatomical domain (e.g., paranasal air sac system, dentition, skull frame, musculature, integumentary system, etc.) is involved in reshaping the entire head skeleton, making juveniles and mature specimens so different that they have been taken to be different taxa (*Molnar, 1980*; *Bakker, Williams & Currie, 1988*). The occurrence of multiple transformative events across the head skeleton (i.e., not a single event) is the primary evidence for metamorphosis (*Rose & Reiss, 1993*). This case of secondary metamorphosis coincides with the trophic shift from juvenile to subadult, and, almost certainly, with sexual maturity, whereas it precedes somatic maturity (*Erickson et al., 2004*). This hypothesis can be rejected if, upon discovery of relatively complete large juvenile and subadult specimens, a quantitative comparison of bite force between large juveniles and subadults does not show a discontinuous increase in the magnitude of bite force between the growth categories.

## Bite force and maturity

Different methods have been used to estimate bite forces of *T. rex*, including adults (e.g., *Erickson et al., 1996*; *Gignac & Erickson, 2017*) and a juvenile (*Bates & Falkingham, 2012*; Table 10). In each case, the maximum estimated bite force of the juvenile is consistently an order of magnitude lower than those for adults (Table 10). The only exception to this is the allometric-scaling based estimate of *Meers (2002)* for adult *T. rex*, which is an order of magnitude lower than the other estimates of maximum bite force that were obtained by other methods; regardless, the minimum estimate (7,600 N; *Meers, 2002*) is greater than that of the maximum bite force of the juvenile (3,850 N; *Bates & Falkingham, 2012*).

Like crocodylians (*Gignac & O'Brien, 2016*), *T. rex* has a high bite force for its body size (*Bates & Falkingham, 2012*). Bite force estimates for juvenile *T. rex* (2,600–4,010 N; BMRP

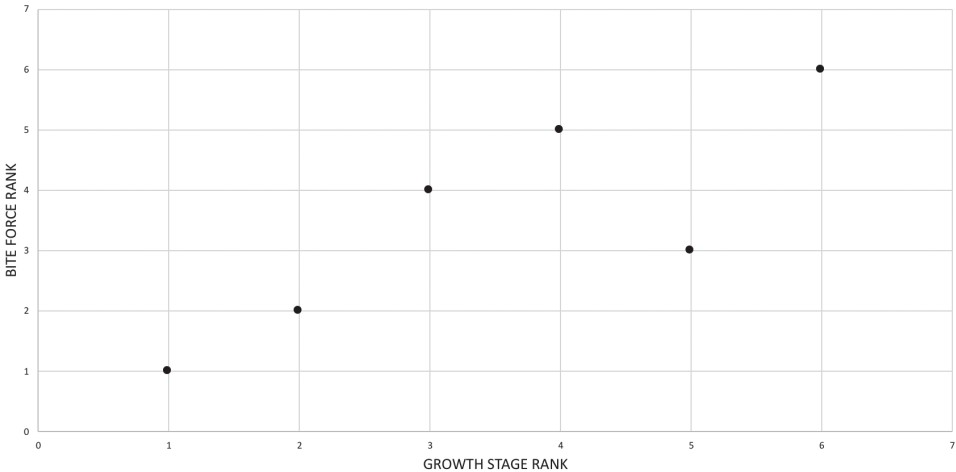

**Figure 13 Scatterplot showing the congruence in *Tyrannosaurus rex* between bite force (i.e., and maturity).** Growth stage rank, corresponding to the increasing sequence of nodes in Fig. 2, is along the *x*-axis; growth stage rank refers to the relative maturity of the specimens for which bite force has been estimated. Increasing bite force rank is along the *y*-axis; raw bite force data are from *Bates & Falkingham (2012)* and *Gignac & Erickson (2017)*. A Spearman correlation test on these data resulted in a significant correlation coefficient, indicating that bite force increases with maturity.

2002.4.1) are lower than those of adults (35,000–57,000 N) and show a growth trend of positive allometry, when identical methods of bite force estimation have been used for both growth categories (*Bates & Falkingham, 2012*); that is, bite forces in adults are relatively and absolutely higher than those seen in juveniles.

In light of these data, a comparison of bite force was made here between a sample of several adult *T. rex* (*Gignac & Erickson, 2017*) and a juvenile (*Bates & Falkingham, 2012*). Given that different methods were used to estimate bite force in the juvenile, on the one hand, and its adult counterparts on the other, the raw estimates were converted into ranks for a Spearman rank correlation test of the congruence between maturity and bite force (Table 10). The data were tested for normality and a Shapiro-Wilk test; the growth ranks and bite force ranks are normally distributed ($p = 0.961$ for both). The Spearman rank correlation test resulted in a significant ($p = 0.042$) correlation coefficient ($r_S = 0.829$). Therefore, an ontogenetic progression of increasing bite force from juvenile to adult, and throughout adulthood (Fig. 13), is seen in *T. rex*, which reflects the condition that is reported in crocodylians (*Gignac & Erickson, 2014*).

Tooth cross sectional shape serves as a proxy for bite force, where the tooth width to length ratio in crocodylians continuously increases with the ontogenetic increase in bite force (*Gignac & Erickson, 2014*). In *T. rex*, the tooth width to length ratio doubles from juveniles (e.g., BMRP 2002.4.1, maxillary tooth 4, width to length ratio: 54%) to adults (e.g., MOR 008, maxillary tooth 4, width to length ratio: 98%), which might be evidence for a delay in the necessity for indenting and fracturing the bones of prey in juveniles, in contrast to the continuous pattern that is seen in crocodylians (*Bates & Falkingham, 2012*; *Gignac & Erickson, 2014*). Ergo, a complete sample of bite force estimates across the
metamorphic transition in *T. rex* is required to test this hypothesis, which is based on gross morphology.

The relationship between bite force and size is linear in extant crocodylians, despite the great changes in the size and type of prey species, a clade that, unlike *T. rex*, does not undergo an abrupt, wholesale ontogenetic transformation in skull shape (*Erickson, Lappin & Vliet, 2003*; *Erickson et al., 2013*). If bite forces in *T. rex* do increase abruptly across metamorphosis, then this is fundamentally unlike the situation in crocodylians, where bite force matches or exceeds prey shear forces in crocodylians (*Gignac & Erickson, 2014*). In contrast, apical tooth pressures in the dinosaur will be found to not equal or exceed the maximum shear strength of significantly larger prey until metamorphosis has occurred. If true, then the ontogenetic change reflects a discontinuous and abrupt dietary change to significantly larger prey that was necessary to sustain the metabolic demands of a 1.8-tonne animal on its way to becoming a 3+ tonne predator.

### Tooth morphology

A comparison of tooth width to length ratios gives some indication of whether or not an abrupt, discontinuous increase in bite force is reasonable to expect in *T. rex* growth. The comparisons made here are based on the lateral teeth of the maxilla and dentary; that is, excluding the incisiform first two teeth of the maxilla and the two mesial subconical teeth of the dentary that have higher width to length ratios than the successive ziphiform teeth. In large juveniles (BMRP 2002.4.1, $n = 34$) the mean width to length ratio across all teeth is 54%; in young adults (MOR 1125, $n = 20$) the ratio is 68%, and in adults (MOR 008, $n = 14$; LACM 23844, $n = 6$; MOR 980, $n = 2$; RSM 2523.8, $n = 10$) the ratio is 79%, 71%, 85%, and 85%, respectively (Table 11).

Bite force estimates for the juvenile are 2,400 N for the mesial teeth and 3,850 N for the distal teeth (*Bates & Falkingham, 2018*); the minimum bite force estimates for a subadult (TMP 1981.006.0001) is 12,197 N, and in adults (MOR 008, LACM 23944, MOR 980) the estimates are 13,736 N, 16,352 N, and 14,201 N, respectively (*Gignac & Erickson, 2017*). Therefore, the order-of-magnitude increase in bite force, and the 12% increase in tooth width that is seen between juveniles and subadults, are naively predicted here to occur abruptly in the subadult stage, no later than 15 years old.

## Tooth count and maturity

In *T. rex*, tooth count in the maxilla and dentary initially increases, and then decreases over the course of the growth series. Despite this general trend, variation is seen from the young adult growth stage onwards; therefore, the congruence between maturity and the tooth number in the maxilla and the dentary was tested using Spearman rank correlation, assuming a null hypothesis of noncorrelation.

### Correlation with maxillary tooth count

The bivariate scatterplot of ranked data (Fig. 14; Table 12) shows that maxillary tooth number initially increases between the first two growth stage ranks (from 15 to 16 teeth)
**Table 11 Summary of width to length ratios of maxillary and dentary teeth in adult *Tyrannosaurus rex* compared with representative adults of other tyrannosaurids.** In *T. rex* the maxillary teeth tend to be wider than in other tyrannosaurids, and also at the mesial end of the tooth row of the dentary. However, the characterization by *Osborn (1906)* of the teeth in *T. rex* as generally wider than long is not supported by these data.

| | T. rex | | | | | | | D. torosus | D. horneri | A. sarcophagus |
|---|---|---|---|---|---|---|---|---|---|---|
| | MOR 008 | RSM 2523.8 | LACM 23844 | MOR 980 | CM 9380 | T. rex mean | BMRP 2002.4.1 | CMN 8506 | MOR 1130 | TMP 1981.010.0001 |
| Mx1 | – | 85% | 71% | – | – | 78% | ?, 150% | 87% | – | – |
| Mx2 | – | – | – | 91% | – | 91% | 73%, 83% | 64% | 79% | 56% |
| Mx3 | – | 65% | 69% | – | – | 67% | 48%, ? | – | – | 68% |
| Mx4 | 98% | – | – | – | – | 98% | ? 54% | 100% | – | 59% |
| Mx5 | – | 77% | 72% | – | – | 74.5% | 50%, ? | – | – | 56% |
| Mx6 | – | – | 71% | 85% | – | 78% | 50%, 56% | 74% | – | 63% |
| Mx7 | 74% | 81% | – | 85% | – | 80% | 45%, ? | – | – | 63% |
| Mx8 | 76% | – | – | – | – | 76% | 53%, 51% | 69% | – | 61% |
| Mx9 | 84% | – | – | – | – | 84% | 54%, ? | – | – | – |
| Mx10 | 84% | 72% | – | – | 77% | 77.7% | ?, 54% | 73% | – | – |
| Mx11 | 84% | – | – | – | – | 84% | ?, 50% | 72% | – | – |
| Mx12 | – | – | n/a | – | – | – | 47%, ? | – | – | 63% |
| Mx13 | n/a | n/a | n/a | n/a | n/a | – | 49%, 51% | – | – | – |
| Mx14 | n/a | n/a | n/a | n/a | n/a | – | 50%, ? | – | – | – |
| Mx15 | n/a | n/a | n/a | n/a | n/a | – | 54%, 53% | – | – | – |
| Mx16 | n/a | n/a | n/a | n/a | n/a | – | ?,? | – | – | – |
| Dn1 | – | 170% | – | – | 140% | 155% | 130%, 120% | – | – | – |
| Dn2 | – | 110% | 73% | – | 94% | 92.3% | 83%, 84% | – | – | – |
| Dn3 | – | – | – | – | 88% | 88% | ?, 70% | – | – | – |
| Dn4 | 82% | 87% | – | – | 84% | 84.3% | 64%, 65% | 78% | – | – |
| Dn5 | – | 120% | 73% | – | 77% | 90% | 58%, ? | 70% | – | – |
| Dn6 | 83% | – | – | – | – | 83% | 61%, 58% | – | – | – |
| Dn7 | 79% | 130% | 76% | – | 82% | 91.8% | ?, 54% | 90% | – | – |
| Dn8 | – | 79% | – | – | – | 76% | 53%, ? | – | – | – |
| Dn9 | 77% | – | – | – | – | 77% | 53%, ? | 80% | – | – |
| Dn10 | 69% | – | – | – | 76% | 72.5% | 57%, ? | 86% | – | – |
| Dn11 | 69% | – | – | – | – | 69% | 53%, ? | 74% | – | – |
| Dn12 | 70% | – | 67% | – | 76% | 71% | ? | – | – | – |
| Dn13 | 80% | 60% | n/a | – | – | 70% | 50%, ? | 75% | – | – |
| Dn14 | n/a | 79% | n/a | – | n/a | 79% | 50%, ? | 72% | – | – |
| Dn15 | n/a | n/a | n/a | – | n/a | – | 58%, ? | – | – | – |
| Dn16 | n/a | n/a | n/a | – | n/a | – | 53%, ? | 83% | – | – |
| Dn17 | n/a | n/a | n/a | – | n/a | – | 65%, ? | n/a | – | – |

before it abruptly decreases (from 16 to 12 teeth) at the third rank (Fig. 14). Tooth count rank is constant until the sixth rank, where variation, toward a lower tooth count (from 12 to 11 teeth), is first seen (Fig. 14). Prima facie, tooth loss covaries with maturity, as is seen in other tyrannosaurids (*Carr et al., 2017*).

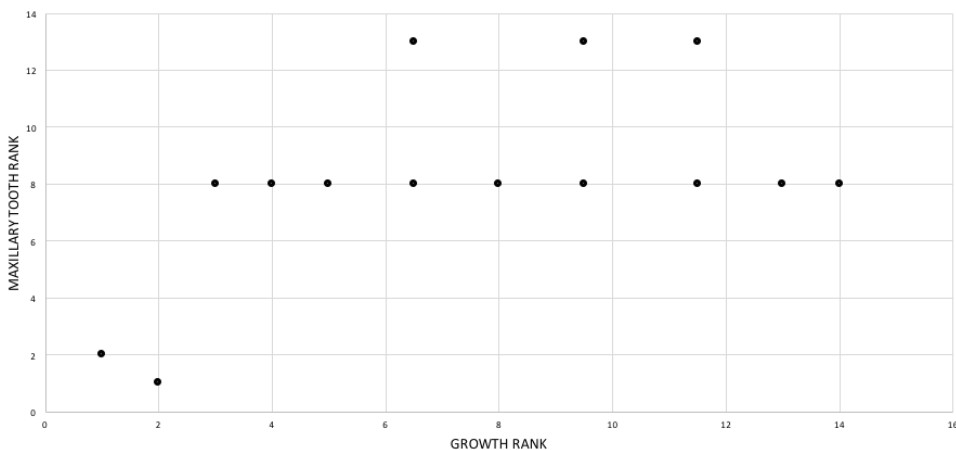

**Figure 14 Bivariate scatterplot showing the relationship between maxillary tooth count with maturity among 14 specimens of *Tyrannosaurus rex*.** Growth rank increases away from the origin (i.e., maturity increases to the right) and corresponds to growth stages for which maxillary tooth count was available for a given specimen; that is, the rank does not correspond to growth stage. Maxillary tooth rank corresponds to relative tooth count, where low ranks correspond to high tooth counts and low ranks correspond to high tooth counts.

**Table 12 Summary of maxillary tooth count data and growth rank in *Tyrannosaurus rex*.** Summary of raw and ranked data used for the Spearman correlation test between maturity and maxillary tooth count in *T. rex*. Boldface indicates the ranks used in the correlation test.

| Specimen | Maxillary tooth # | Maxillary tooth # rank | Maxillary tooth midranks | Growth stage | Growth stage rank | Growth stage midranks |
|---|---|---|---|---|---|---|
| CMNH 741 | 15 | 2 | **2** | 3 | 1 | **1** |
| BMRP 2002.4.1 | 16 | 1 | **1** | 4 | 2 | **2** |
| MOR 1125 | 12 | 3 | **7** | 7 | 3 | **3** |
| TMP 1981.006.0001 | 12 | 4 | **7** | 8 | 4 | **4** |
| UWBM 99000 | 12 | 5 | **7** | 11 | 5 | **5** |
| RSM 2523.8 | 11 | 12 | **13** | 12 | 6 | **6.5** |
| SDSM 12047 | 12 | 6 | **7** | 12 | 7 | **6.5** |
| AMNH FARB 5027 | 12 | 7 | **7** | 13 | 8 | **8** |
| MOR 555 | 12 | 8 | **7** | 16 | 9 | **9.5** |
| MOR 980 | 11 | 13 | **13** | 16 | 10 | **9.5** |
| LACM 23844 | 11 | 14 | **13** | 17 | 11 | **11.5** |
| CM 9380 | 12 | 9 | **7** | 17 | 12 | **11.5** |
| MOR 008 | 12 | 10 | **7** | 18 | 13 | **13** |
| FMNH PR2081 | 12 | 11 | **7** | 20 | 14 | **14** |

A Shapiro–Wilk test found that the maxillary tooth rank data are not normally distributed ($p = 0.004$), whereas the growth rank data are normally distributed ($p = 0.725$); a Spearman rank correlation test of growth rank and tooth count ranks was done, which resulted in a statistically nonsignificant ($p = 0.073$) correlation coefficient ($r_S = 0.494$).

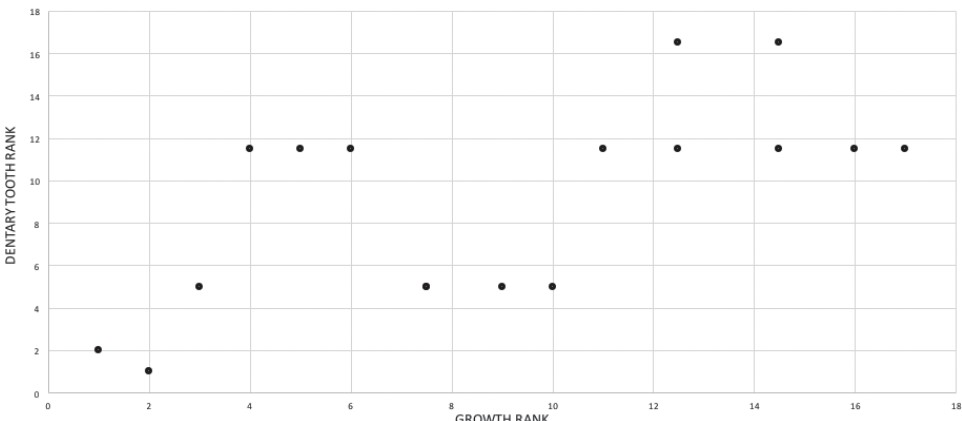

**Figure 15 Bivariate scatterplot showing the relationship between dentary tooth count with maturity among 16 specimens of *Tyrannosaurus rex*.** Growth rank increases away from the origin (i.e., maturity increases to the right) and corresponds to growth stages for which dentary tooth count was available for a given specimen; that is, the rank does not correspond to growth stage. Dentary tooth rank corresponds to relative tooth count, where low ranks correspond to high tooth counts and low ranks correspond to high tooth counts.

Therefore, a general trend of decrease in tooth count is not seen across the entire data set. This result must be regarded with the caveat that there is a gap in the data, where maxillary tooth counts are currently unknown for subadult specimens. It is possible that tooth counts from future discoveries of subadults will bridge the gap between the extremes and result in a significant value.

In addition to that, the hypothesis that tooth count is a reliable proxy for maturity from young adults to senescent adults was tested by limiting the statistical test to the corresponding portion of the data set (growth stage ranks 3–14); a statistically nonsignificant result was obtained ($r_S = 0.112$, $p = 0.729$), indicating that the null hypothesis of no difference could not be rejected. Ergo, maxillary tooth count is an unreliable proxy for maturity among adult-sized specimens.

### Correlation with dentary tooth count

As for the maxilla, an initial increase (from 16 to 17) followed by a decrease in tooth count (from 17 to 14) is seen in the dentary, after which variation is seen (from 14 to 12) among adults (Fig. 15). The raw data were converted to ranks (Table 13), and a Shapiro–Wilk test of normality found that the dentary tooth count rank data are not normally distributed ($p = 0.021$), whereas the growth rank data are normally distributed ($p = 0.616$); a Spearman correlation test of the data resulted a significant ($p = 0.005$) correlation ($r_S = 0.648$), indicating a trend of decrease in tooth count across the data set. Among adults, the hypothesis of dentary tooth count as a proxy for maturity was tested by obtaining the correlation coefficient from young adults to senescent adults (from growth rank 3–17), which resulted in a nonsignificant ($p = 0.074$) correlation ($r_S = 0.475$), indicating that dentary tooth count is not a reliable proxy for estimating maturity among adult-sized animals.

**Table 13 Summary of dentary tooth count data and growth rank in *Tyrannosaurus rex*.** Summary of raw and ranked data used in the Spearman rank correlation test of maturity and dentary tooth count in *T. rex*. Boldface indicates the ranks used in the correlation test.

| Specimen | Dentary tooth # | Dentary tooth # rank | Dentary tooth # midranks | Growth stage | Growth rank | Growth midranks |
|---|---|---|---|---|---|---|
| CMNH 7541 | 16 | 2 | **2** | 3 | 1 | **1** |
| BMRP 2002.4.1 | 17 | 1 | **1** | 4 | 2 | **2** |
| MOR 1125 | 14 | 3 | **5** | 7 | 3 | **3** |
| TMP 1981.006.0001 | 13 | 8 | **11.5** | 8 | 4 | **4** |
| LACM 150167 | 13 | 9 | **11.5** | 9 | 5 | **5** |
| UWBM 99000 | 13 | 10 | **11.5** | 11 | 6 | **6** |
| RSM 2523.8 | 14 | 4 | **5** | 12 | 7 | **7.5** |
| SDSM 12047 | 14 | 5 | **5** | 12 | 8 | **7.5** |
| AMNH FARB 5027 | 14 | 6 | **5** | 13 | 9 | **9** |
| NHMUK R7994 | 14 | 7 | **5** | 14 | 10 | **10** |
| NMMNH P-3698 | 13 | 11 | **11.5** | 15 | 11 | **11** |
| MOR 555 | 12 | 16 | **16.5** | 16 | 12 | **12.5** |
| MOR 980 | 13 | 12 | **11.5** | 16 | 13 | **12.5** |
| LACM 23844 | 12 | 17 | **16.5** | 17 | 14 | **14.5** |
| CM 9380 | 13 | 13 | **11.5** | 17 | 15 | **14.5** |
| MOR 008 | 13 | 14 | **11.5** | 18 | 16 | **16** |
| FMNH PR2081 | 13 | 15 | **11.5** | 20 | 17 | **17** |

## Congruence between chronological age, size, and mass with maturity

Congruence between age, size, and mass (dependent variables) with maturity (independent variable) was tested using Spearman rank correlation. As used here, maturity refers to position of stages along the ontogram. The chronological age of BMRP 2006.4.4. is incongruent with maturity, where its chronological age of 15 years is greater than the more mature LACM 23845 that is estimated as 14 years old (*Erickson et al., 2004*). Given this discrepancy, the age of LACM 23845 is regarded here as an underestimate.

### Correlation with age

A bivariate scatterplot shows congruence of ranked data between growth rank and age rank (Fig. 16; Table 14); a Shaprio–Wilk test found that the chronological age ranks and growth ranks are normally distributed ($p = 0.885$ and $0.933$, respectively). A significant ($p = 0.000$) correlation ($r_S = 0.970$) was obtained between age and maturity, indicating that LAG counts are an accurate predictor of maturity. The $r_S$ of less than 1.0 is caused by the chronological age of one specimen (LACM 23845) that is almost certainly an underestimate (see above), and two specimens (MOR 1125, TMP 1981.006.0001) that have the same LAG count of 18. Therefore, chronological age can be used as a proxy for relative maturity throughout the growth series as long as sampling from a skeleton is sufficiently thorough (cf. *Woodward et al., 2020*).

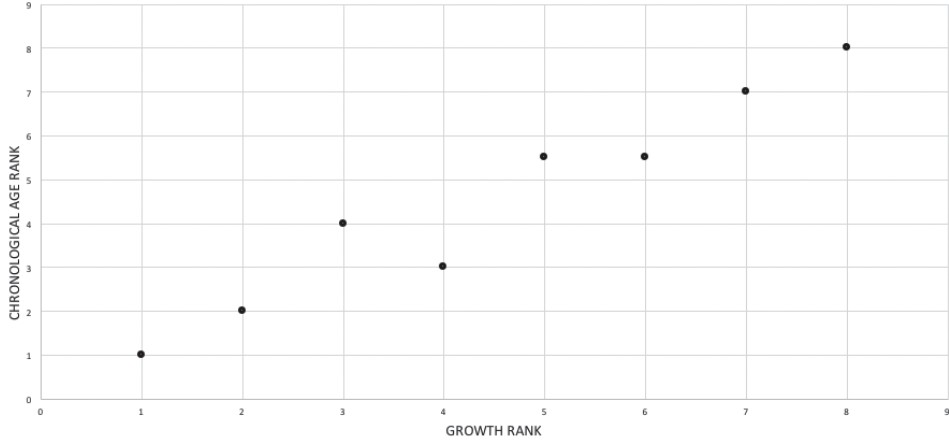

**Figure 16 Bivariate scatterplot showing the relationship between chronological age with maturity among eight specimens of *Tyrannosaurus rex*.** The comparison is limited to specimens that have been histologically aged; growth stages (*x*-axis) and chronological age (*y*-axis) have been converted to ranks. See Table 14 for the raw data.

**Table 14 Summary of chronological age data and growth rank in *Tyrannosaurus rex*.** Summary of the raw and ranked data used in the Spearman correlation test of maturity and chronological age in *T. rex*. Boldface indicates the ranks used in the correlation test.

| Specimen | Chronological age | Chronological age rank | Chronological age midranks | Growth stage | Growth stage rank |
|---|---|---|---|---|---|
| LACM 28471 | 2 | 1 | **1** | 1 | **1** |
| BMRP 2002.4.1 | 13 | 2 | **2** | 4 | **2** |
| BMRP 2006.4.4 | 15 | 4 | **4** | 5 | **3** |
| LACM 23845 | 14 | 3 | **3** | 6 | **4** |
| MOR 1125 | 18 | 5 | **5.5** | 7 | **5** |
| TMP 1981.006.0001 | 18 | 6 | **5.5** | 8 | **6** |
| TMP 1981.012.0001 | 22 | 7 | **7** | 11 | **7** |
| FMNH PR2081 | 28 | 8 | **8** | 20 | **8** |

### Correlation with size

The scatterplot between maturity and size of ranked data shows an overall increasing trend, but with variation among the adult specimens (Fig. 17; Table 15). A Shapiro–Wilk test of normality found that the ranked size data and growth rank data are normally distributed ($p = 0.166$ and $0.678$, respectively). The Spearman rank correlation test resulted in a significant ($p = 0.000$) correlation ($r_S = 0.903$), showing that size can serve as a proxy for maturity; however, this congruence is seen among animals that are less than adult size. A second test was run on the animals in the adult size range (growth ranks 5–15), where a significant ($p = 0.008$) correlation ($r_S = 0.748$) was obtained, indicating that among adult-sized animals, size is a reliable proxy for maturity. Even with MOR 1125, a relatively small young adult (femur length: 1.16 m), excluded (limiting the comparison to growth ranks 6–15), the results are significant ($r_S = 0.661$, $p = 0.038$). However, when the test is limited to large adults (skull length greater than 1.2 m; growth ranks 8–15) the
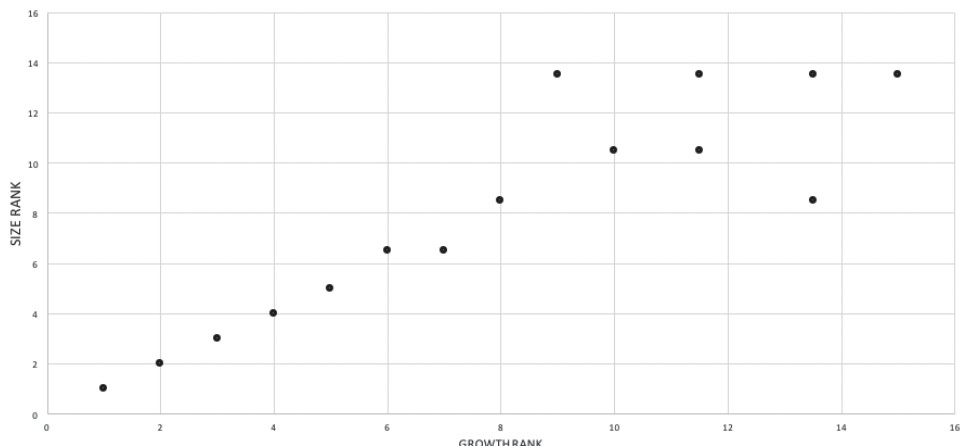

**Figure 17 Bivariate scatterplots showing the relationship between size with maturity among 15 specimens of *Tyrannosaurus rex*.** The comparison is limited to specimens that have comparable size data; growth stages (*x*-axis) and size (*y*-axis) have been converted to ranks. See Table 15 for the raw data.

**Table 15 Summary of size data and growth rank in *Tyrannosaurus rex*.** Summary of raw and ranked data used in the Spearman correlation test of size rank and growth rank in *T. rex*. Boldface indicates the ranks used in the correlation test.

| Specimen | Size (cm) | Size rank | Size midranks | Growth stage | Growth stage rank | Growth stage midranks |
|---|---|---|---|---|---|---|
| LACM 28471 | 40 | 1 | **1** | 1 | 1 | **1** |
| CMNH 7541 | 57 | 2 | **2** | 3 | 2 | **2** |
| BMRP 2002.4.1 | 74 | 3 | **3** | 4 | 3 | **3** |
| LACM 23845 | 80 | 4 | **4** | 6 | 4 | **4** |
| MOR 1125 | 116 | 5 | **5** | 7 | 5 | **5** |
| TMP 1981.006.0001 | 120 | 6 | **6.5** | 8 | 6 | **6** |
| LACM 150167 | 120 | 7 | **6.5** | 9 | 7 | **7** |
| UWBM 99000 | 130 | 8 | **8.5** | 11 | 8 | **8** |
| RSM 2523.8 | 140 | 12 | **13.5** | 12 | 9 | **9** |
| AMNH FARB 5027 | 136 | 10 | **10.5** | 13 | 10 | **10** |
| MOR 980 | 136 | 11 | **10.5** | 16 | 11 | **11.5** |
| LACM 23844 | 140 | 14 | **13.5** | 16 | 12 | **11.5** |
| CM 9380 | 130 | 9 | **8.5** | 17 | 13 | **13.5** |
| MOR 008 | 140 | 13 | **13.5** | 17 | 14 | **13.5** |
| FMNH PR2081 | 140 | 15 | **13.5** | 20 | 15 | **15** |

results are nonsignificant ($r_S = 0.312$, $p = 0.451$), indicating that size cannot serve as a proxy for maturity among the largest adults.

### Correlation with mass

The scatterplot between congruence between maturity and mass of ranked data (Fig. 18; Table 16) shows a cloud of points with an increasing trend, indicating correlation. A Shapiro–Wilk test found that the ranked mass data and growth rank data are normally

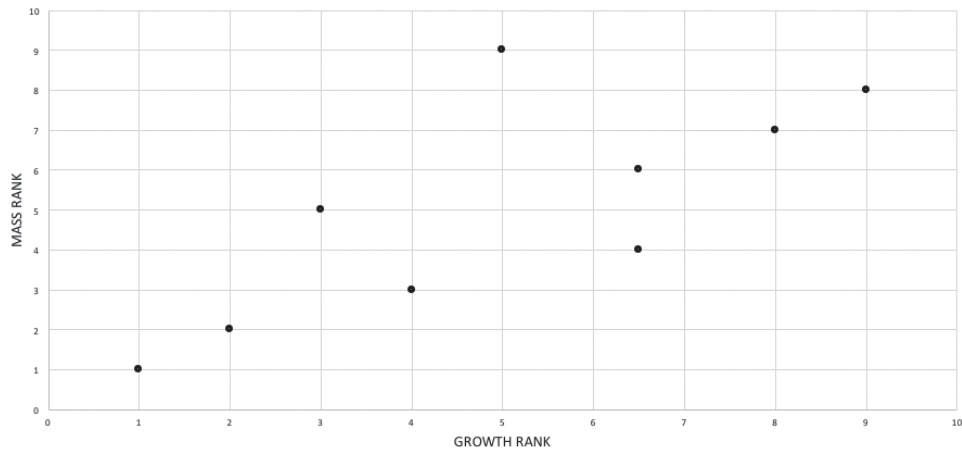

**Figure 18** **Bivariate scatterplots showing the relationship between mass with maturity among nine specimens of *Tyrannosaurus rex*.** The comparison is limited to specimens that have published mass estimates; growth stages (*x*-axis) and mass (*y*-axis) have been converted to ranks. See Table 16 for the raw data.

**Table 16 Summary of mass data and growth rank for *Tyrannosaurus rex*.** Summary of raw and ranked data for the Spearman correlation test of mass and maturity in *T. rex*. Masses are drawn from *Erickson et al. (2004)* and *Persons, Currie & Erickson (2019)*. Boldface indicates the ranks used in the correlation test.

| Specimen | Mass (kg) | Mass rank | Growth stage | Growth stage rank | Growth stage midranks |
|---|---|---|---|---|---|
| LACM 28471 | 25 | 1 | 1 | 1 | **1** |
| LACM 23845 | 98.9 | 2 | 6 | 2 | **2** |
| MOR 1125 | 6,100 | 5 | 7 | 3 | **3** |
| TMP 1981.006.0001 | 4,469 | 3 | 8 | 4 | **4** |
| RSM 2523.8 | 8,870 | 9 | 12 | 5 | **5** |
| MOR 555 | 6,264 | 6 | 16 | 6 | **6.5** |
| MOR 980 | 5,112 | 4 | 16 | 7 | **6.5** |
| CM 9380 | 6,740 | 7 | 17 | 8 | **7** |
| FMNH PR2081 | 8,462 | 8 | 20 | 9 | **8** |

distributed ($p = 0.914$ and $0.870$, respectively). The Spearman correlation rank test resulted in a significant ($p = 0.019$) correlation ($r_S = 0.753$) between the variables; a second run of the test was made for the adult-sized animals (growth stage ranks 3–9) since this region of the distribution shows variation. The test resulted in a nonsignificant ($p = 0.289$) correlation ($r_S = 0.468$), indicating that mass is a poor predictor of maturity among the adult growth categories.

## Congruence between nonbiological factors and maturity

The possibility that the growth series obtained here results from the influence of abiotic factors, namely geographic location and stratigraphic position, was tested using the Spearman rank correlation test since horizontal and vertical spatial data are hierarchical in

**Table 17 Summary of data used in the correlation test between geographic latitude and maturity of _Tyrannosaurus rex_ specimens.** For each fossil, the columns are organized by specimen number, growth stage rank, geographic location rank, and level of resolution of the locality data. Specimens are listed by descending location rank (i.e., from north to south). Co, county; km, kilometer; m, meter; mi, miles; S, section; asterisk (*) indicates the resolution is almost certainly an underestimate.

| Specimen number | Maturity midrank | Location midrank | Resolution |
| --- | --- | --- | --- |
| TMP 1981.012.0001 | 12.5 | 29 | km |
| TMP 1981.006.0001 | 9 | 28 | km |
| RSM 2990.1 | 5.5 | 27 | m |
| RSM 2523.8 | 15 | 26 | m |
| CM 9380 | 22.5 | 25 | S |
| LACM 28471 | 1 | 24 | S |
| LACM 23844 | 22.5 | 22.5 | S |
| LACM 23845 | 7 | 22.5 | S |
| MOR 980 | 20 | 21 | m |
| UWBM 99000 | 12.5 | 20 | m |
| MOR 1131 | 20 | 19 | m |
| MOR 1125 | 8 | 18 | m |
| MOR 555 | 20 | 17 | m |
| UCMP 118742 | 25 | 16 | m |
| MOR 008 | 25 | 15 | m* |
| DDM 344.1 | 2 | 14 | m |
| LACM 150167 | 10 | 13 | m |
| BMRP 2002.4.1 | 4 | 12 | mi |
| BMRP 2006.6.4 | 5.5 | 11 | mi |
| MOR 2822 | 27.5 | 10 | m |
| CMNH 7541 | 3 | 9 | S |
| UWGM 181 | 27.5 | 8 | S |
| FMNH PR2081 | 29 | 7 | mi |
| SDSM 12047 | 15 | 6 | mi |
| AMNH FARB 5117 | 17 | 5 | Co |
| CM 1400 | 11 | 3.5 | Co |
| NHMUK R7994 | 15 | 3.5 | Co |
| UMNH 11000 | 25 | 2 | mi |
| NMMNH P-3698 | 18 | 1 | m |

time and space. For example, the growth series might reflect the north to south distribution of a latitudinally variable population, the anagenetic transformation of a population, or both trends. To test these hypotheses, geographic location was ranked by latitude along a north–south gradient that follows the long axis of Laramidia (Table 17; Fig. 19), and stratigraphic position was ranked according to position within the lower, middle, and upper units of the Hell Creek Formation (HCF; Table 18; Fig. 20). To protect specimen localities and prevent trespassing, the precise locality coordinates are not provided here; locality data are available, to qualified researchers, from the museums where

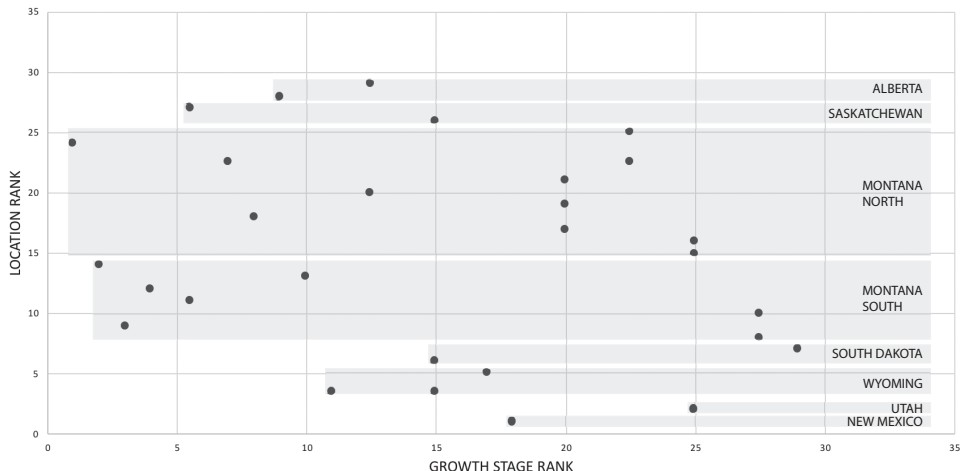

**Figure 19 Bivariate scatterplot showing the relationship between geographic location with maturity among 28 specimens _Tyrannosaurus rex_.** Growth stages (_x_-axis) and geographic location (_y_-axis) have been converted to ranks. See Table 17 for the ranked data. "Montana North" refers to the region of Dawson, Garfield, and McCone counties, and "Montana South" refers to the region of Yellowstone and Carter counties. Maturity increases to the right along the _x_-axis; the _y_-axis follows the north-south axis of North America.                               

**Table 18 Summary of raw and ranked data used in the Spearman correlation test of stratigraphic level in the Hell Creek Formation (and equivalents) and maturity for _Tyrannosaurus rex_.** The specimens are organized by stratigraphic level and the values used in the correlation test are given in boldfaced columns; tied HCF ranks were converted to midranks. HCF, Hell Creek Formation.

| Specimen | HCF level | HCF level rank | HCF rank | HCF level midranks | Growth stage | Growth stage rank |
|---|---|---|---|---|---|---|
| RSM 2990.1 | Upper | 1 | 1 | **2** | 5 | **3** |
| RSM 2523.8 | Upper | 1 | 2 | **2** | 12 | **5** |
| MOR 555 | Upper | 1 | 3 | **2** | 17 | **7** |
| DDM 344.1 | Middle | 2 | 4 | **4.5** | 2 | **1** |
| BMRP 2002.4.1 | Middle | 2 | 5 | **4.5** | 4 | **2** |
| NMMNH P-3698 | Lower | 3 | 6 | **7** | 15 | **6** |
| MOR 1125 | Lower | 3 | 7 | **7** | 7 | **4** |
| FMNH PR2081 | Lower | 3 | 8 | **7** | 20 | **8** |

specimens are accessioned. Data for stratigraphic position was obtained from the literature (_Fowler, 2017_; _Gates, Gorscak & Makovicky, 2019_; _Harrison et al., 2013_; _Horner, Goodwin & Myhrvold, 2011_; _Leslie et al., 2018_), aside from the specimen (DDM 344.1) whose position was obtained from museum records.

### Correlation with geographic position

A Shapiro–Wilk test found that the ranked location data and growth rank data are normally distributed (_p_ = 0.277 and 0.228, respectively). The geographic location test of ranked data (Table 17) resulted in a statistically nonsignificant (_p_ = 0.219) correlation ($r_S$ = −0.236), indicating that the location of the specimens along the growth series was

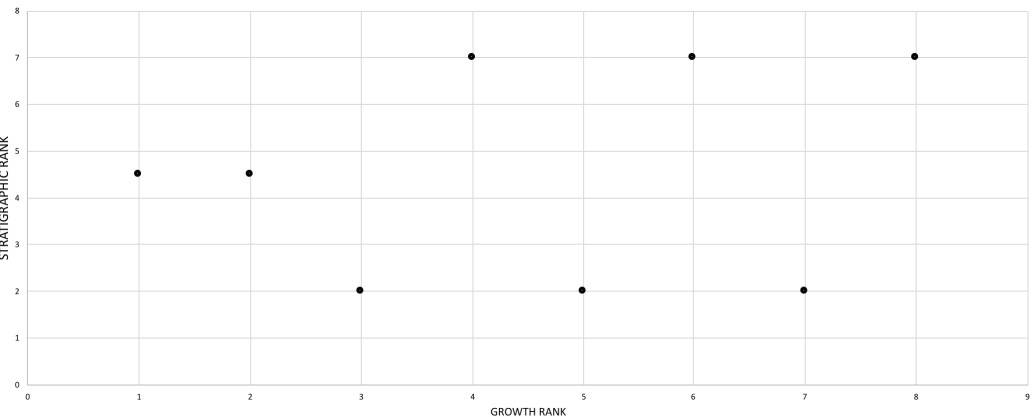

**Figure 20 Bivariate scatterplot showing the relationship between stratigraphic position with maturity among nine specimens of *Tyrannosaurus rex*.** Growth stages (*x*-axis) and stratigraphic position (*y*-axis) have been converted to ranks. See Table 18 for the raw and ranked data. Maturity rank increases to the right; stratigraphic rank decreases from the origin (i.e., the upper HCF is closest to the origin, whereas the lower HCF is furthest from the origin).

not influenced by geographic location. This result is robust given that it was possible to code a geographic location for nearly every specimen in the growth series (Fig. 19). The correlation between unranked data (UTM coordinate northings) and growth stage was tested using a Pearson correlation test; data for 17 specimens were available for this test and the data are skewed to the right. The test resulted in a nonsignificant ($p = 0.588$) correlation coefficient ($r = -0.141$), which is consistent with the test on the ranked data.

### Correlation with stratigraphic level

The stratigraphic schema that is presented here for late Maastrichtian formations from western North America is largely based on *Fowler (2017)*, where the threefold stratigraphic division of the HCF of Montana is used as the basis of comparison with other units. As such, the comparison spans the last million years of the late Maastrichtian, which is divided into three chronostratigraphic slices, in descending order: 398 Kya (equivalent to chron 29r), ~286 Kya, and ~286 Kya (*Fowler, 2017*). Using this approach, the Frenchman Formation of Saskatchewan is equivalent to the upper HCF (*Fowler, 2017*). The radiometric dates and stratigraphic revisions of *Leslie et al. (2018)* were followed to correlate the Tornillo Formation of Texas with the upper HCF. Given the imprecise dating of the Scollard and Willow Creek Formations of Alberta, and ambiguous correlation between them and the upper and middle members of the HCF, specimens from those units were excluded. This comparison was made aware of the fact that the stratigraphic position of specimens might be affected by time transgression and so might not be equivalent. Resolving that issue is beyond the scope of this work, the results of which are offered here as a hypothesis for further, more rigorous testing of stratigraphic correlation.

A Shapiro–Wilk test found that the ranked stratigraphic data are not normally distributed ($p = 0.030$), whereas the growth rank data are normally distributed ($p = 0.933$). A Spearman rank correlation test was run, which recovered a nonsignificant ($p = 0.654$)

correlation ($r_S$ = 0.189) between stratigraphic position and maturity (Fig. 20; Table 18). However, this result is not robust given that only eight specimens are included in the sample and only two are from the middle member of the HCF (BMRP 2002.4.1; *Harrison et al., 2013*; DDM 344.1) (Fig. 20). Therefore, more specimen data are required for a rigorous test of the noncorrelation between maturity and stratigraphic position. Regardless, the scatterplot shows that a young adult, two adults, and senescent adult have been collected from the lower member of the HCF, two juveniles from the middle unit, and one subadult and two adults from the upper member; that is, broadly overlapping growth series have been recovered throughout the unit.

### Dimorphism hypothesis (sensu *Carpenter, 1990*)

*Carpenter (1990)* identified two sets of characters in the skull and postcranium of *T. rex* that he considered to represent patternless variation (noise; i.e., individual variation) on the one hand, and patterned variation (signal; i.e., sexual variation) on the other. Noise included: maxilla depth, size of the maxillary fenestra, shape of the maxillary fenestra, size of the antorbital fenestra, shape of the antorbital fenestra, position of the lacrimal process of the maxilla, position of the jugal process of the maxilla, the shape of the jugal process of the maxilla, and shape of the dentary. In contrast, signal included: robust and gracile morphs of cervical vertebrae and angle of the ischium from the caudal series. The cladistic results obtained here agree that, among adults, the depth of the maxilla and size and shape of the maxillary and antorbital fenestrae, the position and shape of the lacrimal and jugal processes of the maxilla, and shape of the dentary, are noise.

Also, *Carpenter (1990)* found that the maxilla TMM 41436-1 was an outlier in contrast to the sample of adult maxillae, which were scaled to the rostral end of the maxilla and the rostral end of the maxillary fenestra. The incomplete and small TMM 41436-1 was scaled to the height of the maxillary fenestra (*Carpenter, 1990*), which is proportionately taller than in adults (e.g., AMNH FARB 5027). The comparison showed that it is different from the others in that the rostral margin of the bone falls short of the adults, which was used by *Carpenter (1990)* to argue the specimen represents a new taxon; however, based on the results found here, this difference almost certainly arises from the fact its teeth are not as enlarged in adults and the internal sinuses are not expanded, changes that greatly reshape the bone in adults. Therefore, the difference in shape is most simply explained by maturity; when included in the cladistic analysis of ontogeny, the specimen falls out in a polytomy that includes subadult, young adult, and adult specimens (Fig. 1B).

*Carpenter (1990)* found, qualitatively, that the cervical series of AMNH FARB 5027 was less massive than that of NHMUK R7994. In particular, *Carpenter (1990)* drew attention to the form of the atlantal intercentrum and that of the spinous process of the axis and the third cervical vertebra, characters that were included in the cladistic analysis presented here (characters 1,210, 1,256, and 1,280, respectively; see Data S1). The results found that those specimens are ontogenetically sequential, where AMNH FARB 5027 is less mature than NHMUK R7994, indicating that the differences (slender to massive) are ontogenetic. It is worth pointing out that the illustrations that were used to show the
differences between the specimens are mislabeled, where the massive NHMUK R7994 is mistakenly labeled as the slender AMNH FARB 5027, and vice versa (*Carpenter, 1990*: Fig. 10.4A and B).

Finally, *Carpenter (1990)* compared the several features of the ischium between three specimens of *T. rex*. He identified the orientation of the ischia as informative, where in two (CM 9380, TMP 1981.006.0001) the bone extends sharply posteroventrally whereas the third (AMNH FARB 5027) extends at a lower angle. *Carpenter (1990)* hypothesized that the steep condition was required for passing eggs in life, and the bearers were female, whereas the bearer of the other form was male. The results of the cladistic analysis here found that of these three specimens, TMP 1981.006.0001 is the least mature and that CM 9380 is the most mature. Therefore, a growth sequence from steep (TMP 1981.006.0001) to less steep (AMNH FARB 5027), and back to steep (CM 9380) is seen. If the angles of divergence are real, then a lack of signal (noise) is probably the best explanation for the observation.

However, the position of the tip of the ischium is an important landmark for this comparison, but it is missing from CM 9380 (*Osborn, 1906*: fig. 7) whereas it is shown as complete in *Carpenter (1990*: fig. 10.5). The complete condition presumably reflects restoration of the bone as indicated by the presence of a small boot at its tip (*Carpenter, 1990*: fig. 10.5), a structure that is not seen in complete specimens of *T. rex* or in tyrannosaurids in general. Also, the shaft of the ischium in CM 9380 is intermediate in position between the two other specimens in that it overlaps the dorsal margin of the ischium of TMP 1981.006.0001 (*Carpenter, 1990*: fig. 10.5). Given these issues, the character was included a posteriori in the cladistic data matrix and only the three specimens mentioned by *Carpenter (1990)* were coded in the analysis, using the state assignments given by him. The analysis recovered one 3,054-step ontogram and the topology is unaffected. Therefore, based on this sample, ischial divergence does not group specimens together, indicating that it is not dimorphic; regardless, a larger data set for this character is required for a rigorous test of whether the two states are valid to begin with.

## Sexual dimorphism and "*Tyrannosaurus "x"*" (sensu *Larson, 2008*)

A topological pattern of sexual dimorphism was not recovered in the ontogram, which is linear and pectinate (Fig. 2). If dimorphism was present in the data, which covers the entire skeleton, and if the sample included examples of each sex, then a branch of females that includes BMRP 2006.4.4 and MOR 1125, the only purportedly unambiguous females in the sample, should be united by the presence of shared skeletal sexual correlates, aside from femoral medullary bone (*Schweitzer, Wittmeyer & Horner, 2005*; *Woodward et al., 2020*). The failure to recover a female branch indicates that such correlates are absent from *T. rex*. Neither did a comparison of individual variation recover a dimorphic pattern (see above). The alternative explanation for these results is that BMRP 2006.4.4 and MOR 1125 are the only females in the sample, but this is improbable given the high number of adult-grade specimens in the analysis.

Regardless, several specimens included in this study have been hypothesized as male (TMP 1981.006.0001, MOR 008, MOR 555) or female (MOR 1125, RSM 2523.8, CM 9380,

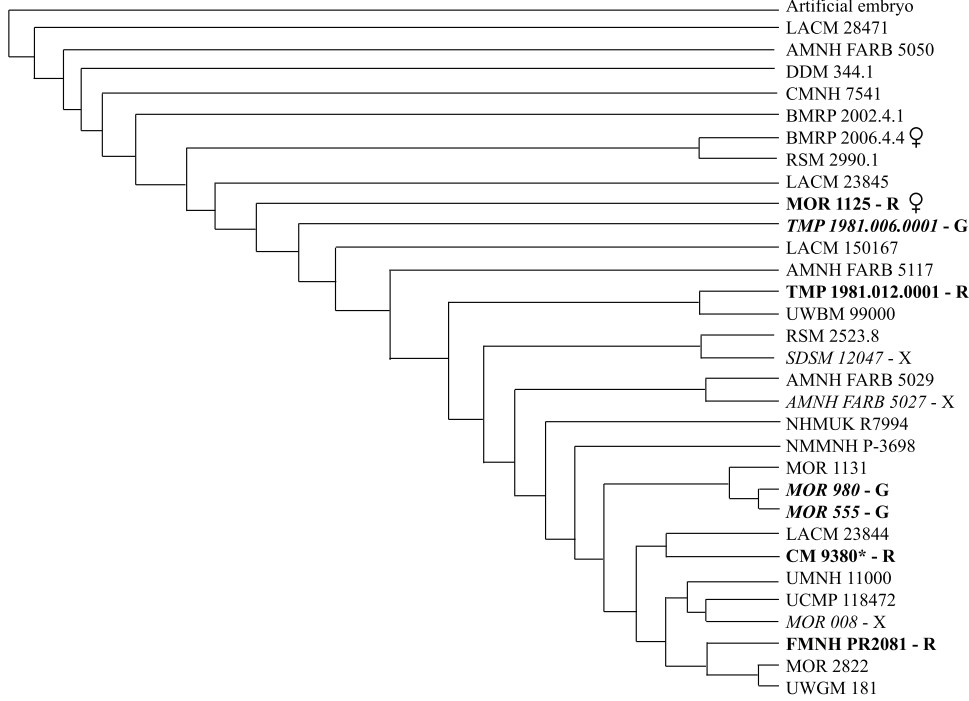

**Figure 21 Sex dimorphs and the taxon "*Tyrannosaurus "x""* of *Larson (2008)* mapped onto the ontogram of *Tyrannosaurus rex*.** A transitional pattern is not seen between gracile and robust morphs; if sexual dimorphism was present, then the "gracile" and "robust" morphs should group along separate branches, which is not seen. Also, specimens referred to the taxon "*T.* "*x*"" do not form a clade, indicating that it is not a valid taxon. The pattern seen here is what is expected for a species without sexual dimorphism. Specimens considered in *Larson (2008)* as gracile are in *boldface italics* with a boldface "G"; specimens considered in *Larson (2008)* as robust are in boldface with a boldface "R"; specimens considered in *Larson (2008)* as referable to "*T.* "*x*"" are in *italics* and marked with an "X".

FMNH PR2081) and so an account of the difference between those results (*Larson, 2008*) and those obtained here is required. No grouping pattern is seen when male ("gracile") and female ("robust") morphs (sensu *Larson, 2008*) are mapped onto the ontogram, where both "morphs" are interspersed among the adult specimens (Fig. 21). A biologically meaningful signal would be indicated if, for example, gracile morphs occurred earlier in growth than the robust morphs, or vice versa. Given that both morphs are seen in young adult and adult growth stages, the simplest explanation is that they do not represent a developmental pattern aside from individual variation upon the attainment of the asymptote of size.

Several lines of evidence were offered to support the inference of a binary difference between male and female morphs (*Larson, 2008*). However, it was found here that those purported dimorphic features actually show a continuum of variation (e.g., circumference to length ratio of the humerus, height to length ratio of the ilium, circumference to length ratio of metatarsal II), whereas others show no pattern (e.g., ratio of third to second dentary teeth, divergence of the ischium, circumference to length ratio of the femur) (Fig. 22).

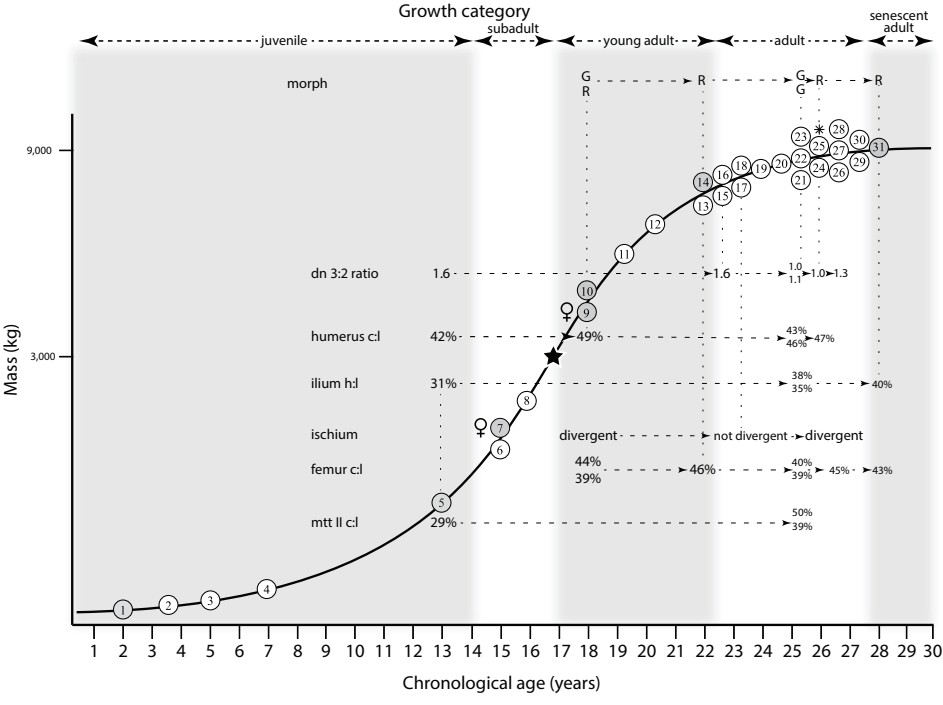

**Figure 22 Sex dimorphs of *Larson (2008)* mapped onto the growth curve of *Tyrannosaurus rex*.** A transitional pattern is not seen between gracile and robust morphs; if sexual dimorphism was present, then the "gracile" and "robust" morphs should grade into each other, which is not seen. Likewise, an ontogenetic progression among the cranial and postcranial indices is not seen. See text for details. Key to specimens numbered on the growth curve is in Fig. 12. 

*Larson (2008)* referred three specimens (AMNH FARB 5027, MOR 008, SDSM 12047) to the taxon "*Tyrannosaurus* "*x*"". If this taxon was valid, then the specimens should be recovered sharing a single branch, separate from the other adults. However, the specimens are scattered throughout the adult growth category of the ontogram, indicating that the characters thought to unite them (size of lacrimal pneumatic foramen, maxillary and dentary tooth count, form of second dentary tooth, ratio of the length of the third to second dentary tooth) actually reflect ontogenetic or individual variation, not taxonomically informative characters (Fig. 22). Finally, a comparison of individual variation (Data S5) was done to identify any shared characters that might provide evidence of common identity. Individual variation was recovered for AMNH FARB 5027 and SDSM 12047, but not for MOR 008. In addition to that, there are no shared characters between the two former specimens; therefore, no character evidence was found that supports the purported taxon.

## Dentary groove

It has been claimed by *Schmerge & Rothschild (2016a)* that the neurovascular groove along the alveolar row of foramina of the dentary in tyrannosaurids has a binary expression of present and absent. Among latest Cretaceous tyrannosaurids of Laramidia, they hypothesized that the groove is absent from *T. rex*, regardless of size, whereas it is present

in "*Nanotyrannus lancensis*", which otherwise has been shown to be an invalid taxon (*Carr & Williamson, 2004*).

Several of the *T. rex* specimens that *Schmerge & Rothschild (2016a*, *2016b)* claimed to lack the dentary groove were re-examined for this study and it was found that in two of them the groove is present, but greatly reduced in extent (AMNH FARB 5027, MOR 008). In addition to that, a reduced groove is seen in all adult *T. rex*, including the type specimen (CM 9380, LACM 23844, MOR 555, MOR 980, MOR 1125, UWBM 99000) (Data S1). These grooves are true sulci, and are not artifacts of foramen shape, and so are not "pseudo-grooves" (sensu *Schmerge & Rothschild, 2016b*).

Although the states of this character were not unambiguously optimized on the ontogram, the specimens show a stepwise progression: the groove is absent from the least mature specimen (LACM 28471), the groove is present along the entire row in more mature juveniles (e.g., BMRP 2002.4.1), and, finally, the groove is reduced to short segments in adults (e.g., FMNH PR2081) and so cannot be said to be absent from them. The groove is almost certainly obscured in adults by the overall expansion of the bone in response to growth, higher loads imposed by ontogenetic increase in body size, tooth size, and bite force (*Bates & Falkingham, 2012*).

## Synthesis of ontogeny and functional morphology

*Tyrannosaurus rex* has been central in recent quantitative studies of the functional morphology of tyrannosaurids and other large theropods (*Henderson, 2002*; *Henderson & Snively, 2004*; *Therrien, Henderson & Ruff, 2005*; *Snively, Henderson & Phillips, 2006*; *Snively et al., 2019*). Those results are re-examined here in the framework of the ontogram to assess the completeness of the hypotheses, provide a specific ontogenetic context for major functional changes, identify the gaps in the data, and propose hypotheses of where in the growth series major morphological changes occurred.

### Orbital fenestra (Henderson, 2002)

Quantifiable correlates of the height of the skull frame include the size, shape, and orientation of the orbital fenestra, the width of the pre- and postorbital bars that frame it, and the stress regime of the skull at the orbit (*Henderson, 2002*). One juvenile (CMNH 7541) and adult (AMNH FARB 5027) were compared by *Henderson (2002*; Fig. 23). Comparison of the results of *Henderson (2002)* with the growth curve reconstructed here (Fig. 23) shows that the details of the transition are incomplete, including the precise stage at which the transformation occurred owing to the absence of suitably complete subadult specimens from the current sample in museum collections. Given the presence of correlates of a tall skull in the exemplars of the subadult category (LACM 23845, RSM 2990.1) it is predicted here that the reduction in orbit size, increase in ellipticity, increase in vertical orientation, increase in pre- and postorbital bar length, and increase in stress regime at the orbit first occurred abruptly in subadults.

### Skull & jaw strength (Snively, Henderson & Phillips, 2006)

A study of the bending strength of the skull and teeth of *T. rex* by *Snively, Henderson & Phillips (2006)* included comparison between one juvenile (BMRP 2002.4.1), two adults

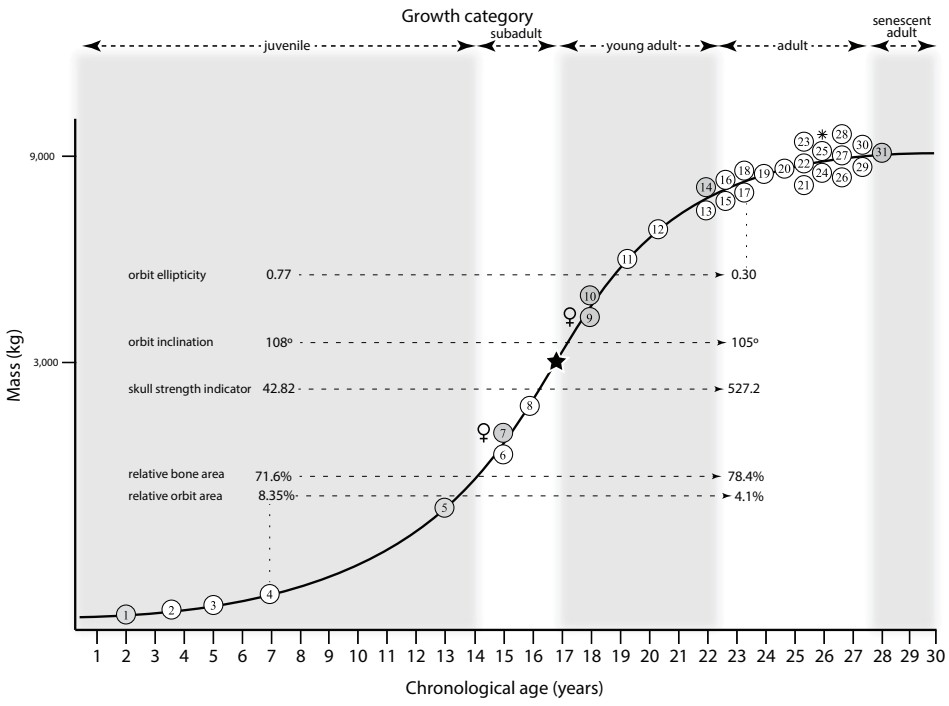

**Figure 23** **The results of** *Henderson (2002)* **mapped onto the growth curve of** *Tyrannosaurus rex.* All measures of the correlates of orbital fenestra size and shape change from juvenile to adult categories. It is predicted here that this transition occurred early in ontogeny, at the subadult growth stage, given the presence of correlates of a tall skull in subadult specimens. Key to specimens numbered on the growth curve is in Fig. 12.           

(AMNH FARB 5027, LACM 23844), and one senescent adult (FMNH PR2081). Their results were mapped onto the growth curve to obtain an ontogenetically constrained hypothesis of the comparisons of vertical bending strength, lateral bending strength, torsional strength, and tooth bending strength (Fig. 24).

*Snively, Henderson & Phillips (2006)* provided a summary hypothesis of the phylogenetic progression of skull strengthening attributes of tyrannosaurids. This phylogenetic progression was compared with the ontogenetic progression for *T. rex* found here to test the hypothesis of congruence between ontogeny and phylogeny. The sequences of changes are congruent, aside from the occurrence of a wide adductor chamber; in *Snively, Henderson & Phillips (2006)* this character is unique to *T. rex*, and is the last character to evolve, whereas it is present in the second growth stage as shown by its presence in CMNH 7541, a juvenile, and all specimens that are more mature than it.

As in the phylogenetic transition, fused nasals are seen earliest, rostrally wide nasals are seen in large juveniles, and the increased nasal cross section, peg-in-socket nasomaxillary suture, and increased tooth strength occur simultaneously in subadults. The simultaneity is almost certainly an artifact of a low sample size, and a higher sample of subadults is required to test the hypothesis that acquisition of those features occur in the same order as is seen in the phylogenetic sequence. All of these features occur in the growth of *T. rex* before the 3,000 kg threshold is reached, indicating that they are not an

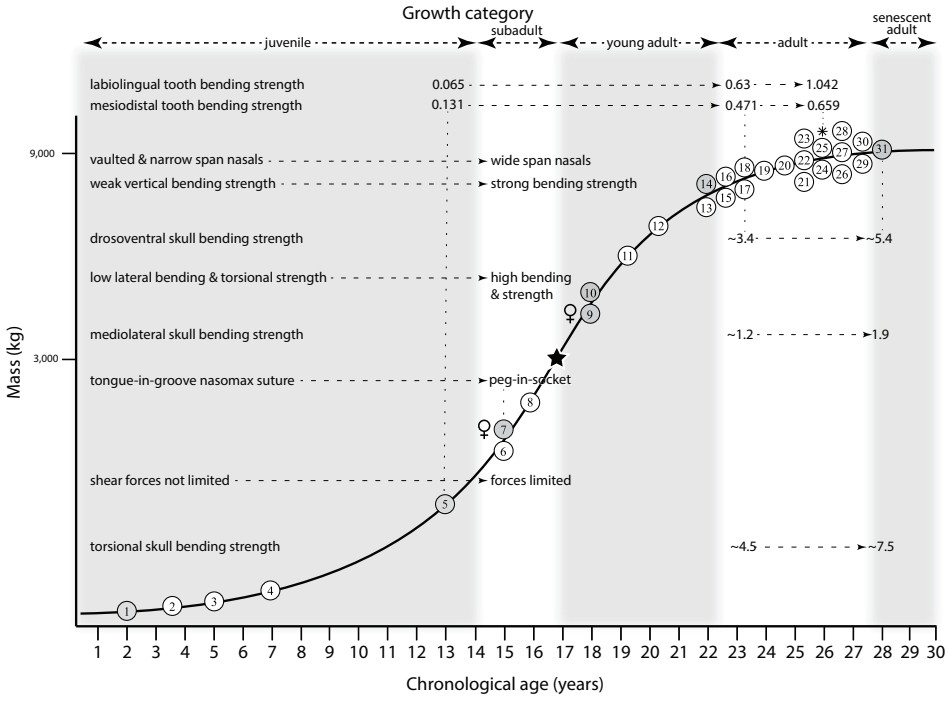

**Figure 24 Skull bending strength mapped onto the growth curve of *Tyrannosaurus rex*.** The results of *Snively, Henderson & Phillips (2006)* showing that the subadult growth stage was an important functional transition point during ontogeny between the long and low skulls of adults and tall and sturdy skulls of more mature animals. Their results show a progression in strength of the skull frame and dentition throughout the adult categories. Key to specimens numbered on the growth curve is in Fig. 12.

artifact of the descendant (i.e., *T. rex*) size range; indeed, these features are seen in other, smaller tyrannosaurines (e.g., *Daspletosaurus*, *T. bataar*, *Zhuchengtyrannus*).

The data for maxillary tooth bending strength are from the juvenile (BMRP 2002.4.1) and adult (AMNH FARB 5027, LACM 23844) stages (*Snively, Henderson & Phillips, 2006*), which corresponds to the 5th, 14th, and 18th growth stages of the ontogram (Figs. 2 and 24). Labiolingual tooth bending strength increases by an order of magnitude between the juvenile and adult growth stages; the juvenile bending strength is comparable to those of juvenile *Daspletosaurus torosus* and *Albertosaurus libratus* (*Snively, Henderson & Phillips, 2006*), which are lower than those of nontyrannosaurids.

Mesiodistal bending strength increases from juvenile to adult, where that of the juvenile is most comparable to narrow-toothed *Allosaurus*, *Monolophosaurus*, and *Albertosaurus libratus* (*Snively, Henderson & Phillips, 2006*). This transition in tooth strength almost certainly occurred between the 5th and 6th growth stages since the 7-shaped lacrimal (RSM 2990.1), regarded here as a correlate of high bites forces, is seen at the 6th growth stage, and wide teeth (LACM 23845) are present no later than growth stage 7. Also, the correlates of high bite forces identified by *Snively, Henderson & Phillips (2006)*, including a peg-in-socket nasomaxillary suture, is present in subadults (LACM 23845); the other correlates, such as a tall maxilla with a wide palatal shelf are unequivocally present by growth stage 8 (MOR 1125) and are predicted here to be present in subadults once
more complete specimens are found. If true, then the teeth, along with the entire craniodental mechanism rapidly (<3 years) transformed as an integrated system well in advance of somatic maturity. The increase in both tooth strength indicators among adults indicates that this trend of increasing strength continued into later growth stages; however, this observation is based on two specimens (AMNH FARB 5027, LACM 23844) and data from additional adults are required to test that hypothesis (*Snively, Henderson & Phillips, 2006*).

Juvenile *T. rex* started life with snouts that are strengthened by fusion of the internasal suture and the vaulted cross section of the nasals (*Snively, Henderson & Phillips, 2006*). However, the dorsoventrally shallow skull and the wide and flat frontal ramus of the nasals produces a low vertical bending strength (*Snively, Henderson & Phillips, 2006*). The tongue-in-groove nasomaxillary joint surface and the narrow span across the nasals are correlates of a low lateral bending and low torsional strength (*Snively, Henderson & Phillips, 2006*).

Based on their observations of adults, *Snively, Henderson & Phillips (2006)* identified two correlates of high torsional strength and lateral bending strength of the skull: a peg-in-socket nasomaxillary suture and a wide span across rostral end of the nasals, which are seen in the subadult category (LACM 23845), and so the transition from a weak snout to a strong snout occurred in that earlier growth stage and was carried forward into adulthood. The change from a low to a tall skull is a central part of the transition from a weak to a strong skull (*Snively, Henderson & Phillips, 2006*); correlates of a tall skull, such as a wide preorbital bar (RSM 2990.1) in the subadult growth stage, shows that the transition from shallow to tall also occurred from the juvenile to the subadult growth categories.

As seen in the maxillary teeth, a trend of increased bending strength of the skull is seen among adults when the results of *Snively, Henderson & Phillips (2006)* are mapped onto the ontogram (Fig. 24). However, only two specimens are sampled (AMNH FARB 5027, FMNH PR2081) and so additional adult specimens, and juveniles, are required to test this hypothesis. It is predicted that the trend of increasing bending strengths occurred throughout ontogeny, with an abrupt increase at the transition between juveniles and subadults.

### Mandibular ramus strength (*Therrien, Henderson & Ruff, 2005*)

*Therrien, Henderson & Ruff (2005)* found that during the ontogeny of *T. rex*, from juvenile to adult, the rostral end of the mandible could resist high torsional loads and, between growth stages, an increase in bite force was seen. Measurements were obtained from the mid-dentary and at the second alveolus (*Therrien, Henderson & Ruff, 2005*). Two sets of measurements were taken, dorsoventral bending strength and relative strength (the ratio of dorsoventral and mediolateral bending strengths). The results of *Therrien, Henderson & Ruff (2005)* were mapped onto the growth curve to test their hypotheses of ontogenetic change in the mandibular bending strengths of *T. rex* (Fig. 25).

In comparison with the growth curve, dorsoventral bending strength at the second alveolus shows a progressive increase from juveniles to young adults to adults, but the

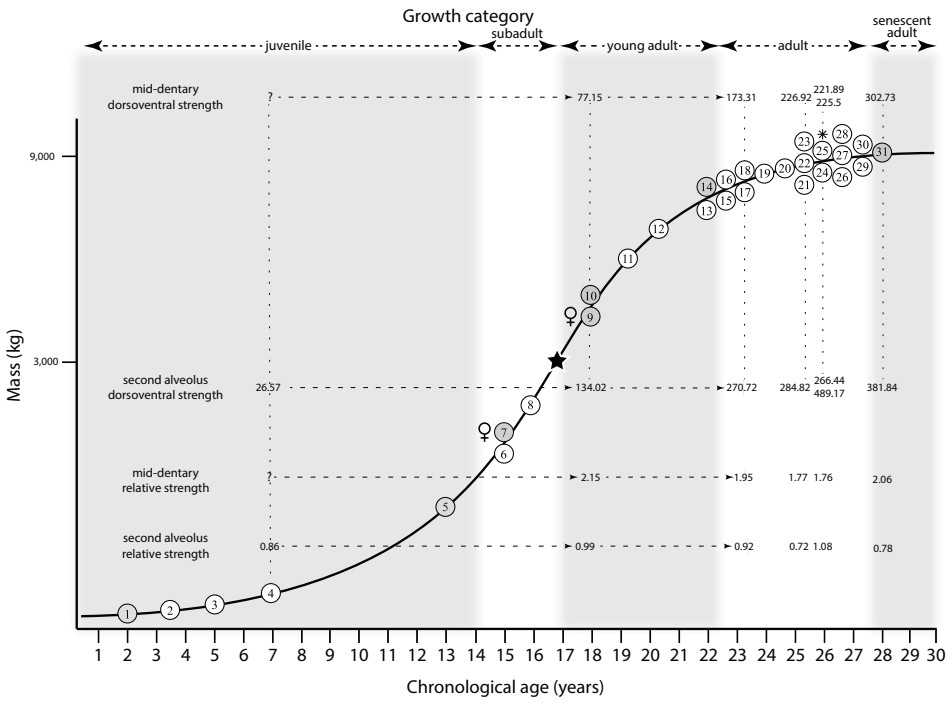

**Figure 25 The results of *Therrien, Henderson & Ruff (2005)* compared with the growth curve of *Tyrannosaurus rex*.** Vertical bending strength and relative bending strength (sensu *Therrien, Henderson & Ruff, 2005*) mapped onto the growth curve of *T. rex*. Values for juveniles are missing for mid-dentary dorsoventral strength and mid-dentary relative strength. In general, strength increases ontogenetically, a trend that becomes obscured in adulthood. Key to specimens numbered on the growth curve is in Fig. 12.           

trend is lost among adults (Fig. 25). This loss of pattern is not unexpected for specimens of the variable adult size. It is predicted here that the strength of subadults will be lower than adults but will found to be closer to the adult values than to those of juveniles.

Although mid-dentary vertical bending strength is not available for juveniles, an increasing trend of strength is seen from the young adult to adult to senescent growth stages; in contrast, a clear pattern of increase is not seen among adults although the highest value is seen in the senescent adult (Fig. 25). It is predicted here that the bending strength of subadults at the mid-dentary will be less than those of the adult growth categories and that of juveniles will be significantly less than subadults.

Relative strength was also estimated at the second alveolus and at the mid-length of the dentary tooth row (*Therrien, Henderson & Ruff, 2005*). Relative strength at the second alveolus increases from the juvenile to young adult growth categories, but a trend is not seen throughout the growth series (Fig. 25). In addition to that, values for some adults are lower (e.g., 0.72) than that for the juvenile growth stage (e.g., 0.86), indicating a high degree of variation. It is predicted here that, if there is a trend, the relative strength of subadults will be intermediate between the values for the juvenile and young adult categories.

At the mid-dentary, relative strength decreases from young adults to adults, but no trend is seen among adults and senescent adults. It is predicted here, based on the trend in

Fig. 25 and the values for small juvenile tyrannosaurids in *Therrien, Henderson & Ruff (2005)*, that the values for subadults and juveniles will be sequentially greater than those of the adult category.

In short, vertical bending strength of the mandibular ramus increases ontogenetically, a trend that becomes obscured at senescence. In contrast, a trend in relative strength has a broadly decreasing trend at mid-dentary, but no clear trend at the second alveolus; that is, more variation is seen in relative strength than in dorsoventral strength. It is predicted that data from the future discovery of subadults will be consistent with the mid-dentary pattern but will not clarify the situation at the second alveolus.

### Summary of craniomandibular strength

Taken together, the results of *Henderson (2002)*, *Snively, Henderson & Phillips (2006)*, and *Therrien, Henderson & Ruff (2005)* mapped onto the growth curve of *T. rex* provides evidence that the ontogenetic transformation of the skull and jaws marks an abrupt change from the juvenile morphotype to a mature morphotype. The skull of juveniles is open framed with a large orbital fenestra and low bone area, which results in low skull strength; also, the bending strength (labiolingually and mesiodistally) of the narrow teeth is low. However, a more powerful bite in juvenile *T. rex* relative to other theropods of the same size is indicated by the presence of a wide adductor region and fused, vaulted nasals (cf. *Snively, Henderson & Phillips, 2006*). The vertical and relative bending strengths of the mandibular ramus in juveniles are low, which is consistent with the low strength of the cranium.

The abrupt transition to the mature morphotype (i.e., a tall skull with increase in relative bone area) is seen in subadults, where the nasals have an increased cross section, the nasomaxillary suture acquires a peg-in-socket form, and tooth strength increases (see above for correlates). Although the dorsoventral and relative bending strengths of the mandibular ramus have not been estimated for subadults, they are predicted here to be closer to the high values of adults than to the low values of juveniles.

In the adult growth categories, the mature morphotype persists largely unchanged aside from continued reduction in orbit ellipticity, inclination, and area; and increase in bone area of the skull, skull strength, maxillary tooth strength, and dorsoventral and relative mandibular strength. Overall, the strength of the craniomandibular skeleton of *T. rex* increases throughout adulthood.

## Ontogenetic change and strain distribution

Two sets of heat maps of the skull in lateral, dorsal, and ventral views were drafted to show, in relatively high-resolution, the distribution and magnitude of growth changes in each bone (Fig. 26A) and, to reflect a lower-resolution pattern, the frequency in each module (Fig. 26B; Table 4). In the diagram, the percentage of changes was doubled so that the lightest shades of gray could be seen (i.e., 10% of total changes was rendered as 20% gray, and so on). In the heat map of individual bones, the greatest number of changes are seen from the orbit to the snout, specifically in the lacrimal, and maxilla, jugal, postorbital, frontal, and the dentary (Fig. 26A). An intermediate number of changes are

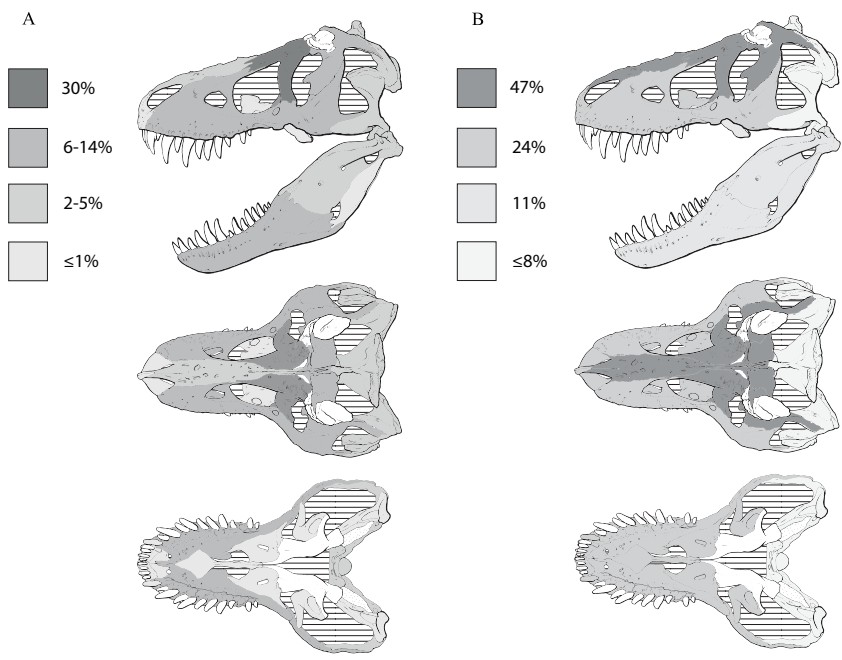

**Figure 26 Heat maps of the ontogenetic changes seen in the skull and mandible of *Tyrannosaurus rex*.** Illustrations show per centage of the total number of unambiguously optimized synontomorphies per bone (A) and functional module (B). Darker shades of gray indicate higher proportions of growth change, whereas lighter shades indicate lower proportions of change. The results show that the greatest amount of growth changes are at the lacrimal (A) or along the dorsal skull roof (B). Hatchure indicates empty space; stipple indicates unprepared matrix. 

seen in the nasal, squamosal, quadratojugal, parietal, and braincase, and in the surangular and prearticular of the mandibular ramus (Fig. 26A). Finally, the fewest changes of the cranium are seen in the palate and quadrate, and the angular of the lower jaw (Fig. 26A).

In the heat map of the modules, the greatest amount of change is seen in the dorsal skull roof, including the nasals, lacrimal, postorbital, and frontals (Fig. 26B). An intermediate number of changes is seen in the snout, including the premaxilla, maxilla, jugal, postorbital, and palate (Fig. 26B). Finally, the fewest changes are seen in the suspensorium, parietal, and braincase, and in the mandibular ramus (Fig. 26B). In both sets of heat maps, the greatest number of changes occur in the circumorbital region whereas the fewest occur in the suspensorium, parietal, braincase, and postdentary moiety of the mandibular ramus.

These distributions were compared to two-dimensional models of strain in the cranium of *T. rex* (*Rayfield, 2004*) to test the hypothesis that the patterns of growth and strain are no different. Two models of the skull (fused, mobile) were loaded vertically (i.e., biting load) and horizontally (i.e., tearing load); compressive, tensile, and shearing stress were compared (*Rayfield, 2004*). In light of recent work showing that the skull of *T. rex* was akinetic (*Cost et al., 2019*), the fused skull model results of *Rayfield (2004)* are discussed here.

During a vertical bite, compressive stress loads the lower half of the skull, especially from the internal antorbital fenestra to the lower temporal bar; tensile stress is highest in
the same region (*Rayfield, 2004*). In contrast, shear stress is highest above the orbital and internal antorbital fenestra and, to a lesser extent, along the interfenestral strut (*Rayfield, 2004*). The ontogenetic heat map for individual bones does not correspond to the compressive or tensile stress distributions, where the region below the orbital- and laterotemporal fenestrae (largely represented by the jugal and quadratojugal) have the lowest proportions of growth changes, with the exception of the high proportion of change seen in the maxilla and lacrimal (Fig. 26A). In contrast, the high proportion of growth changes in the lacrimal and maxilla do reflect the high shear stress seen along the corresponding dorsum of the snout and the interfenestral strut (*Rayfield, 2004*; Fig. 26A).

In comparison with the module-based heat map, the region of lowest strain occurs where the greatest amount of change is seen (nasals, lacrimal, postorbital), and the region of highest strain corresponds to the region of an intermediate amount of growth change (maxilla, jugal; Fig. 26B). However, the regions of maximum shear stress do correspond with the region of the greatest growth change, and one of the two regions of lowest shear stress (at the suspensorium), and of tensile stress, does correspond to where the lowest amount of growth changes are seen (Fig. 26B).

During a horizontal tear, the compressive loadings are similar to that of a bite, except that highest loadings are seen adjacent to the loaded maxillary teeth (*Rayfield, 2004*). Tensile stress is similar to that of a bite, concentrated along the lower margin of the snout and orbitotemporal regions (*Rayfield, 2004*). Shear stress is partly similar to the pattern seen in a bite, where stress is highest at the dorsal ramus of the lacrimal, but it is also high below the postorbital bar (*Rayfield, 2004*). The ontogenetic heat map does not directly correspond to the stress distributions, aside from the high compressive stress on the maxilla and the shear stress seen on the lacrimal (Fig. 26A). Therefore, it appears that skull growth did not track stress loads on the skull, whether imposed by biting or tearing. If skull growth was controlled by stress, then higher proportions of change would be seen in the jugal, postorbital, and suspensorium.

In comparison with the module-based heat map, the pattern of maximum strain and tensile stress, along the subfenestral regions of the maxilla and jugal, do not correspond with the heat map, where the greatest changes are seen along the snout dorsum. In contrast, one of the two regions of maximum shear stress, along the snout dorsum, corresponds to the maximum number of growth changes seen there.

*Cost et al. (2019)* tested hypotheses of palatal kinesis and strain in the skull of *T. rex* using 3-dimensional FEA modeling; the ontogenetic heat maps were compared to their results to test the null hypothesis that the ontogenetic and strain patterns are no different. *Cost et al. (2019)* found that strains, in each of the three loading postures (neutral, rostrocaudal, mediolateral) were concentrated along the rostrodorsal margin of the internal antorbital fenestra, palate (including the quadrate), the maxilla adjacent to the palatine, the preorbital bar, base of the postorbital bar, and the lower temporal bar.

The ontogenetic heat map for individual bones is consistent with those results in that the highest number of growth changes are seen in the maxilla and lacrimal, but it is inconsistent in that the palate, and the postorbital and lower temporal bars have the lowest proportion of growth changes (Fig. 26A). Therefore, growth changes do not match the

strain patterns, indicating that growth does not follow skull loading, aside from the immediate preorbital region, and so a tightly corresponding pattern of remodeling is not seen (Fig. 26A). These results are consistent with what is seen in the stress models of *Rayfield (2004)*.

In comparison with the module-based heat map, the pattern in lateral view differs, where the regions of maximum strain (subfenestral regions of maxilla and the jugal, and the quadrate) in the model do not correspond to those of the maximum amount of growth changes (snout dorsum). In ventral view, correspondence is seen where the highest strain occurs in the palate and snout, whereas the lowest amount of strain is seen in the braincase (Fig. 26B). Prima facie, the general noncongruence between growth change distribution and strain distribution indicates that ontogenetic changes were not limited by the requirements for architectural stability and force transmission.

## Cephalic musculature lever arm and extension (*Molnar, 2013*)

*Molnar (2013)* graphed the lever arm and extension of each adductor muscle and the mandibular depressor for juvenile (CMNH 7541) and adult (AMNH FARB 5027) *T. rex*, and adult *D. torosus* (CMN 8506). The anterior pterygoid muscle is excluded from this comparison, as it has been shown by *Witmer (1997)* that the antorbital fossa was apposed to a paranasal sinus, and not the origin for that muscle.

A marked transition in the lever arm is seen from juvenile to adult in several adductors. When the mouth is closed, the lever arm increases in the posterior mandibular adductor, superficial and medial external mandibular adductor, and the mandibular depressor (*Molnar, 2013*). In contrast, that of the deep external mandibular adductor and the pseudotemporal stays the same and that of the dorsal pterygoid decreases (*Molnar, 2013*). In *D. torosus*, the lever arms are approximately intermediate between juvenile and adult for the posterior mandibular adductor, and the superficial and medial external mandibular adductor (*Molnar, 2013*).

When the mouth is opened, the lever arm increases from juvenile to adult in the posterior mandibular adductor, superficial and medial external mandibular adductor, and the dorsal pterygoid, whereas that of the deep external mandibular adductor, pseudotemporal, and mandibular depressor stays the same (*Molnar, 2013*). In *D. torosus*, the lever arms are intermediate between juvenile and adult for the superficial and medial external mandibular adductor (*Molnar, 2013*). If the ontogenetic progression is congruent with the phylogenetic progression, it is predicted that the lever arms in subadult *T. rex* will match what is seen in *D. torosus* for the posterior mandibular adductor, and the superficial and medial external mandibular adductor when the mouth is closed, and the superficial and medial external mandibular adductor when the mouth is agape. The relative magnitude of the lever arm for the muscles is the same in all three exemplars (*Molnar, 2013*).

A transition in percentage extension is seen from juvenile to adult *T. rex*, where an increase occurs in the posterior mandibular adductor, superficial, medial external mandibular adductor, and mandibular depressor (*Molnar, 2013*). In contrast, a decrease is seen in the pseudotemporal and the deep external mandibular adductor (*Molnar, 2013*).

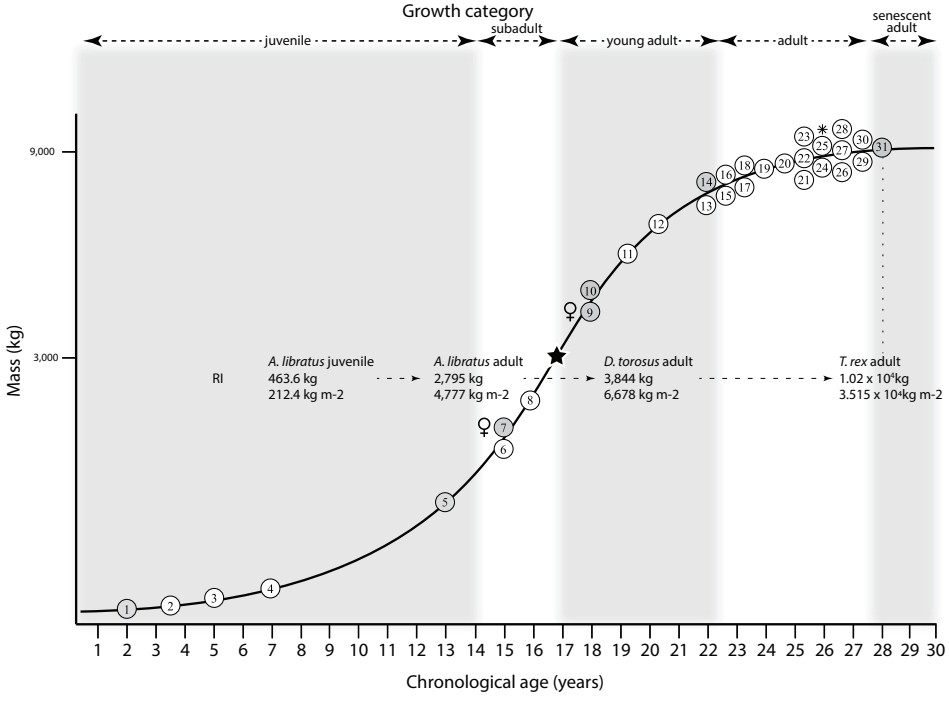

**Figure 27 Comparison of the results of Henderson & Snively (2004) with the growth curve of *Tyrannosaurus rex*.** The rotational inertia (RI) of smaller and progressively distant sister taxa of *T. rex* serve as predictive proxies for the RIs of young adult and juvenile *T. rex*. Given the larger size of *T. rex* in contrast to non-tyrannosaurine tyrannosaurids, the RI of young adult *T. rex* will almost certainly be more comparable to that of adult *D. torosus* than to adult *A. libratus*. The positions of the taxa, aside from *T. rex* (FMNH PR2081), are relative and are not intended to correspond to exact locations along the growth curve. Key to specimens numbered on the growth curve is in Fig. 12.

The percentage extension in *D. torosus* is intermediate for the posterior mandibular adductor and the superficial and medial external mandibular adductor (*Molnar, 2013*); assuming that the ontogenetic progression will map onto the phylogenetic transition, the expected percentage in subadult *T. rex* is predicted to be the same as in *D. torosus*. The percentage extension in *D. torosus* is lower than in juvenile and adult *T. rex* for the pseudotemporal, deep external mandibular adductor, and the mandibular depressor, indicating that the high extension of those muscles is autapomorphic in *T. rex* (*Molnar, 2013*). It is predicted here that the value in subadults will be intermediate between the values for juvenile and adult *T. rex*, but it will be closer to adults than to juveniles.

## Rotational inertia (*Henderson & Snively, 2004*)

*Henderson & Snively (2004)* quantified the rotational inertia (RI) of several tyrannosaurids, including a juvenile and adult of *A. libratus*, an adult of *D. torosus*, and an adult *T. rex*. Based on mass, it is predicted here that (1) the RI of juvenile *T. rex* (less than 500 kg) will approximate that of juvenile *A. libratus*, (2) *T. rex* in the transition between subadult and adult (~2,800 kg) will be similar to that of adult *A. libratus*, and (3) *T. rex* in the young adult stage (~3,800 kg; this mass has since been revised to 3,085 kg in *Snively et al., 2019*) will be comparable to that of adult *D. torosus*; ergo, the ontogenetic and

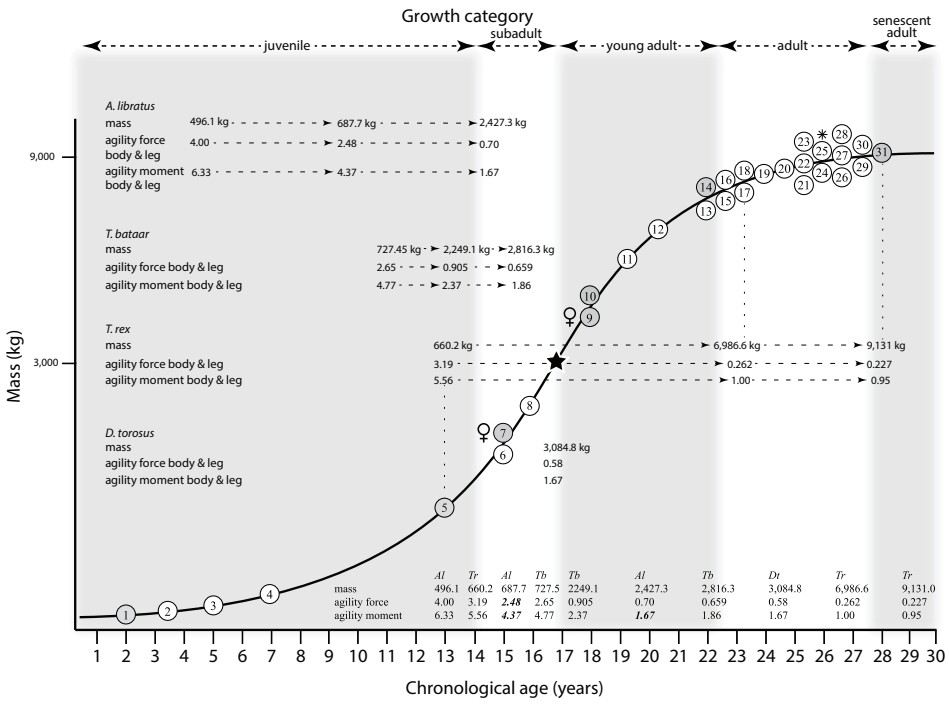

**Figure 28 Comparison of the results of *Snively et al. (2019)* with the growth curve of *Tyrannosaurus rex.*** A comparison of the agility values between *T. rex* and other tyrannosaurids. The values for *T. bataar* and *D. torosus* serve as predictive proxies for the corresponding values in subadult and young adult *T. rex.* The values for *T. rex* are calibrated to the growth series, but those of the other taxa are positioned relative to the values seen in *T. rex.* Inset of the data in table form shows the trends in the data; low values for *Albertosaurus libratus* are in boldface italics. Key to specimens numbered on the growth curve is in Fig. 12. *Al, Albertosaurus libratus*; *Dt, Daspletosaurus torosus*; *Tb, Tyrannosaurus bataar*; *Tr, Tyrannosaurus rex.*

phylogenetic trend of tyrannosaurid RI will predict the ontogenetic RI trend of *T. rex* (Fig. 27).

## Agility (*Snively et al., 2019*)

In all tyrannosaurids, including *T. rex*, agility decreases from the juvenile to adult growth stage (*Snively et al., 2019*), a trend that continues into senescent adulthood (Fig. 28); however, the adult sample for *T. rex* includes only two specimens (AMNH FARB 5027, FMNH PR2081), so the trend may disappear with additional fossils. Using the mass of other tyrannosaurids as a guide, the transition from high to low agility between the tyrannosaurines is continuous, whereas the values for *A. libratus* are slightly lower, but are consistent with the overall trend in reduction (*Snively et al., 2019*; Fig. 28). Since the extreme growth changes in *T. rex* are concentrated in the head, there is no reason to expect an abrupt growth-related decrease of agility in the transition from juvenile to subadult.

## Suture morphology

*Herring (1993)* stated that, in vertebrates, there "seem to be no cases of beveled sutures becoming butt-ended (sensu *Moss, 1957*)". However, in *T. rex* this extreme transformation is seen at the frontonasal and surangulodentary contacts. The joint surface for the nasal

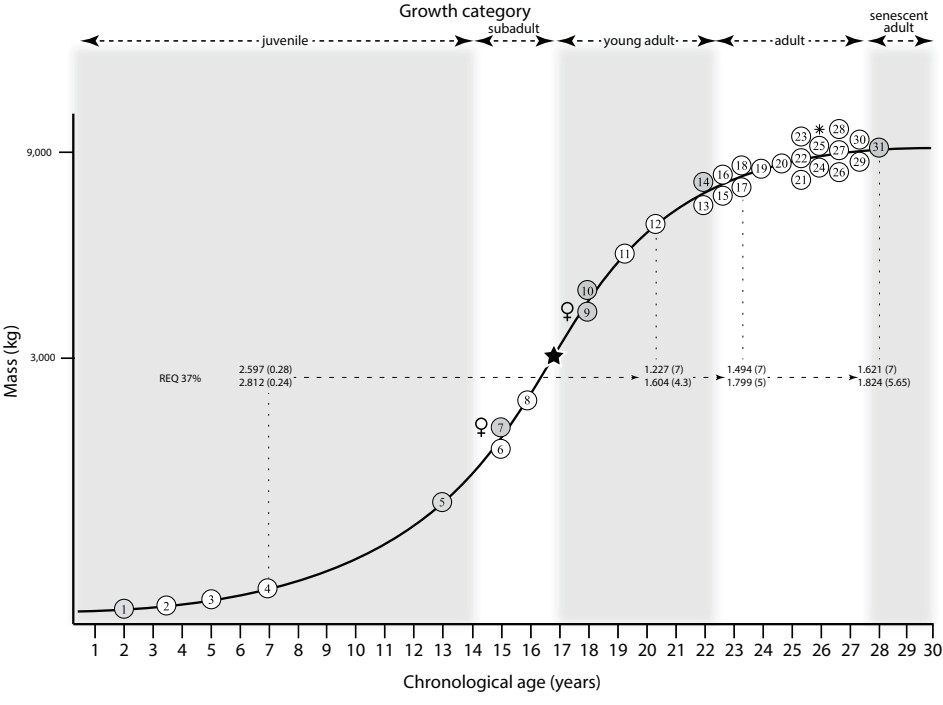

**Figure 29 Reptile Encephalization Quotients (REQs) of *Hurlburt, Ridgley & Witmer (2013)* mapped onto the growth curve of *Tyrannosaurus rex.*** The REQ is based on a brain mass to endocranial volume ratio of 37% and the parenthetical values following the REQs corresponds to the two different body mass estimates, in metric tonnes, from which the REQs were derived (see *Hurlburt, Ridgley & Witmer, 2013* for details). Overall, the REQ of the juvenile greatly exceeds that of adults, and the adults show an increasing ontogenetic progression of REQ values, as first reported by *Hurlburt, Ridgley & Witmer (2013)*. Key to specimens numbered on the growth curve is in Fig. 12. 

on the frontals in juveniles (e.g., DDM 344.1) is a low, inclined facet, whereas in adults (e.g., MOR 2822) the joint surface for the nasal is deeply incised and slot-like. The surangulodentary joint surface is a narrow, low angled and shingle-like overlapping contact in juveniles (e.g., BMRP 2002.4.1), whereas in adults (e.g., MOR 980), the contact is a wide, coarse, and flattened contact at which both bones are abruptly mediolaterally expanded.

## Suture closure in the braincase

Although no braincase suture is unambiguously optimized as a synontomorphy, they can be visualized using the Trace function in MacClade (*Maddison & Maddison, 2005*). The otoccipitobasioccipital suture of the occipital condyle closes first, no later than the start of the young adult category. This is followed by the laterosphenoidoparietal suture and the otoccipitobasioccipital suture of the subcondylar region, which close by the end of the young adult category. The laterosphenoidoprootic suture is closed by the onset of the young adult category, but there is no data for less mature specimens. Finally, neither the laterosphenoidoorbitosphenoid suture or the parietootoccipital suture shows a hierarchical pattern; both tend to remain open.

## Encephalization quotients

The results of *Hurlburt, Ridgley & Witmer (2013)* were mapped onto the growth curve (Fig. 29), which show a high Reptile Encephalization Quotient (REQ; brain mass to endocast volume ratio of 37%) in a juvenile (CMNH 7541) that exceeds those of adults. This growth pattern is also seen in *Alligator mississippiensis* (*Hurlburt, Ridgley & Witmer, 2013*). Although adult REQs are significantly lower than what is seen in the juvenile, nonetheless an increasing progression in REQ is seen among the three specimens as first hypothesized, based on body size, by *Hurlburt, Ridgley & Witmer (2013)*. It is naïvely predicted here that the reduction in REQ, from the juvenile value to the adult range, will coincide with the transition from juvenile to subadult growth stage as a part of the complex of changes that reshapes the skull from a shallow to a deep morphotype along with corresponding endocranial alteration.

## Congruence between ontogeny and phylogeny

Comparison of the frequency of phylogenetic synontomorphies (ontogenetic characters that are homologous with phylogenetic characters that have been optimized as supporting a node) and nonphylogenetic changes (ontogenetic characters that do not have a phylogenetic homolog) shows that in all cases but one, the nonphylogenetic changes exceed those of phylogenetic changes (Fig. 5). Overall they share the same peaks, except for a peak at growth stage 4 in the phylogenetic changes (Fig. 5). The corresponding 214-character region of the data matrix shows that relatively immature specimens are coded with the plesiomorphic character states, whereas relatively mature specimens are coded with apomorphic character states. It is therefore reasonable to expect that the ontogenetic character transformations would be congruent, *en masse*, with their phylogenetic counterparts.

However, only 10 of the 329 unambiguously optimized synontomorphies are homologous with the phylogenetic synapomorphies, and a limited congruence with the phylogenetic pattern recovered in *Carr et al. (2017)* is seen (Fig. 30; Table 19). A Shapiro–Wilk test found that the ranked growth data and clade rank data are normally distributed ($p = 0.690$ and $0.767$, respectively). A Spearman-rank correlation test results in a nonsignificant ($p = 0.693$) correlation coefficient ($r_S = -0.143$; Table 19; Fig. 31) between growth stage rank and clade rank. The phylogenetic context of the conflicting data are illustrated in Fig. 30. It is possible that this result is an artifact of the relatively poorly sampled juvenile and subadult growth categories, which indicates that the discovery of new specimens in those categories of maturity will simultaneously increase the frequency and congruence of phylogenetically homologous character changes.

## Individual variation

By growth category, the frequency of individual variation in *T. rex* is, in descending order: adult (151), young adult (98), juvenile (22), senescent adult (19), subadult (6). The low amount of variation at the subadult category almost certainly results from the fact that the growth stage is represented by three incomplete specimens (BMRP 2006.4.4, LACM 23845, RSM 2990.1). Nearly twice as much variation is seen in adults than is seen in young

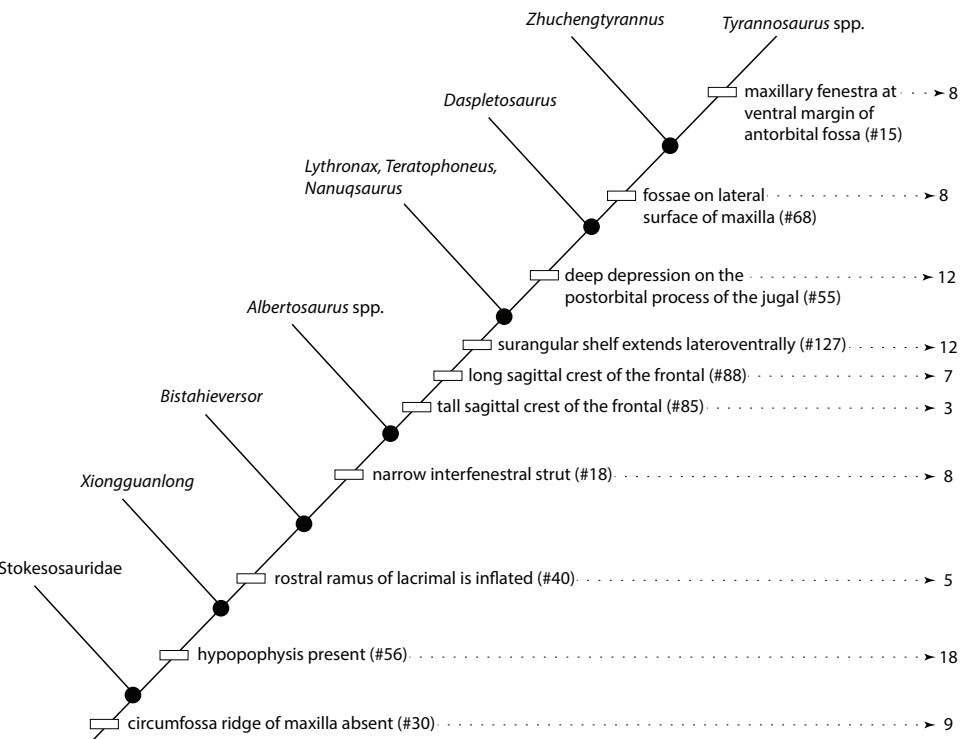

**Figure 30 Comparison of recapitulatory synontomorphies of *Tyrannosaurus rex* with tyrannosauroid phylogeny.** Ten unambiguously optimized synontomorphies are congruent with unambiguously optimized synapomorphies of tyrannosauroid phylogeny, providing limited evidence of recapitulation (see text for details). Numbers to the right correspond to the growth stages in Fig. 2. If recapitulation was present, then the growth stage numbers should increase with progressively exclusive clades; that pattern is not seen here.

**Table 19 Data used in the correlation test between ontogeny and phylogeny for *Tyrannosaurus rex*.** Phylogenetic topology of *Carr et al. (2017)* was followed for clade ranks. Boldfaced column gives the means for tied ranks (i.e., midranks). The category "derived tyrannosaurines" (i.e., *Lythronax, Nanuqsaurus, Teratophoneus*) occurs three times given the presence of three ontogenetic characters recovered in the ontogram that are homologous with corresponding synapomorphies. The 12th growth stage occurs twice given that two synontomorphies occr at that growth stage but their homologs occur at different internodes on the phylogenetic hierarchy. Boldface indicates the ranks used in the correlation test.

| Clade | Clade rank | Clade midranks | Growth stage | Growth rank | Growth midranks |
|---|---|---|---|---|---|
| "Stokesosaurids" + derived tyrannosauroids | 1 | **1** | 9 | 7 | **7** |
| *Xiongguanlong* + derived tyrannosauroids | 2 | **2** | 18 | 10 | **10** |
| *Bistahieversor* + derived tyrannosauroids | 3 | **3** | 5 | 2 | **2** |
| Tyrannosauridae | 4 | **4** | 8 | 4 | **5** |
| derived tyrannosaurines | 5 | **6** | 3 | 1 | **1** |
| derived tyrannosaurines | 6 | **6** | 7 | 3 | **3** |
| derived tyrannosaurines | 7 | **6** | 12 | 8 | **8.5** |
| *Daspletosaurus* + advanced tyrannosaurines | 8 | **8** | 12 | 9 | **8.5** |
| *Zhuchengtyrannus* + *Tyrannosaurus* | 9 | **9** | 8 | 5 | **5** |
| *Tyrannosaurus* | 10 | **10** | 8 | 6 | **5** |

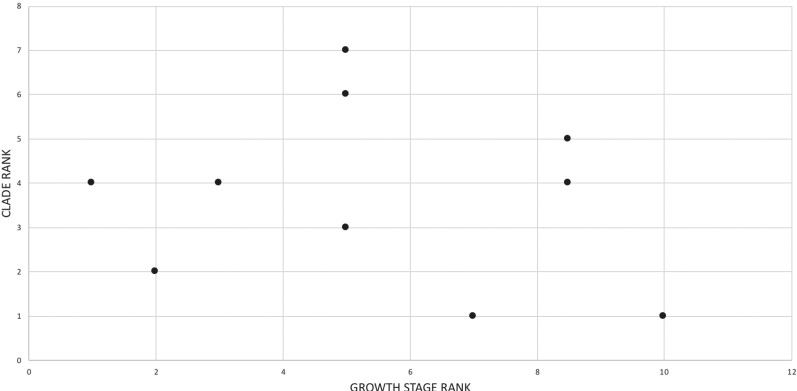

**Figure 31 Bivariate scatterplot showing the test of ontogenetic recapitulation of phylogenetic novelties in *Tyrannosaurus rex*.** Growth stage rank (increases away from the origin) is along the *x*-axis; clade rank (increases away from the origin) is along the *y*-axis. If recapitulation is present, then the ranks will increase montonically from the origin. A recapitulatory pattern is not seen in *T. rex*; see text for details.

adults; given the completeness of specimens in these growth stages, this difference almost certainly approximates the real pattern. In nearly every skull module, the number of changes increases with ascending ontogenetic categories (Table 4), which is evidence of extensive remodeling throughout the phases of life that correspond to sexual maturity.

In comparison with the frequency of synontomorphies at each node, with two exceptions, individual variation outnumbers synontomorphies from the young adult category forward (Fig. 2). The correlation between the individual variation on each branch and the number of synontomorphies at each node was assessed under two correlation tests (Table 20). The raw data are skewed to right and were run under a Pearson correlation test that obtained a nonsignificant ($p = 0.559$) correlation ($r = -0.109$). The skewness was reduced upon converting the data to ranks (Table 20), and a Shapiro–Wilk test found that the ranked data are not normally distributed for the individual variation ranks ($p = 0.015$), whereas the data for the synontomorphy ranks are normally distributed ($p = 0.144$). A Spearman rank correlation test was run, which returned a nonsignificant ($p = 0.695$) coefficient ($r_S = -0.073$) (Fig. 32; Table 20). Therefore, the number of synontomorphies does not influence the amount of individual variation. A lower-resolution Spearman rank correlation test was done, at the level of growth category (Table 4), which also resulted in a nonsignificant ($p = 0.397$) correlation coefficient ($r_S = 0.429$), showing there is no correlation at that low level of comparison.

In terms of craniomandibular modules, the frequency of individual variation, in descending order, is: dorsum of snout (100), sides of snout (57), mandibular ramus (42), braincase (29), suspensorium (23), and parietal (23) (Table 4; Data S5). As such, no region of the skull is exempt from individual variation. A second correlation test was run to compare the amount of synontomorphies with individual variation in each module (Table 21); Shapiro–Wilk normality tests found that the synontomorphy data and the individual variation data are normally distributed ($p = 0.961$, $0.537$, respectively). The correlation test resulted in a significant ($p = 0.008$) correlation coefficient ($r_S = 0.928$).
Table 20 Summary of data used in the Spearman correlation test of individual variation per specimen per node and number of synontomorphies per node for *Tyrannnosaurus rex*. The rows are organized by node, followed by the sequence of number of synontomorphies and individual variation by number, rank, and midranks. Boldfaced columns indicate values used in the correlation test.

| Node | # Synontomoprhies/node | Synontomorphy rank | Synontomorphy midrank | Individual variation/branch | Individual variation rank | Individual variation midrank |
|---|---|---|---|---|---|---|
| 1 | 0 | 1 | **1.5** | 0 | 1 | **4.5** |
| 2 | 2 | 4 | **5.5** | 0 | 2 | **4.5** |
| 3 | 0 | 2 | **1.5** | 3 | 11 | **12** |
| 4 | 3 | 8 | **8.5** | 3 | 12 | **12** |
| 5 | 50 | 29 | **29** | 16 | 26 | **26** |
| 6 | 90 | 30 | **30.5** | 0 | 3 | **4.5** |
| 6 | 90 | 31 | **30.5** | 0 | 4 | **4.5** |
| 7 | 16 | 23 | **23.5** | 6 | 18 | **18** |
| 8 | 31 | 28 | **28** | 49 | 31 | **31** |
| 9 | 16 | 24 | **23.5** | 6 | 18 | **18** |
| 10 | 12 | 19 | **19** | 42 | 29 | **29** |
| 11 | 9 | 14 | **15** | 17 | 27 | **27** |
| 12 | 9 | 15 | **15** | 2 | 10 | **10** |
| 12 | 9 | 16 | **15** | 0 | 5 | **4.5** |
| 13 | 20 | 26 | **26.5** | 5 | 15 | **15.5** |
| 13 | 20 | 27 | **26.5** | 9 | 21 | **21** |
| 14 | 10 | 17 | **17.5** | 10 | 22 | **22** |
| 14 | 10 | 18 | **17.5** | 5 | 16 | **15.5** |
| 15 | 3 | 9 | **8.5** | 3 | 13 | **12** |
| 16 | 1 | 3 | **3** | 4 | 14 | **14** |
| 17 | 2 | 5 | **5.5** | 15 | 25 | **25** |
| 17 | 2 | 6 | **5.5** | 41 | 28 | **28** |
| 17 | 2 | 7 | **5.5** | 47 | 30 | **30** |
| 18 | 6 | 10 | **10.5** | 12 | 24 | **24** |
| 18 | 6 | 11 | **10.5** | 8 | 20 | **20** |
| 19 | 15 | 20 | **21** | 0 | 6 | **4.5** |
| 19 | 15 | 21 | **21** | 0 | 7 | **4.5** |
| 19 | 15 | 22 | **21** | 1 | 9 | **9** |
| 20 | 7 | 12 | **12.5** | 11 | 23 | **23** |
| 20 | 7 | 13 | **12.5** | 6 | 18 | **18** |
| 21 | 19 | 25 | **25** | 0 | 8 | **4.5** |

The similar amount of individual variation and number of synontomorphies in each skull module suggests that the number of unambiguously optimized characters are dependent upon the proportion of characters that is scored for each module in the data matrix. Indeed, a tally from the character list (Data S1) shows the frequency of characters per module, in descending order: snout and palate (399), skull roof (334), mandible (141), suspensorium (140), braincase (85), and parietal (25). The distribution broadly

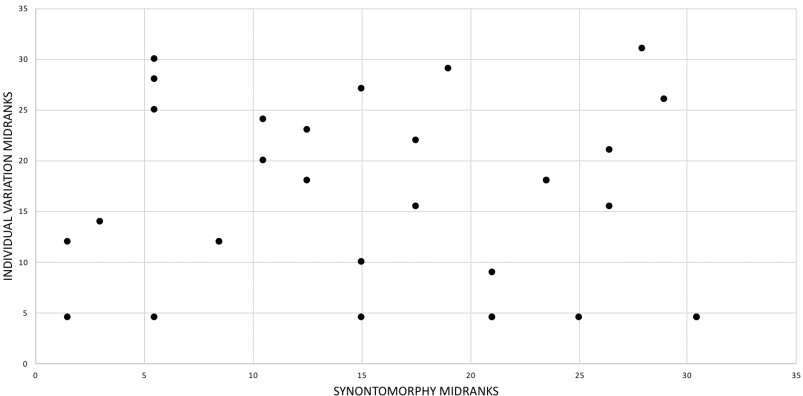

**Figure 32 Bivariate scatterplot showing the congruence between individual variation per specimen per node compared with the number of unambiguously optimized synontomorphies per node in *Tyrannosaurus rex*.** The number of synontomorphies per node are along the *x*-axis; the amount of individual variation (i.e., unambiguously optimized character states per branch) is along the *y*-axis. Both values increase away from the origin. If the amount of individual variation is controlled by the number of synontomorphies per node, then the variables should increase monotonically. In this case, no congruence is seen between the variables in *T. rex*.

**Table 21 Summary of the number of synontomorphies and individual variation per craniomandibular module.** Raw and corrected data used in the correlation test between the number of synontomorphies in each skull module with the amount of individual variation per module. Boldface indicates the ranks used in the correlation test.

| Module | Synontomorphies | Synontomorphies rank | Individual variation | Individual variation rank | Individual variation midrank |
|---|---|---|---|---|---|
| Skull roof | 117 | **1** | 100 | 1 | **1** |
| Snout & palate | 60 | **2** | 57 | 2 | **2** |
| Parietal | 12 | **6** | 23 | 5 | **5.5** |
| Suspensorium | 13 | **5** | 23 | 6 | **5.5** |
| Braincase | 25 | **4** | 29 | 3 | **3** |
| Mandible | 27 | **3** | 27 | 4 | **4** |

matches that of the individual variation and synontomorphies, except for the reversal in order of the snout and skull roof modules, and the suspensorium and braincase modules.

Among all the characters that are optimized unambiguously as individual variation, the number of reversals (from a mature to an immature character state) is close to the number of progressions (from an immature state to the mature condition), 165–175, respectively (Fig. 2; Data S5). Progressions are the only type of change in the first seven growth stages; after that, progressions outnumber reversals until the 11th (exemplar: AMNH FARB 5117) whereupon reversals establish their numerically dominant and constant presence (Fig. 2; Data S5).

Only eight of the 165 reversals pertain to characters of display features, which include the presence of the ridge around the external antorbital fossa of the maxilla, the texture of the dorsal ramus of the lacrimal, and the height and size of the cornual process of the

postorbital (Data S5). The low frequency of variation in cephalic ornaments indicates that these presumed intraspecific display structures are not any more labile than nonornamental characteristics.

In summary, individual variation is seen across the entire skeleton, although it is more frequent in the skull and jaws (274) than in the postcranium (26). Almost certainly this discrepancy is, in part, an artifact of the lower number of characters scored for postcranial bones. Individual variation is not limited to any single domain; for example, in LACM 150167, a specimen with 33 optimizations, cranial variation includes pneumatization, bone shape, muscle scar orientation, alveolus size, foramen magnum size, skull frame, neurovascular foramen position, and alveolar skirt presence, to name a few. In the same specimen, postcranial variation includes, among others, the form of the scapular shaft, prominence of a scapular muscle scar, size of the cuppedicus fossa, form of the ischium, form of the distal joint surface on the fibula, and the orientation of the dorsal margin of the medial fossa of the fibula. As such, individual variation has an unlimited presence across the entire skeleton, which limits the precision of estimating relative maturity without a comprehensive character analysis such as the one presented here.

## Oversplit characters

A review of the synontomophies provides the opportunity for identifying characters that are oversplit (Data S4; Table 22). An oversplit character is defined here as two or more synontomorphies that support the same node and are subsets of the same morphological structure; for example, if the glenoid fossa of the scapula and coracoid are reoriented at the same node, then it is simplest to consider them to be the same change instead of two discrete changes. In contrast, if the glenoid fossa of the scapula changes orientation at an early node, whereas the glenoid fossa of the coracoid changes at a later node, then it is reasonable to treat them as independent characters.

In growth stage 5, several characters might be oversplit, including several pertaining to skull height, the antorbital fossa, dorsal margin of the postorbital, facial subcutaneous texture, and maxillary tooth count (Table 22). In growth stage 6, characters of inflation of the dorsal ramus of the lacrimal and skull frame might be oversplit (Table 22). In stage 7, oversplit characters pertain to inflation of the dorsal ramus of the lacrimal (Table 22). In stage 9, oversplit characters might pertain to alveolus size (Table 22). The test of these hypotheses is the addition of new specimens to the analysis that are sufficiently complete and of the appropriate growth stage. For instance, the great morphological differences between growth stages 4 and 5, and 5 and 6, indicate that specimens of intermediate morphology have not yet been found, and so the numerous changes seen in growth stages 5 and 6 that appear to be oversplit may have not developed simultaneously.

Regardless, the effect of collapsing hypothetically oversplit characters into a single transformation series was tested by running a second analysis. The transformation series were compared in three steps: (1) side-by-side comparison of possibly homologous transformation series, (2) matching transformation series were grouped together, (3) exemplar transformation series (i.e., those scored for the maximum number of specimens) were identified, and (4) the analysis was run with only the exemplar

**Table 22 Comparison of hypothetically oversplit characters in the data set for *Tyrannosaurus rex* ontogeny with decisions for deletion of redundant transformation series.** The rows are organized by growth stage, followed by the skull region, identification of oversplit characters, and the decisions (based on redundancy) to retain (=exemplar) or exclude transformation series from the analysis. Identical transformation series have matching codings, although one might be more complete than the other. Groups of identical transformation series are separated by semicolons. Exemplar transformation series are the most completely coded examples of their sets and were run in the analysis, whereas excluded transformations series were not.

| Growth stage | Domain | Hypothetically oversplit characters | Identical transformation series | Exemplar transformation series | Excluded transformation series |
|---|---|---|---|---|---|
| 5 | Skull height | 4, 57, 66, 238, 239, 240, 254, 260, 570 | 4, 66; 570; 57, 238, 239, 240; 260 | 4, 240, 260, 570 | 66; 57, 238, 239 |
| 5 | Antorbital fossa | 374, 375 | n/a | n/a | n/a |
| 5 | Dorsal margin of postorbital | 620, 623 | n/a | n/a | n/a |
| 5 | Subcutaneous surface | 556, 621 | 556, 621 | 621 | 556 |
| 5 | Maxillary tooth count | 132, 397, 1176 | n/a | n/a | n/a |
| 6 | Lacrimal inflation | 38, 40, 41, 45, 46, 436, 437, 447, 450, 453, 454, 459, 461, 462, 464, 465, 468, 471, 482, 485 | 40, 45, 436, 447, 454, 462, 465, 468; 46; 437, 450; 464; 453; 471; 459; 41, 461; 39; 38, 482, 485 | 447; 46; 450; 464; 453; 471; 459; 41; 39; 38 | 40, 45, 436, 454, 462, 465, 468; 437; 461; 482, 485 |
| 6 | Skull frame | 517, 522, 626 | 517, 626; 522 | 517; 522 | 626 |
| 7 | Lacrimal inflation | 39, 41, 43, 453 | n/a | n/a | n/a |
| 9 | Alveolus size | 117, 121 | n/a | n/a | n/a |

transformation series whereas the redundant transformation series were excluded from the analysis (Table 22). In cases where the transformation series are different, and so clearly test different morphologies or different aspects of the same structure, the characters were regarded as independent. Based on differences between transformation series, the characters pertaining to the antorbital fossa, dorsal margin of the postorbital, maxillary tooth count, lacrimal inflation, and alveolus size were not regarded as oversplit (Table 22).

The analysis recovered 20,000 MPTs (the limit of memory capacity); a strict consensus ontogram shows several polytomies among juveniles (AMNH FARB 5050, CMNH 7541, DDM 344.1, LACM 28471), subadults (BMRP 2006.6.4, LACM 23845, RSM 2990.1), and adults distal to MOR 1125. Within the adult polytomy, one sister pair was recovered (MOR 555 + MOR 980). These regions of the ontogram have low Bremer (1–4), but relatively high bootstrap (62–97), and jackknife (61–97) values, indicating that the conflicting results are almost certainly an artifact character removal that decreased the proportion of topologically informative data.

## DISCUSSION

### Diagnosis of *Tyrannosaurus rex*

The ontogenetic position of CM 9380 among the most mature specimens in the sample (Figs. 1 and 2) indicates that the taxon, *T. rex*, is indeed based upon a mature adult and, by

extension, the type is a defensible name bearer, foundation for diagnosis, and point of comparison for other specimens. The results here support one of the diagnostic features of *T. rex* first identified by *Osborn (1905)*, namely gigantic size. The other characters are not diagnostic; *T. rex* did not have a long humerus and the absence of armor plates is symplesiomorphic for Tyrannosauridae, if not Ornthodira. *Osborn (1905)* observed that *Dynamosaurus imperious* (junior subjective synonym of *T. rex*) was distinguished by its number of dentary teeth "twelve to thirteen"; indeed, the results here find that *T. rex* has the lowest number of dentary teeth (12) among tyrannosaurids. The same is also true of the maxilla, which has as few as 11 teeth (e.g., LACM 23844).

In 1906, Osborn synonymized *D. imperiosus* with *T. rex*, and he expanded the list of diagnostic characters. Of these, several are symplesiomorphies: presence of the internal antorbital fenestra and the maxillary fenestra, horizontal bar of the squamosal, presence of the subnarial foramen, small first dentary tooth, presence of interdental plates, 23 presacral vertebrae, five sacral vertebrae, sacral spinous processes fused into a single plate, atlantoaxial complex composed of six segments, reduced scapula and humerus, presence of abdominal ribs, pelvic construction, hollow limb and girdle bones, and long hind limbs.

Some characters (*Osborn, 1906*) are mischaracterizations, such as the abbreviated skull (whereas it is dorsoventrally deep, not rostrocaudally short) and the primary metatarsals partly co-ossified (although tightly apposed, these bones are separate structures). Also, his observation of teeth in *T. rex* that are "very broadly oval in section, transverse exceeding anteroposterior diameters" (*Osborn, 1906*: 283), deserves comment. A comparison of width to length ratios among adult *T. rex* and representative adults of *Daspletosaurus* spp. and *A. sarcophagus* (aside from the first two maxillary teeth and the first dentary tooth, which tend to be incisiform or conical and so incomparable with so-called lateral teeth that are ziphiform in shape) shows that dentary teeth that are wider than long have a scattered distribution among *T. rex* adults (Table 11). Maxillary teeth with widths that exceed their lengths are not seen in *T. rex*, but, in general, the maxillary teeth of adult *T. rex* tend to be wider than what is seen in its close relatives, and the dentary teeth are somewhat narrower (Table 11). In contrast, the maximum width to length ratio of lateral teeth (excluding wide mesial teeth) in young adults, such as MOR 1125, is only 68% (mean of all MOR 1125 teeth: 72%), indicating that autapomorphically wide teeth occur late in growth. Therefore, only three of Osborn's diagnostic characters are seen here: gigantic size, the low count of twelve dentary teeth, and the extreme width of the maxillary- and mesial dentary teeth.

### Individual variation

In *T. rex* individual variation increases with maturity, which contradicts the naïve expectation that variation should decrease upon the passage through an accumulation of developmental constraints throughout growth (see "Results" above). Evidently, constraints are tightest early in growth, as shown by the absence of character reversals from juveniles and the least mature young adult. The presence of reversals throughout adulthood is evidence that variation is less tightly constrained, allowing characters to freely

reverse and progress. Also, the onset of reversals coincides with crossing the plesiomorphic 3,000 kg threshold in mass (Fig. 2).

The presence of more reversals than progressions in FMNH PR2081, presumably the most mature specimen in the sample, raises the question of whether or not an individual specimen should be considered the heuristic representative of the terminal growth stage on the ontogram. Among adult *T. rex*, each node is supported by a relatively low number of synontomorphies that include progressions and reversals; the characters that diagnose FMNH PR2081 are dominated by reversals, which is the pattern seen in the individual variation of other adults (AMNH FARB 5027, AMNH FARB 5029, AMNH FARB 5117, LACM 23844, MOR 555, MOR 980, MOR 1131, MOR 2822, NMMNH P-3698, RSM 2523.8, SDSM 12047, UWBM 99000, UWGM 191). Among synontomorphies, where both reversals and progressions are seen, in only one case do the progressions outnumber the reversals, which emphasizes the difference seen in FMNH PR2081.

It is possible that the reversals might have been actual reversals over the lifetime of the individual (e.g., 0 to 1 to 0), or juvenile characters that were carried unchanged into adulthood. However, it is not possible to make the distinction between reversals and static character states of individuals, on the one hand, from true ontogenetic reversals on the other, without knowing the entire ontogeny of the individual organism in question.

### Sexual maturity

Among extant archosaurs (e.g., *Alligator mississippiensis*), sexual maturity is reached at approximately half adult size (*Wilkinson & Rhodes, 1997*; *Foth et al., 2018*; *Wilkinson et al., 2016*). If this pattern is plesiomorphic for Archosauria (Crocodylia + Dinosauria), the naïve null hypothesis for *T. rex* is that the onset of sexual maturity occurred when skull length reached 70 cm. Ergo, specimens such as BMRP 2002.4.1 (skull length: 74 cm) indicate that sexual maturity occurred before the onset of the extreme transition from long and low skulls to the deep and stocky frame of subadults and adults (Fig. 12).

At the very least, it is improbable that the 15-year-old female subadult BMRP 2006.4.4 represents the earliest onset of sexual maturity in *T. rex* (*Woodward et al., 2020*). If sexual maturity occurred earlier, then *T. rex* is an organism with determinate growth in which growth continues after the onset of sexual maturation and stops before senescence (sensu *Lincoln, Boxshall & Clark, 1982* in *Wilkinson et al. (2016)*). As in *A. mississippiensis*, the growth rate in *T. rex* reduces once sexual maturity is reached, where it is relatively low in large juveniles and subadults (*Woodward et al., 2020*) and it nearly ceases in the young adults and adult categories (*Horner & Padian, 2004*). If these patterns of growth rate are comparable, then it is plesiomorphic and unaffected by metabolism (i.e., ectothermy in crocodylians, endothermy in *T. rex*). It is thought that the pattern seen in *A. mississippiensis* evolved as an energy expense-optimization strategy (*Wilkinson et al., 2016*), which might be the case for all nonavian archosaurs. Given that female *A. mississippiensis* can produce viable eggs for their entire lives, the term "senescent" as used here for *T. rex* does not imply that senescent adults are nonreproductive.

### Sexual dimorphism

Sexual dimorphism is marked by one or more distinct phenotypic differences between males and females, excluding differences in size (*Padian & Horner, 2011*). The inference of the absence of sexual dimorphism in *T. rex* is supported by three lines of evidence: (1) the ontogram is linear and pectinate, lacking a distinct bifurcation that unites females on one branch and males on the other; (2) the number of specimens more mature than MOR 1125 is high (22), indicating that (a) the sample size is sufficient to capture the signal of two distinct morphs and (b) the probability of sampling only one sex over the other is vanishingly low; and (3) individual variation does not separate out two identifiable groups among subadult and adult specimens. In other words, aside from subtle differences, all adult *T. rex* simply look alike.

The absence of dimorphism pertains to the osteodental characters included in the analysis; in life, dimorphism might have been expressed by differences in soft tissues (e.g., integument pigmentation) that do not preserve as fossils. The possibility that dimorphism was expressed by size and growth rate, as is seen in living crocodylians (*Taylor et al., 2016*), requires a sample of unambiguously female specimens, which is currently unavailable for *T. rex*. Finally, it is notable that among living crocodylians (e.g., *C. yacare*, *Crocodylus johnstoni*) dimorphism in size is seen between hatchlings, where females are smaller than males (*Campos et al., 2014*; *Edwards et al., 2017*). Therefore, should size dimorphism be found in *T. rex*, it might also be present, ontogenetically, from start to finish. Therefore, the absence of a basal bifurcation in the ontogram shows that early size-independent dimorphism was absent from *T. rex*.

Among crocodylians, size-independent sexual dimorphism is limited to the shape of the external naris in *Gavialis gangeticus* and *Caiman latirostris* and to the width of the interorbital end of the temporal roof in *Crocodylus porosus* (*Foth, Bona & Desojo, 2013*). In contrast, variation in the shape of the bony naris and the interorbital region in *T. rex* is ontogenetically controlled and dimorphism is not seen. In summary, *T. rex* shows the pattern of variation (high ontogenetic variation, dimorphism absent, high individual variation) that is expected where only species recognition is at work (*Padian & Horner, 2011*).

### Size, maturity, and age

The suggestion has recently been made that size might correlate with maturity in *T. rex* (*Persons, Currie & Erickson, 2019*), which was based on histological evidence. The specimen in question (RSM 2523.8), the youngest adult identified here, was hypothesized to exceed FMNH PR2081 in maturity based on the presence of a high degree of remodeling in the fibula that has produced a dense system of secondary osteons (*Persons, Currie & Erickson, 2019*). It is expected that if remodeling is congruent with maturity, then on the ontogram RSM 2523.8 would share a most recent growth stage with FMNH PR2081 to the exclusion of all other specimens.

Among living archosaurs (e.g., *Melanosuchus niger*, *Caiman yacare*) variation in size is high, which is evidenced by high standard deviations (e.g., *Platt et al., 2011*; *Taylor et al., 2016*), wide 95% confidence intervals (*Campos et al., 2014*), individual variation in

growth rates (*Campos et al., 2014*), and by the relative inaccuracy of age estimation from size (*Campos et al., 2014*). Individual variation tends to be high among large, mature animals (e.g., *C. acutus*; *Platt et al., 2011*). Also, agreement between growth rate data and growth curve models are dependent upon sample size (*Taylor et al., 2016*). This suggests that currently the sample size for *T. rex* (seven specimens in *Erickson et al. (2004)*) is insufficient for accurately capturing the congruence between mass and age.

Also, size in crocodylians is affected by multiple extrinsic variables, including seasonality, seasonality of growth, incubation conditions, mean temperature (although this effect is not always found; *Campos et al., 2014*), rainfall, prey type, prey abundance, population density, social milieu, and disease (*Campos et al., 2014*; *Taylor et al., 2016*). Although not all of these variables would have affected *T. rex*, an endothermic dinosaur, to the same extent that is seen in ectothermic crocodylians, it is not surprising that the sizes seen among the most mature specimens are not congruent with maturity. Alternatively, if extrinsic factors had a negligible effect upon body size, then sexual dimorphism might explain the noncongruence between maturity and size (but as shown, it does not). Among living crocodylians, linear measurements of adult animals are more variable than what is seen in juveniles (*Platt et al., 2011*), and *T. rex* is consistent with that pattern.

In *A. mississippiensis*, growth rate effectively ceases at less than half the maximum age (*Wilkinson et al., 2016*); a similar pattern is seen in *T. rex*, where growth rate is greatly reduced at slightly greater than half the maximum age (18–28 years; *Horner & Padian, 2004*). The relatively wide range of size and mass for *T. rex* (*Erickson et al., 2004*; *Persons, Currie & Erickson, 2019*; *Snively et al., 2019*;) has two possible explanations: (1) it had determinate growth type II (sensu *Sebens, 1987* in *Wilkinson et al. (2016)*) where although final size is genetically controlled it is affected by extrinsic factors, resulting in a wide range of adult sizes; alternatively (2) it had determinate growth type I where it has sexual size dimorphism, resulting in the scatter of adult sizes, but the environment has little effect upon asymptotic size. Therefore, the evidence shows that *T. rex* had determinate type II growth.

The variation in size among the categories of adult (i.e., the noncongruence between size and maturity) is evidence of determinate growth, where the greatest amount of variation in size is seen that reflects the idiosyncratic differences in accumulated growth rate between individuals. In other words, the predictive value of size in *T. rex* ceases once the individual asymptote is reached (*Monteiro, Cavalcanti & Sommer, 1997*; *Tucker et al., 2006*).

## Plesiomorphic archosauriform growth patterns

A literature review of craniomandibular ontogeny in extinct (*Bhullar et al., 2012*; *Ezcurra & Butler, 2015*; *Foth, Hendrick & Ezcurra, 2016*; *Horner & Goodwin, 2006*, *2009*) and extant (*Dodson, 1975*; *Piras et al., 2010* in *Foth, Bona & Desojo (2013)*; *Platt et al., 2011*; *Wu et al., 2006* in *Foth, Bona & Desojo (2013)*) archosaurs was done to identify the ancestral growth trends inherited by *T. rex* from its archosauriform ancestor.

Evidence for growth trends established in the common ancestors Archosauriformes (i.e., the common ancestor of *Proterosuchus* + Archosauria, and all of its descendants, living and extinct), Archosauria (i.e., the common ancestor of Crocodylia + Dinosauria, and all of its descendants, living and extinct), Dinosauria (i.e., the common ancestor of *Triceratops* and *Passer*, and all of its descendants, living and extinct), Saurischia (i.e., the common ancestor of *Massospondylus* + *T. rex*, and all of its descendants, living and extinct), and Neotheropoda (i.e., the common ancestor of *Coelophysis* + *T. rex*, and all of its descendants, living and extinct), were drawn from comparisons between *T. rex*, a sample of saurischian dinosaurs (*Bhullar et al., 2012*; *Foth, Hendrick & Ezcurra, 2016*), extant crocodylians, and the extinct *Proterosuchus* (*Ezcurra & Butler, 2015*); the comparisons are summarized in Table 23. The character states were optimized onto a topology that was reconstructed in MacClade (*Maddison & Maddison, 2005*) to identify ancestral states in the phylogenetic hierarchy. The informal terms "archosauriforms" and "archosaurs" are used in place of their formal names in this discussion; the name *Tyrannosaurus bataar* is used in place of "*Tarbosaurus*." A summary of the phylogenetic framework, the taxa included in this comparison, and the results are shown in Fig. 33.

### Dorsoventral deepening of the skull frame

Ontogenetic deepening of the skull frame comes from several lines of evidence, including the dorsoventral height of skull regions, specific bones, and fenestrae. In the archosauriforms *Tyrannosaurus rex*, crocodylians and *Proterosuchus*, the horizontal ramus of the maxilla increases in dorsoventral height (*Ezcurra & Butler, 2015*) (Fig. 33).

Another correlate of increased skull height in archosauriforms is the decreased height of the postorbital that is seen in *T. rex*, where the contribution of the bone to skull height is reduced by the dorsoventral expansion of the jugal, and *Proterosuchus*, where its height relative to total skull length decreases (*Ezcurra & Butler, 2015*). A reduction in the length of the orbital fenestra is another correlate of skull deepening, which is seen in *Coelophysis* (*Bhullar et al., 2012*) and *T. rex*.

In archosaurs, the snout and jugal of crocodylians increase in depth, as also occurs in *T. rex*, in *Melanosuchus niger* (*Foth, Bona & Desojo, 2013*), *C. acutus*, *Mecistops cataphractus*, and *Tomistoma schlegelii* (*Foth, Bona & Desojo, 2013*); likewise, a shallow snout is a feature shared between juvenile caimans (*Foth et al., 2018*) and *T. rex*. This change is one of several correlates of the ontogenetic increase in height of the entire skull frame. Analysis of morphometric data quantifies the deepening of the skull, which is seen in *Massospondylus*, the megalosaurid *Dubreuillosaurus*, *Allosaurus*, and *T. bataar* (*Foth, Hendrick & Ezcurra, 2016*). Optimization of the trends of increase in snout height and skull height onto the phylogenetic topology shows that both are ancestral for Archosauriformes, a trend that is not seen in *Coelophysis* (Fig. 33).

### Expansion of the adductor region

In the archosauriforms *T. rex*, *Proterosuchus*, and all crocodylian taxa, the orbital fenestra is ontogenetically reduced in size, a correlate of adductor expansion (*Ezcurra & Butler, 2015*; *Foth, Bona & Desojo, 2013*; *Foth et al., 2018*). In *T. rex* and *Proterosuchus* the orbital

Table 23 Summary of the character evidence for shared ontogenetic trends throughout the ontogeny of Archosauriformes. Rows correspond to major growth trends identified in the left hand column and, in the following cells, the osteological correlates of the trend for each taxon; columns correspond to taxa. The taxonomic distribution of the reduction of cephalic ornamentation is given in the text. emf, external mandibular fenestra. See text for details and sources; see Fig. 33 for an illustrated summary.

| Character/ taxon | Proterosuchus | Crocodylia | Massospondylus | Coelophysis | Dubreuillosaurus | Allosaurus | Tyrannosaurus bataar | T. rex |
|---|---|---|---|---|---|---|---|---|
| Increase in snout height | height maxillary horizontal ramus | snout height | snout height | ? | snout height | snout height | snout height | height maxillary horizontal ramus, snout height |
| Increase skull height | height orbital & laterotemporal fenestrae | ? | enlarged postorbital, dorsal shift bony naris | ? | caudally inclined lacrimal, postorbital enlarged | lacrimal ventral ramus size, postorbital region height, angulation between maxilla and jugal | lacrimal ventral ramus size, postorbital region height, dorsal shift | height orbital & laterotemporal fenestrae, caudally inclined lacrimal, postorbital enlarged, postorbital region height, angulation between maxilla and jugal |
| Increase in adductor chamber size | length orbital fenestra | length orbital fenestra | length orbital fenestra | length orbital fenestra | length orbital fenestra | length orbital fenestra | length orbital fenestra | length orbital fenestra |
| Increase in skull width | width frontoparietal region | width frontoparietal region, width snout | ? | width snout | ? | ? | ? | width frontoparietal region, width snout |
| Antorbital fossa length | ? | ? | decrease | increase | decrease | decrease | decrease | ? |
| Rostrocaudal jaw joint position | caudal | ? | rostral | caudal | caudal | rostral | caudal | ? |
| Dorsoventral jaw joint position | ventral | ? | ventral | dorsal | no change | ventral | ventral | ? |
| Increase of mandible height | ? | height mandible & emf | ? | ? | ? | ? | ? | height mandible & emf |
| Tooth number | increase maxillary teeth | ? | ? | ? | ? | ? | ? | increase maxillary teeth |
| Bite force | ? | increase | ? | ? | ? | ? | ? | increase |
| Deltopectoral crest | ? | ? | ? | enlarged crest | ? | ? | ? | enlarged crest |
| Trochanteric crest | ? | ? | ? | enhanced crest | ? | ? | ? | enhanced crest |

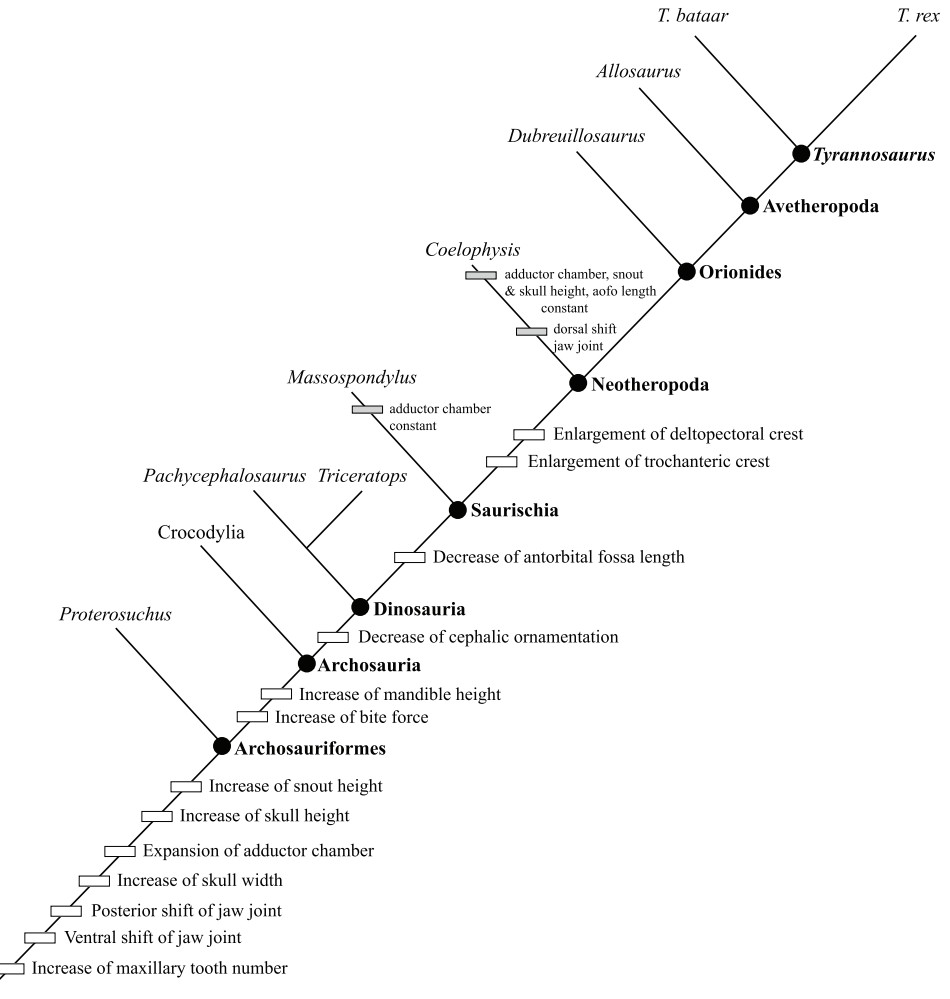

**Figure 33 A simplified cladogram of living and extinct Archosauriformes showing 13 cranial and postcranial growth changes that are optimized as synapomorphies.** Most of the growth changes are ancestral for Archosauriformes. The position of several characters at progressively exclusive clades is almost certainly an artifact of missing data (e.g., increase in mandible height, enlargement of muscle attachments, etc.) and they are predicted to be synapomorphic for Archosauriformes once the appropriate data are acquired. This comparison shows that highly derived species such as *Tyrannosaurus rex* do not deviate from the ancestral growth trends that first evolved in significantly smaller taxa. See text for sources; see Table 23 for the distribution of character states among the taxa.

fenestra decreases in length and increases in height (*Ezcurra & Butler, 2015*). Ergo, this reduction in size is almost certainly the result of two processes, namely the rostrocaudal expansion of the adductor chamber and dorsoventral expansion of the skull frame. The adductor region, including the laterotemporal fenestra, increases in height in *T. rex* and *Proterosuchus* (*Ezcurra & Butler, 2015*). Geometric morphometric comparison shows that the adductor region is not expanded in *Massospondylus* or *Coelophysis*, whereas it is expanded in *Dubreuillosaurus*, *Allosaurus*, and *T. bataar* (*Foth, Hendrick & Ezcurra, 2016*). Optimization of character states onto the phylogenetic

topology recovers adductor chamber expansion as plesiomorphic for Archosauriformes (Fig. 33).

### Mediolateral widening of the skull frame

In archosauriforms, an overall increase in snout width is seen in *T. rex* and in the crocodylians *Alligator sinensis* (*Wu et al., 2006* in *Foth, Bona & Desojo (2013)*), *Crocodylus acutus*, *Mecistops cataphractus*, and *Tomistoma schlegelii* (*Piras et al., 2010* in *Foth, Bona & Desojo (2013)*; *Platt et al., 2011*), and also in *A. mississippiensis* (*Dodson, 1975* in *Foth, Bona & Desojo (2013)*) and *C. novaeguineae* (*Hall & Portier, 1994* in *Foth, Bona & Desojo (2013)*). Widening of the snout in crocodylians is thought to coincide with changes in diet, for example from invertebrates to vertebrates (*Platt et al., 2011*).

An ontogenetic increase in frontal (=interorbital) width is seen in *Tyrannosaurus rex* and all crocodylians except *A. sinensis* (*Wu et al., 2006*; *Foth, Bona & Desojo, 2013*; *Edwards et al., 2017*). The ontogenetic widening of the parietals that is seen in *Proterosuchus* might be homologous to the increase in width seen in *T. rex* and crocodylians (*Ezcurra & Butler, 2015*).

Ontogenetic widening of the skull in archosaurs is indicated by correlates seen in the snout and dorsal skull roof. In *T. rex*, caimans (e.g., *C. latirostris*, *C. sclerops*, *C. yacare*), and crocodiles (e.g., *Crocodylus acutus)* the rostral end of the snout widens (Platt at al., 2011). The premaxilla shortens in *Melanosuchus niger* (*Foth, Bona & Desojo, 2013*) and *Coelophysis* (*Bhullar et al., 2012*), which corresponds to the widening and medialward reorientation of the premaxilla (i.e., increase in snout width) in *T. rex*. Optimization of skull widening on the topology recovers it as ancestral for Archosauriformes (Fig. 33).

### Antorbital fossa length

Geometric morphometric comparisons recover a shortening of the antorbital fossa during growth, a trend that is ancestral at least for Saurischia, aside from an apomorphic lengthening in *Coeolophysis* (*Foth, Hendrick & Ezcurra, 2016*; Fig. 33). This landmark-based comparison has not been done for *Proterosuchus* or *T. rex*, but it is predicted here that it will be recovered in both taxa and, hence, the trend is plesiomorphic for Archosauriformes.

### Jaw joint position

Geometric morphometric comparisons have recovered the dorsoventral and rostrocaudal growth trends of the cranial jaw joint for *Proterosuchus* (*Ezcurra & Butler, 2015*) and Saurischia (*Foth, Hendrick & Ezcurra, 2016*). Optimization of jaw joint position recovers a caudalward shift as the ancestral state for Archosauriformes, with two independently acquired apomorphic rostral shifts in *Massospondylus* and *Allosaurus* (Fig. 33). It is predicted here that *T. rex* will be found to have the same caudalward shift with *T. bataar*. Likewise, landmark-based comparisons (*Ezcurra & Butler, 2015*; *Foth, Hendrick & Ezcurra, 2016*) show that a ventralward shift of the jaw joint is plesiomorphic for Archosauriformes, with an apomorphic dorsalward shift in *Coelophysis* and no dorsoventral shift in *Dubreuillosaurus* (Fig. 33). It is predicted here that *T. rex* will be found to share a ventralward shift of the jaw joint with its sister species, *T. bataar*.

### Dorsoventral deepening of the mandibular ramus

The ontogenetic deepening of the mandibular ramus in archosaurs is evidenced by the entire ramus and by the external mandibular fenestra. In *T. rex* and crocodylians the external mandibular fenestra and the mandibular ramus increase in dorsoventral height (*Monteiro & Soares, 1997*). Optimization of the trend of increasing mandible height onto the phylogenetic topology recovers it as a synapomorphy of Archosauria (Fig. 33). The lower jaw was not included in the study of *Proterosuchus* ontogeny and so it is unknown if the mandible deepened as well (*Ezcurra & Butler, 2015*).

### Increase in maxillary tooth count

An ontogenetic increase in tooth count of archosauriforms is seen in *T. rex* and *Proterosuchus* (*Ezcurra & Butler, 2015*); in contrast, tooth number is ontogenetically constant in extant crocodylians (*Brown et al., 2015*). Optimization of the trend in tooth count onto the phylogenetic topology recovers an increase in number as plesiomorphic for Archosauriformes (Fig. 33).The presence of ontogenetic tooth increase in other dinosaurs and the extant sister clade of Archosauria, Squamata, indicates that the pattern seen in crocodylians is the apomorphic condition (*Brown et al., 2015*).

### Bite force

A growth-related increase in bite forces is seen in crocodylians and in *T. rex*, a trend that is continued into adulthood (*Bates & Falkingham, 2012*; *Erickson, Lappin & Vliet, 2003*; *Erickson et al., 2013*; *Gignac & Erickson, 2014*, *2017*; *Gignac & O'Brien, 2016*). Optimization of this trend onto the phylogenetic topology recovers increasing bite force as a synapomorphy of Archosauria; it is expected that this trend is pleiomorphic for Archosauriformes, once data are obtained for *Proterosuchus* or other basal taxa.

### Cephalic ornamentation

Growth-related reduction in cephalic ornamentation is seen in *Triceratops* (*Horner & Goodwin, 2006*), *Pachycephalosaurus* (*Horner & Goodwin, 2009*), and in *T. rex*. Optimization of the growth trend onto the phylogenetic topology recovers this pattern as synapomorphic for Dinosauria (Fig. 33); it is predicted that this trend will be an archosauriform synapomorphy as appropriate data are documented.

### Appendicular skeleton

*T. rex* shares two postcranial growth changes with *Coelophysis* (*Griffin, 2018*), which indicates synapomorphic growth changes that possibly unite Neotheropoda. These include the thickened deltopectoral crest of the humerus and enhancement of the trochanteric shelf of the femur. When optimized onto the phylogenetic topology, both trends are recovered as synapomorphies of Neotheropoda; it is predicted that these trends will be found to be synapomorphies of Archosauriformes.

### Summary

Taken together, the ancestral growth pattern of the archosauriform skull that was inherited by *T. rex*, was marked by (1) increase in height of the entire skull, (2) rostrocaudal and dorsoventral expansion of the adductor chamber, (3) increase in skull width,

(4) caudalward shift of the jaw joint, (5) ventralward shift of the jaw joint, and an (6) increase in maxillary tooth number (Fig. 33). The ancestral growth pattern of the archosaurian skull that was inherited by *T. rex* included (1) increase in mandible height and (2) an increase in bite force. The ancestral growth pattern of the dinosaurian skull that was inherited directly by *T. rex* was the reduction in cephalic ornamentation. The ancestral growth pattern of the saurischian skull that was inherited by *T. rex* included the decrease in the length of the antorbital fossa. The ancestral growth pattern of Neotheropoda that was inherited directly by *T. rex* included increase in size of (1) the deltopectoral crest of the humerus and (2) the trochanteric crest of the femur. It is expected that the more exclusive trends will eventually be seen in *Proterosuchus* and other basal archosauriformians once adequate data are obtained.

## Early onset of adult skull shape

The unique skull shape of *T. rex*, in having a mediolaterally narrow snout but wide orbitotemporal region (Figs. 12 and 26), is seen in the youngest of the most complete specimens in the sample (e.g., CMNH 7541) that is approximately 40% of adult skull length. This phenomenon of the presence of the adult shape in juveniles (i.e., developmental capture; sensu *Padian, 1995*) is not unusual among archosaurs; for example, it is also seen in caimans (*Monteiro & Soares, 1997*). This phenomenon is notable in that the adult skull shape in *T. rex* is derived in contrast to its closest relatives, but it ontogenetically precedes the occurrence of the suite of other evolutionary novelties that sets *T. rex* apart from *T. bataar* and other advanced tyrannosaurines.

This observation is evidence that skull shape in *T. rex* is not recapitulatorily locked to a later growth stage; rather, continuity is seen in skull function and behavior between juvenile and adult *T. rex*, and niche partitioning between those small and large growth stages was primarily governed by increased bite force instead of fundamental differences in sensorimotor integration and functional morphology (aside from the consequences of the corresponding increase in skull and jaw height) and mode of forage. From its earliest growth stage, *T. rex* had already fundamentally departed from its ancestral skull morphology and, presumably, function.

## Juvenile to 'Adult' skull morphotype transition

The results found here show that the extreme ontogenetic transition in skull shape of *T. rex* is often mischaracterized as between juvenile to adult, whereas the switch is actually much earlier, between juvenile and subadult. In living crocodiles, transitions in skull shape mark changes in diet (e.g., *C. acutus*; *Platt et al., 2011*), and the same is doubtlessly true in *T. rex*. Aside from skull shape in living crocodiles (e.g., *C. niloticus*), size is an important factor that drives juveniles from their natal river range to a lacustrine habitat (*Hutton, 1989*); likewise, the extreme transition in skull shape in *T. rex* might also mark a dispersal stage with movement from, say, a closed habitat that provided protection from adults to an open setting where safety from conspecifics was no longer a priority. Presumably, the deep-skulled morphotype resulted in an expansion of range among

**Table 24 Predicted LAG counts for *Tyrannosaurus rex* specimens extrapolated from the ontogram.** Estimated ages are indicated by a tilde (~), histologically aged specimens are in boldface.

| Specimen | Age (years) |
|---|---|
| **LACM 28471** | **2** |
| AMNH FARB 5050, CMNH 7541, DDM 344.1 | ~3–12 |
| **BMRP 2002.4.1** | **13** |
| **BMRP 2006.6.4** | **15** |
| RSM 2990.1, LACM 23845 | ~15–17 |
| **MOR 1125, TMP 1981.006.0001** | **18** |
| AMNH FARB 5117, LACM 150167 | ~18–22 |
| **TMP 1981.012.0001** | **22** |
| UWBM 99000 | ~22 |
| AMNH FARB 5027, AMNH FARB 5029, NHMUK R7994, CM 9380, MOR 008, MOR 555, MOR 980, MOR 1131, MOR 2822, LACM 23844, NMMNH P-3698, RSM 2523.8, SDSM 12047, UCMP 118742, UMNH 11000, UWGM 181 | ~23–27 |
| **FMNH PR2981** | **28** |

subadults and adults, where females would be limited to nesting areas and males would have been more widely distributed across the landscape (cf. *Hutton, 1989*).

## Ontogeny and module integration

Uniform changes across the skull implies functional integration of the skull in contrast to modularity (*Monteiro, Cavalcanti & Sommer, 1997*). In *T. rex*, the skull modules follow the same general pattern of change, starting with few changes (<10), followed by a peak (>10) from the 5th to 6th growth stages, followed thereafter by a low rate of change (<10) (Table 4; Fig. 11). The skull roof has a peak of change at growth stage 21, showing its independence from the other modules. Beyond that instance, the modules do not tend to change independently from each other from young adulthood onwards, indicating that the modules are tightly integrated with each other, which is consistent with the akinetic hypothesis (*Cost et al., 2019*) in contrast to the flexible skull hypothesis (cf. *Werneburg et al., 2019*).

## Predictions

1. EFS will be found in AMNH FARB 5027, AMNH FARB 5029, NHMUK R7994, CM 9380, LACM 23844, MOR 008, MOR 980, MOR 1131, MOR 2822, NMMNH P-3698, UCMP 118742, UMNH 11000, and UWGM 181, should appropriate bones be available for analysis. However, an EFS is absent from MOR 555 (*Horner & Padian, 2004*), which suggests that the congruence between the presence of an EFS, maturity, and chronological age is not tightly constrained. Specific predictions of LAG-based chronological ages for specimens in this study are given in Table 24.

2. EFS will be absent from AMNH FARB 5117, LACM 150167, and UWBM 99000.

3. If senescence in *T. rex* does not imply reproductive nonviability, then medullary bone will be found in pregnant senescent females.

4. If the distinct ridges of the subcutaneous surface of the maxilla result from ossification of the dermis, then histological sections will show an overlay of metaplastic bone apposed to the pericortical surface (sensu *Horner & Goodwin, 2009*). That is, if flat facial scales precede the onset of armor-like skin and ossification of the dermis, then histological analysis will show a transition from the absence of metaplastic bone to its presence.

5. If secondary metamorphosis is present, the relationship between bite force and independent measures of body size will be nonlinear.

6. In terms of quantitative functional morphology, complete subadult skulls will be found to be more similar to adults than to juveniles, indicating an abrupt and wholesale morphological transition from low to tall skulls, whereas the postcranium will show a continuum from juvenile to adult.

## CONCLUSIONS

1. A single growth series for *T. rex* was obtained that can be divided into 21 growth stages; FMNH PR2081 was recovered as the most mature specimen, whereas RSM 2523.8 is one of the least mature adults.

2. Specimens coded with as few as 1.8% of the characters are in the backbone ontogram.

3. Specimen completeness does not influence the number of unambiguously-optimized synontomorphies that support each node.

4. Five growth categories (juvenile, subadult, young adult, adult, senescent adult) were diagnosed based on histology, synontomorphies, and, in part, size and mass; sharp boundaries between categories are seen at the subadult and adult categories.

5. The sample is numerically biased toward specimens in the subadult, young adult, adult, and senescent adult categories; that is, there is a large gap in the growth series at the juvenile to subadult transition. Also, juveniles are underrepresented in the sample.

6. The problematic specimen TMM 41436-1 is a subadult, which accounts for its differences from adults.

7. Phylogenetic and nonphylogenetic changes follow a similar frequency distribution; phylogenetic changes are more frequent early in ontogeny (growth stages 5–9) than later in growth.

8. Mandibular changes are completed before cranial changes.

9. The dorsotemporal fossa of the frontal was an origin for adductor musculature.

10. The greatest number of growth changes are seen in the large juvenile and subadult growth categories, at growth stages 5 and 6.

11. The number of growth changes generally decreases during adulthood.

12. The skull roof module undergoes the most growth change.

13. Pneumatic changes are dominated by the antorbital (=paranasal) air sac system.

14. The skull frame is dominant among apneumatic changes in the skull and mandible.

15. Decreases in maxillary and dentary tooth count are broadly congruent with maturity; tooth count in both bones initially increases before it decreases.

16. Most braincase sutures close during the young adult category, whereas two stay open.

17. Changes to the pectoral girdle and pes are dominant early in ontogeny, whereas axial and pelvic girdle changes occur late; the fibula changes throughout. The most postcranial changes happen to the fibula.

18. Body size and mass are not congruent with maturity during adulthood.

19. Maturity is congruent with chronological age, bite force, REQ, the presence and depth of the dentary groove, and decrease in agility.

20. Maturity is incongruent with geographic location, stratigraphic position, phylogeny, and ischial divergence.

21. The transition from a long and low skull to a stout and deep skull (i.e., the juvenile-subadult transition) occurred rapidly within a 2-year time span. Ergo, the osteological correlates of skull strength occurred before adult size (i.e., somatic maturity) was reached.

22. *T. rex* exceeded the plesiomorphic size and mass of Tyrannosauridae between its 15th and 18th years.

23. Sexual maturity in *T. rex* occurred before its 15th year and as early as its 13th year. Ergo, as in crocodylians and other reptiles, including non-avian dinosaurs (*Erickson et al., 2007*; *Lee & Werning, 2008*), *T. rex* reached sexual maturity before reaching asymptotic adult size.

24. Skeletodental sexual dimorphism is absent from *T. rex*.

25. There is no evidence for the taxon that is informally dubbed *Tyrannosaurus* "*x*".

26. The entire skeleton is affected by individual variation; reversals of individual variation did not occur until young adulthood (growth stage 8).

27. Individual variation affects the snout dorsum the most; display features are not disproportionately affected by individual variation.

28. The variability in adult size implies that *T. rex* had determinate growth type II (sensu *Sebens, 1987*).

29. Growth patterns of the skull do not reflect the distribution of stress loads upon it.

30. The adductor chamber rostrocaudally expands at the young adult growth stage.

31. Removal of redundant putative oversplit characters results in loss of topological resolution, which is almost certainly an indication of the patchiness of the character matrix.

32. The type specimen is an adult; three of the characters of Osborn's diagnosis of the taxon are defensible.

## INSTITUTIONAL ABBREVIATIONS

**AMNH**    American Museum of Natural History, New York
**NHMUK**   Natural History Museum, London

| | |
|---|---|
| BMRP | Burpee Museum of Natural History, Rockford |
| CM | Carnegie Museum of Natural History, Pittsburgh |
| CMNH | Cleveland Museum of Natural History, Cleveland |
| DDM | Dinosaur Discovery Museum, Kenosha |
| FMNH | Field Museum of Natural History, Chicago |
| LACM | Los Angeles County Museum of Natural History, Los Angeles |
| MOR | Museum of the Rockies, Bozeman |
| NMMNH | New Mexico Museum of Natural History and Science, Albuquerque |
| OMNH | Sam Noble Oklahoma Museum of Natural History, Norman |
| RSM | Royal Saskatchewan Museum, Eastend |
| SDSM | South Dakota School of Mines and Technology, Rapid City |
| TMM | Texas Memorial Museum, Austin |
| TMP | Royal Tyrrell Museum of Palaeontology, Drumheller |
| UCMP | University of California at Berkeley Paleontology Museum, Berkeley |
| UMNH | Utah Museum of Natural History, Salt Lake City |
| UWBM | Burke Museum, Seattle |
| UWGM | Geology Museum, Madison |

## ACKNOWLEDGEMENTS

For access to specimens I thank: M. Norell, C. Mehling (AMNH); S. Chapman (NHMUK); M. Henderson, J. Mathews, S. Williams (BMRP); M. Dawson, M. Lamanna (CM); M. Ryan, M. Williams (CMNH); P Makovicky, W. Simpson, A. Stroup (FMNH); L. Chiappe, S. McLeod, M. Walsh (LACM); A. Atwater, B. Helms, J. Horner, J. Scannella (MOR); T. Williamson (NMMNH); D. Evans, K. Seymour (ROM); T. Tokaryk (RSM); M. Pinsdorf (SDSM); D. Brinkman, D. Henderson; B. Strilisky (RTMP); M. Holland, G. Wilson (UWBM); C. Eaton (UWGM). For photographs of TMM 41436-1 I thank T. Adams; I thank T. Williamson for images of NMMNH P-3698. For room and board during extended collections visits, I thank J. Scannella, K. Scannella, K. Tremaine, S. Williams (MOR); T. Williamson (NMMNH); K. Seymour (ROM); D. Brinkman, D. Henderson (RTMP); M. Holland (UWBM). I thank the landowners who donated specimens from their private land to public trusts, namely the Buyse (MOR 008) and the Skillman (MOR 002) families. For discussion I thank J. Miller (Carthage College) and T. Miller. For assistance with locality data and maps I thank E. Berry (SDSM), C. Levitt-Bussian (UMNH), A. McGee (CMNH), C. Eaton (UWGM), A. Henrici (CM), P. Holroyd (UCMP), M. Lamanna (CM), G. Liggett (BLM), D. Malinzak (Black Hills State University, Spearfish, SD), R. McKellar (RSM), C. Mehling (AMNH), W. Simpson (FMNH), S. Williams (MOR), T. Williamson (NMMNH), G. Wilson (UWBM), and the US Army Corps of Engineers. I thank D. Evans (ROM) for discussion of HCF stratigraphy and references; and J. Clarke (UT-Austin) for discussion of sexual dimorphism in Dinosauria. I thank M. Seitz for reviewing the character list and A. Zietlow for reviewing the manuscript, figures, and tables prior to submission. I thank C. Sabbar for technical help through Library Services (Carthage College) and P. Kivolowitz (Carthage College) for

additional technical support. The specimens DDM 344.1 and DDM 1536.8 were collected under BLM Paleontological Resources Permits M 103845 and MTM 110440 (issued to TDC), respectively, and G. Liggett, D. Melton, and G. Smith (all of the BLM) are thanked for facilitating the approval process. Finally I thank the reviewers (anonymous, D. Barta, J. Frederickson), the primary editor (A. Farke) for their helpful comments that greatly improved the first draft, and journal staff editors (P. Binfield, E. Jung, S. Kusy, S. Johnson) for their helpful feedback and facilitation during the review process.

### Funding
Travel for this work was funded in part by a Faculty Research Grant from Carthage College. The funders had no role in study design, data collection and analysis, decision to publish, or preparation of the manuscript.

### Grant Disclosures
The following grant information was disclosed by the authors:
Carthage College.

### Competing Interests
The author declares that they have no competing interests.

### Author Contributions
- Thomas D. Carr conceived and designed the experiments, performed the experiments, analyzed the data, prepared figures and/or tables, authored or reviewed drafts of the paper, and approved the final draft.

### Data Availability
The raw data for the cladistic analysis is available in Data S3.

All of the data (except for identifiable locality information) in the correlation tests are available in the primary tables that are an intrinsic part of the article.

### Supplemental Information
Supplemental information for this article can be found online at http://dx.doi.org/10.7717/peerj.9192#supplemental-information.

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
