# Peer review of "A high-resolution growth series of *Tyrannosaurus rex* obtained from multiple lines of evidence"

_PeerJ, doi:10.7717/peerj.9192_

## Round 0.1 · original submission · Major Revisions

This paper provides a detailed look at ontogeny in Tyrannosaurus rex, with appropriately transparent datasets and methods. The reviewers have provided generally positive comments, with a number of mostly minor suggestions for revision. The most critical suggestion concerns the consensus tree used for interpretations (see Reviewer 3 comments). For this reason, I have suggested major revisions, because it potentially (but not necessarily) may have a significant impact upon the results and interpretations.

- Note comments from Reviewer 1 about how geographic and stratigraphic hypotheses are tested; I agree that using latitude rather than state is probably a better test (unless we're misunderstanding how the ranks were assigned, in which case just a little bit of clarifying text is OK). Also, what is meant by "corrected rank", in cases where this term is used?

- Also as noted by Reviewer 1, the existence of sexual dimorphism is probably tough to test with this particular analysis, and it could be manifested in multiple ways. Similarly, the number of species involved might also be hard to test here, especially if they share largely similar ontogenetic trajectories and only differ in a handful of characters.

- For tests related to stratigraphy within the Hell Creek Formation, to what extent might this be affected by time transgression? I.e., depending on proximity to shoreline, the same unit (e.g., lower Hell Creek) may not represent the same temporal slice in South Dakota versus Montana.

- Reviewer 3 raises the issue of using majority rule consensus trees versus strict consensus trees, and a few other methodological concerns. This is a common issue in phylogenetic analyses also, and should be addressed here. It is particularly important to identify specimens that are recovered across various locations ("wildcards"); in some cases, this may be due to incompleteness. You might consider a reduced strict consensus tree or an Adams consensus tree as one solution. This reviewer also mentioned Ontogenetic Sequence Analysis (OSA) as something to consider.

- The recent paper by Woodward et al. (2020) on tyrannosaur growth should be incorporated here as appropriate. Note that lack of inclusion does not factor into my decision, because the author of this study did not have the option to include the information that was unpublished at the time of submission!

·

Basic reporting

This paper is a long-awaited addition to the knowledge of Tyrannosaurus ontogeny from one of the foremost authorities in the field. In total, the 1,604 hypothetical growth characters from 41 specimens represent the largest and most comprehensive ontogenetic data set presented for a dinosaurian species. Generally, this paper was written with precise and clear language throughout; though I have highlighted a few areas below where changes can be made for consistency, accuracy, and brevity.

A. Commas are mostly used to separate multiple internal references, though line 820 and 939 has semicolons separating citations instead.

B. OMNH should be the Sam Noble Oklahoma Museum of Natural History.

C. Line 217, “After 81 hours” seems like unnecessary detail. The timing of the analysis will depend on the machine used.

D. It Is unclear which definitions for growth categories (juveniles, subadults, adults, etc.) you are using in the methods (e.g., “subadult specimens” from Line 154). You define these well in the results, but the best mention of them previously to the methods is on lines 76-77 in the Introduction when discussion Erickson et al. (2004). Which begs the question, which definition of growth categories are you using early in this paper?

E. Line 604 (and others), you mention halfway points in life. What are you considering the maximum lifespan to be?

F. Line 1374 seems unnecessarily enthusiastic with the exclamation point.

G. Figures 24, 25, 26 would be best represented as a table; figure 29 is largely redundant with Figure 1 and could easily be combined; figures 3 and 4 should also be combined; Figure 7 is difficult to read as presented.

The references are thorough and contain all of the most important literature for the discipline. Errors, largely in the form of missing citations, were found throughout the reference section and are outlined below.

A. Lines 137-139, there is no consistency in the order of citations.
B. Line 187, Swofford, 2003 is cited as Swofford 2002 in the references section.
C. Lines 305, 351, Wurneburg et al., 2019 is cited as Werneberg in the references section.
D. Line 820, Leslie et al., 2018 is not in references section.
E. Line 1121, Molnar, 2013 is not in the references section.
F. Line 1176, Moss, 1957 is not in the references section.
G. Line 1196, Hurlburt et al. 2013 is not in the references section.
H. Lines 1302, 1305, Osborn, 1905 is not in the references section.
I. Lines 1316, 1320, Osborn, 1906 is not in the reference section.
J. Line 1348, Wilkinson and Rhodes, 1997 is not in the reference section.
K. Line 1357, Lincoln et al., 1982 is not in the reference section.
L. Line 1447, Pira et al., 2010 is not in the reference section
M. Line 1448, Dodson, 1975 and Hall and Portier, 1977 are not in the reference section.
.

Experimental design

The scope of the paper fits well with other articles published in PeerJ, including a paper I authored in 2014 using this very method. The research question, goals, and assumptions are all well-defined early in the paper. Overall, I had only a few issues with experimental design.

A. I'd like to see more about how you defined the nascent from the adult state in the methods. If the smallest specimens are used to create the outgroup (artificial embryo), it would be proper to describe how these are unequivocally immature by all vertebrate growth proxies (small size, few LAGS, unfused sutures, etc). This will also make your decision to remove juvenile and subadult specimens from some of the Spearmen Rank analysis (such as those comparing size to growth stage) more understandable because the least mature juveniles are being defined as such on the onset of the analysis.

B. Line 154, you discuss reducing your data set due to uninformative characters. I believe greater explanation is needed to warrant pruning data. If there is only a single character state found in all specimens, removal is justified to reduce tree length. However, the other three criteria need more explanation. “Codings lacked pattern” seemingly implies that you were not completely objective when running the analysis and that you had a predicted topology. The same goes for removing potential autapomorphies (autontomorphies?) from subadult specimens. Last, removing characters only scored in 3 or fewer specimens is intuitive but also seems to lack justification.

C. Explain how you decided which specimens to keep from redundantly coded individuals.

D. Line 168, The comment on Holliday et al. (2019) is most relevant in the context of how it affects your coding. I suggest you add a line on which characters these observations change.

E. An a posteriori analysis to determine the most mature specimen is appropriate and your method is with merit. However, on the final tree, FMNH PR2081 has nine autontomorphies, while CM 9380 has nine reversals and two autontomorphies. This begs the question, why can’t the individual with the most adult character states be assumed the most mature?

Validity of the findings

A. Line 237-239, states that the ontogram would divide into branches if sexual dimorphism is present. This is what would theoretically occur if the dimorphic characters outnumbered the synontomorphies. This phenomenon, however, has not been demonstrated in a published analysis and should thus be treated with caution and skepticism. Instead, I suggest investigating reoccurring character reversals found in the adult specimens.

B. In the section on Synontomorphy Trends, the location of peaks will change based on skeletal completeness and the type of optimization used in the analysis. At least two of the peaks (at GS 5 and GS 18) occur before polytomies with relatively incomplete specimens. I suggest you map skeletal completeness based on the percent of scored characters for each specimen in your growth categories and compare those as a discussion of a potential source of bias in the ontogenetic record.

C. I strongly don't believe that the Spearman Rank Correlation can be used to identify geographic or stratigraphic bias in this particular sample. Geographically you may be able to code location for every specimen (line 837) but you’ve lumped them together by state, an artificial invention of modern politics and geography. Instead, I would use latitude to make a more precise delineation of spatial differences. As for the stratigraphic position, the data set is just too small to have any confidence in this result. Even if two species were present in the data, the growth series would likely overlap too significantly to differentiate. As with sexual dimorphism, this is another testable hypothesis that needs to be explored further in future studies.

D. Line 874, are Carpenter (1990)’s characters of ischial orientation included in the analysis? I was not able to find them in the supplementary file based upon the description in the text, but believe they should be included to best capture any potential dimorphism in the sample.

Additional comments

Overall, I found this to be an enjoyable and important paper to review. The data set itself is comprehensive and will undoubtedly prove valuable for future tyrannosaur researchers. Given time constraints and my ignorance regarding Tyrannosaurus anatomy, I have taken your description of specimens at face value and have instead focused on your methods, assumptions, and interpretations. In all cases, this paper is worthy of future publication with minor revisions and clarifications.

Reviewer 2 ·

Basic reporting

Firstly, this is a monumental piece of work requiring a tremendous amount of observations, which is sure to generate substantial future lines of research.

The writing and argumentation is clear and the data employed are well-documented in the body and supplementary materials.

A particular aspect that is appealing is the set of predictions laid out for future testing: I look forward to results of these tests. Additionally, the author is very clear in addressing one by one the various alternative explanations (geographic, stratigraphic, sexually dimorphic, etc.) for variations among the adult specimens.

The figures are very informative; not only the documentation of the essential patterns discovered herein, but also the mapping of the results of other studies onto the newly completed growth patterns. This helps to put other research into a new context.

Experimental design

One issue that should be addressed at least in passing is the possibility of multiple ontogenetic trajectories within one taxon in the same population. This pattern has been documented in dinosauromorphs in papers by Griffin & Nesbitt. To be fair, they conclude that this pattern may be primitive to dinosauromorphs but transformed in derived dinosaurs which seem to have a more rigidly fixed ontogenetic pathway, but they did not have the immense database presented here to confirm that. Thus this might be a good opportunity to at least briefly address this concept, given the substantial data set now available:

Griffin, C.T. & S.J. Nesbitt. 2016. The femoral ontogeny and long bone histology of the Middle Triassic (?late Anisian) dinosauriform Asilisaurus kongwe and implications for the growth of early dinosaurs. Journal of Vertebrate Paleontology 36: e1111224. Doi: 10.1080/02724634.2016.1111224

Griffin, C.T. & S.J. Nesbitt. 2016. Anomalously high variation in postnatal development is ancestral for dinosaurs but lost in birds. PNAS 113: 14757-14762. Doi: 10.1073/pnas.1613813113

Validity of the findings

Lines 149-150 The author is clear on his coding philosophy, allowing for repeatability. However, since parsimony-based phylogenetic analysis software can often incorporate polymorphic observations for individual OTU, it might be informative in this or a future study to run the analysis with both observed states present. We cannot know a priori if it would affect the results, but it might (for instance) reveal particular growth phases of transition found in multiple individuals, and thus more completely characterize the ontogeny of Tyrannosaurus rex. (I would not regard this as a requirement for the present study, however.)

Lines 168-174 It is worth noting that Holliday et al. (2019) did not solely rely on bone surface texture in arguing that the frontoparietal fossa was occupied by vascular tissue. They also discuss the unlikelihood of it being occupied by the adductor musculature in dinosaurs, as it would require a right-angle muscle reorientation with no apparent sesamoid. They might be inaccurate in this presumption, it is true. The present study does not sufficiently explore all the lines of evidence that Holliday et al. used to come to their conclusion. Therefore, it might be more appropriate to say here that based on the present author’s observation of the bone texture of this region, he will accept the traditional view of this fossa rather than the new interpretation. (This was a longwinded way of saying that the present author didn’t actually reject Holliday et al.’s interpretation as he didn’t examine all the relevant lines of evidence but that he is preceding for the purpose of this study with the traditional assessment.)

Line 217 Just for clarification, the author should indicate that the 81 hours was the completion of the analytical run (rather than the author quitting the process prematurely).

Line 336ff For the discussion of growth categories, the author states mass ranges which are given to the kilogram. This is following the results of previous works cited within. But this propagates a problematic issue of these studies: a false impression of precision in mass estimations. These studies often give results far beyond the number of significant digits of the particular measurements actually observable on the fossils.

Additional comments

Again, this is a spectacular piece of research, and I look forward to seeing its publication.

·

Basic reporting

Overall, this work is a major, self-contained synthesis of nearly all available data on T. rex biology, in the context of the most comprehensive cladistic ontogenetic analysis of a fossil taxon to date. The integrative approach provides a clear, replicable framework for distinguishing sources of anatomical variation among tyrannosaurids. The study design and interpretations of ontogenetic data are broadly sound. My most significant comments concern the specifics of how the cladistic methodology was employed. I look forward to the publication of this manuscript following a response to the comments and revisions I outline below.

The manuscript is generally very clearly written. See “Specific Comments” and “Typos” below for instances in which further clarification is needed and spelling mistakes should be corrected. The structure conforms to both PeerJ standards and discipline norms.

The introduction provides a good background on both recent research on tyrannosaurids and the cladistic ontogeny method. Given the high degree of variability present among the mature individuals present in the sample, the author should discuss Ontogenetic Sequence Analysis (Colbert and Rowe 2008), why it was not applied to this study, and what it could reveal about variation in T. rex ontogenetic trajectories if applied. Other instances where further citations/discussion of background literature are needed are listed under “Specific Comments” below.

The figures are generally effective, though I provide some suggestions for improving their clarity under “Figures” below.

Experimental design

The integrative approach to analyzing the “three axes of ontogeny” together is an important step forward in analyses of this kind. It fills a knowledge gap in identifying whether or not size and age correlate with other morphological measures of maturity in T. rex. I have no concerns about the originality or ethics of the work.

Replication of the study could be enhanced by providing a supplementary list of commands/steps performed in the cladistic analysis.

The cladistic results need further interrogation before a tree topology can be settled on for purposes of discussion. Support for clades should be identified by bootstrap, jackknife and Bremer decay analyses. There should be a discussion of which specimens have the most labile placement, and consequently lead to the reduced resolution of the strict consensus tree. The influence of the duplicated characters already identified by the author on the results should be examined. If the author is intent on presenting a majority rule consensus tree, problems with this consensus method (Sumrall et al. 2001) and their influence on the results should be discussed. Confidence in interpretations of relative maturity obtained from a majority rule consensus tree should be tempered appropriately. In any case, the author should emphasize that only those growth stages (“clades”) retained on the strict consensus tree are clearly supported. All other groupings are more ambiguous. See “Specific Changes” below for more detailed comments on these points.

I do not anticipate that these additional experiments will greatly alter the overall conclusions of the study, but they should be performed to fully explore the data and indicate degree of support for each “clade”. I am recommending major revisions largely because the topology of the cladogram is the foundation upon which the results and discussion sections rest, and further work needs to be done to test the robustness of support for any topology presented. If some clades are found to be weakly supported, this may have downstream effects that would necessitate further revisions to the rest of the manuscript. “Major revisions” should not be taken to indicate anything fundamentally flawed about the overall intellectual approach or the interpretation of experimental results. The results and discussion sections are generally excellent and, assuming no major changes in tree topology, should be largely acceptable.

Validity of the findings

The extensive, detailed descriptions provide a comprehensive framework for interpreting anatomical variation in tyrannosaurids.

The discussion of individual variation is thorough, and the thoughtful interpretations provide much-needed clarity to discussions of intraspecific variation in dinosaurs.

Likewise, the discussion of T. rex biomechanics, geographic localities, and stratigraphic positions in ontogenetic context are welcome and greatly add to the already considerable value of this study as an integrative work on tyrannosaurid biology.

The conclusions are appropriately limited to the results presented.

Additional comments

I am commenting as a researcher experienced with methods used to study the ontogeny of fossil vertebrates and not as someone with specialized knowledge of tyrannosaurid anatomy, beyond that reasonably expected from a dinosaur worker who focuses on the anatomy of basal theropods and basal ornithischians.

This paper needs a justification of why Ontogenetic Sequence Analysis (OSA) (Colbert and Rowe 2008) was not used, particularly given the high degree of variability in T. rex ontogenetic characters (analogous to homoplasy in a phylogenetic analysis) that produces a poorly-resolved strict consensus tree. OSA can be used to examine the variation in ontogenetic sequences that a population can undergo (Colbert and Rowe 2008, Griffin and Nesbitt 2016). The author should engage with the arguments of Griffin and Nesbitt (2016—see supplementary information quoted at length below) about the similarities/differences between cladistic ontogeny and OSA, and reasons to use one method or the other. As Griffin and Nesbitt (2016, supplementary information) state:

"OSA is similar to cladistic ontogeny (8, 9) in that it is a parsimony-based method of determining the sequence of skeletal character change through ontogeny. However, whereas OSA uses information from all trees returned in a parsimony analysis of the ontogenetic character data (see Methods), and thus reconstructs all developmental pathways consistent with the data, cladistic ontogeny utilizes a consensus tree, minimizing the ability to observe any variation in developmental sequences in a population. Cladistic ontogeny is therefore a faster method of analyzing morphological changes during growth, and unambiguous ontogenetic character transformations shared by all sequences within a population are easily discernable using that analysis. However, any variation in the population will appear mainly as a lack of resolution in ontogenetic characters, and not an interpretable signal in its own right, limiting ability to understand and quantify variation. Furthermore, it becomes increasingly more difficult to derive a meaningful signal from cladistic ontogeny with increasing amount of sequence polymorphism, so that a dataset with the level of variation of Coelophysis bauri results in a consensus tree consisting of a single large polytomy. Thus, each method has strengths and weaknesses, but except in instances where variation is virtually absent or explicitly intended to be ignored, we advocate using OSA to reconstruct morphological changes during growth. One of our major conclusions of this contribution is that using OSA is necessary in early dinosaurs when reconstructing growth."

I think a dialogue in the literature between the practitioners of these complementary approaches would serve the field well.

The author consistently mentions that T. rex bite force increased through ontogeny to produce the high values seen in adults. However, there are no citations to either the paper that established this quantitatively for the juvenile T. rex BMRP 2002.4.1 (Bates and Falkingham 2012), nor to papers establishing high bite force in adult T. rex (Erickson et al. 1996, Gignac and Erickson 2017), nor any comparisons to the literature on ontogenetic changes in crocodylian bite force (Bates and Falkingham 2012; Erickson et al. 2003; Erickson et al. 2014; Gignac and Erickson 2015, 2016; Gignac and O’Brien 2016). Such comparisons between crocodylian bite force ontogeny and that of T. rex should be discussed in the sections on bite force in the current manuscript.

If the author has not done so already, he should check to make sure that the 2015 corrigendum to Erickson et al. (2004) does not affect any of his interpretations. As there were no major corrections made to the 2004 study, I doubt there will be any effect on the current work, but I bring it to the author’s attention just in case.

Specific Comments:
Line 168: This paragraph on the texture surrounding the dorsotemporal fossa seems somewhat out of place here. I suspect it is included here because the interpretation of this anatomical region is important to character definition and construction, but the author should add a statement clarifying this.

Line 176: A more explicit rationale and justification of the method by which the quantitative characters (e.g., size) were divided into discrete states is needed here. It’s also possible in TNT phylogenetic analysis software to analyze the measurements as continuous characters, without first breaking them into discrete states (Goloboff et al. 2006). This approach could be worth trying to see if it alters the results.

Line 180: This sentence could use some clarification. I think the author meant “relative size is used only when a specimen is represented by a single bone or a partial skull or skeleton.” Note the position of the word “only” in this sentence compared to the original.

Line 187: How many random addition sequences and rounds of branch swapping were performed? This should be stated. The author should consider including a list of the specific PAUP* commands/procedures utilized as supplementary information to ensure ease of replication by future researchers.

Line 206: The use of terminology paralleling phylogenetic analysis (e.g., autontomorphies, synontomorphy) is very effective. As a whole, I hope that this paper will further advance attempts to standardize cladistic ontogeny terminology.

Line 229: Increased resolution is not an appropriate reason to favor presenting a majority rule consensus tree over another type. As Sumrall et al. (2001) state:

"A systematist should acknowledge all most parsimonious trees, whether one is preferred over the others or not; because we can never know if a given tree is the ‘‘real’’ phylogeny, each of the most parsimonious trees must be considered as a viable candidate until explicit criteria for selecting among them (or even for selecting among less optimal trees) are presented. Numerical minority of a node in a set of most parsimonious trees is a spurious rationale for rejection."

In the context of this study, the other 50% of trees obtained by the heuristic search, but not summarized by the 50% majority rule consensus tree, are still equally parsimonious to those presented. A tree that lacks a clade present in 99% of the other trees is still an equally parsimonious “solution” (as measured by tree length) to the “problem” of optimizing a given set of characters for a given set of specimens. In this study, there are equally parsimonious trees that, to give two examples, do not place FMNH PR2081 as more mature than UMNH 11000, or show CMNH 7541 as less mature than BMRP 2002.4.1 (as shown in Figure 1, neither of these groupings was recovered in 100% of the most parsimonious trees). Locally labile terminal specimens in an analysis (i.e., those that bounce between two relatively close placements) can introduce bias into majority rule consensus results (Sumrall et al. 2001). An Adams consensus tree may be useful to identify the specimens with the most labile placement. If the author wishes to use a majority rule consensus tree, the concerns of Sumrall et al. (2001) should be discussed, and confidence in the results qualified appropriately.

As Nixon and Carpenter (1996) state: “If the goal of using consensus trees is to summarize the agreement in grouping among a set of cladograms, then only the “strict” consensus tree fulfills that goal. The other methods may all result in groups that are not supported by the data at hand, or supported only ambiguously.” Alternatively, the author could present a reduced strict consensus tree following “safe” pruning of unstable specimens (Wilkinson 2003). It is necessary to investigate how the choice of consensus method influences the tree topology, because all subsequent parts of the manuscript currently rely uncritically on the topology of the majority rule consensus tree.

Likewise, the frequency at which a clade appears among a set of most parsimonious trees is not really a true measure of support for that clade. The author should perform bootstrap, jackknife, and Bremer decay analyses on this dataset and report the resulting values at the nodes as more informative measures of support for those clades.

Line 456: Is there any precedent among studies of extant taxa for using a functional/ecological definition of somatic maturity? Perhaps the author is getting at this with comparisons to metamorphic extant taxa, but this should be more explicitly stated, especially in terms of how it differs from other definitions of somatic maturity used for extant taxa.

Line 788: The author should restate what he’s using as the “growth rank” and what is meant by “maturity” in this context. Is “maturity” the placement of the specimen on the ontogram?

Line 1085: Which recent work is the author referring to that shows that the T. rex skull was akinetic? I think a citation is needed here for clarity.

Line 1207: It would be helpful to the reader to briefly restate the distinction between a phylogenetic synontomorphy and a nonphylogenetic change here.

Line 1278: What happens when the oversplit characters are either reduced down to single characters or removed? The author should investigate how much influence these characters are exerting on the topology of the resulting ontogram.

Line 1329: I think some discussion is warranted about whether or not the reversals of ontogenetic character states seen in some adult T. rex may have been actual reversals over the lifetime of an individual, or whether those individuals simply retained juvenile character states into adulthood, never passing through a transition for those characters in the first place. In other words, how do we distinguish a reversal in a character state optimization on an ontogram from a reversal over the course of an individuals’ lifetime (such as growth followed by later resorption of the horns of some marginocephalians—e.g., Horner and Goodwin, 2009). I realize that it may not be possible to test this, but to me, it’s important to think conceptually about whether these T. rex individuals arrived at state 0 in adulthood by either a transformation of 0->1->0 or simply retained state 0 from an earlier ontogenetic stage, having never achieved the “1” state. A sentence indicating this distinction would provide a useful conceptual framework for future authors to think about reversals.

Line 1437: The author should explicitly indicate that Proterosuchus is an outgroup to Archosauria (i.e., outside the T. rex + Crocodylia clade). The subsequent discussion of changes in skull shape among archosaurs should take into account additional ontogenetic shape change data gleaned by Bhullar et al. (2012, and associated supp. info.). This study provides additional data on crocodylians, other non-dinosaur archosaurs, and dinosaurs themselves that could help infer plesiomorphic cranial changes.

Line 1539: Point 4: How might this prediction be tested? It is clear enough what evidence is needed to test the other three predictions, but I think the author should explicitly state what a hypothetical specimen either supporting or disproving prediction 4 would look like.

Line 1559: The author might mention that this is certainly not unique to T. rex among non-avian dinosaurs; indeed, it’s probably a plesiomorphic trait shared with crocodylians. I recommended citing Erickson et al. (2007), Lee and Werning (2008), etc.

Line 1562: Point 14. This point is more of a prediction/hypothesis than an observation from the current study. This should be explicitly stated.

Figures
Figure 1. Growth stages should be color-coded on these trees. See comments above about problems with presenting a majority rule consensus tree. At the least, this tree should have Bootstrap/jackknife and Bremer Decay values listed at the nodes.

Figure 3. Font size could be increased on these graphs to enhance readability.

Figures 7 – 9. Given the number of different colors used, the color palette should be checked to see if it can accommodate color-blind people, if the author hasn’t done so already (https://www.color-blindness.com/coblis-color-blindness-simulator/). Additionally, each anatomical domain should be symbolized by differently shaped points along each line (e.g axial skeleton w/ triangles, scapula with squares, etc.)

Figures 11 and 12. Further clarification should be added to the figure captions about what exactly is meant by dentary tooth rank and maxillary tooth rank.

Figure 18. Color coding of specimen numbers should be used in combination with the existing superscript scheme to help distinguish gracile vs. robust specimens. For example, all specimens marked “G” could also be colored blue, and all those marked “R” could also be colored red.

Figure 19. If a color scheme for gracile vs. robust specimens is adopted for Figure 18, incorporate the same color scheme into this figure.

Figure 29. The taxon sample in this figure could be expanded by incorporating data from additional studies of ontogenetic shape change in archosaur skulls (e.g. Bhullar et al. 2012).

Tables:
Given that PeerJ does not have page length restrictions, I advocate keeping these tables as part of the paper and not relegating them to supplementary information. However, I think they would be best placed as appendices at the end of the paper so as not to needlessly interrupt the flow of the main part of the manuscript.

Typos
I did not exhaustively check the manuscript for spelling, but noted typos as I found them.

Line 783: the word “count” should be added after “tooth” in the sentence beginning “Among adults, the hypothesis of dentary tooth as a proxy…”

Line 1084: “compares” should be changed to “compared”.

Lines 1192 and 1193: Both instances of “laterophenoidoprootic” should be changed to “laterosphenoidoprootic” (note the “s”).

Line 1215: I think the following sentence is missing a word: “However, only six of ___ unambiguously optimized synapomorphies are congruent…”

Line 1672: “lestest” in the reference should be “latest”

Line 1728: “Crcocodylia” in the reference should be “Crocodylia”

References cited in this review
Bates, K. T., & Falkingham, P. L. (2012). Estimating maximum bite performance in Tyrannosaurus rex using multi-body dynamics. Biology Letters, 8(4), 660-664.
Bhullar, B. A. S., Marugán-Lobón, J., Racimo, F., Bever, G. S., Rowe, T. B., Norell, M. A., & Abzhanov, A. (2012). Birds have paedomorphic dinosaur skulls. Nature, 487(7406), 223.
Colbert, M. W., & Rowe, T. (2008). Ontogenetic sequence analysis: using parsimony to characterize developmental sequences and sequence polymorphism. Journal of Experimental Zoology Part B: Molecular and Developmental Evolution, 310(5), 398-416.
Erickson, G. M., Van Kirk, S. D., Su, J., Levenston, M. E., Caler, W. E., & Carter, D. R. (1996). Bite-force estimation for Tyrannosaurus rex from tooth-marked bones. Nature, 382(6593), 706.
Erickson, G. M., Lappin, A. K., & Vliet, K. A. (2003). The ontogeny of bite-force performance in American alligator (Alligator mississippiensis). Journal of Zoology, 260(3), 317-327.
Erickson, G. M., Makovicky, P. J., Currie, P. J., Norell, M. A., Yerby, S. A., & Brochu, C. A. (2004). Gigantism and comparative life-history parameters of tyrannosaurid dinosaurs. Nature, 430(7001), 772.
Erickson, G. M., Curry Rogers, K., Varricchio, D. J., Norell, M. A., & Xu, X. (2007). Growth patterns in brooding dinosaurs reveals the timing of sexual maturity in non-avian dinosaurs and genesis of the avian condition. Biology Letters, 3(5), 558-561.
Erickson, G. M., Gignac, P. M., Lappin, A. K., Vliet, K. A., Brueggen, J. D., & Webb, G. J. W. (2014). A comparative analysis of ontogenetic bite‐force scaling among Crocodylia. Journal of Zoology, 292(1), 48-55.
Gignac, P. M., & Erickson, G. M. (2015). Ontogenetic changes in dental form and tooth pressures facilitate developmental niche shifts in American alligators. Journal of Zoology, 295(2), 132-142.
Gignac, P. M., & Erickson, G. M. (2016). Ontogenetic bite‐force modeling of Alligator mississippiensis: implications for dietary transitions in a large‐bodied vertebrate and the evolution of crocodylian feeding. Journal of Zoology, 299(4), 229-238.
Gignac, P. M., & Erickson, G. M. (2017). The biomechanics behind extreme osteophagy in Tyrannosaurus rex. Scientific reports, 7:2012, 1-10.
Gignac, P., & O’Brien, H. (2016). Suchian feeding success at the interface of ontogeny and macroevolution. Integrative and comparative biology, 56(3), 449-458.
Griffin, C. T., & Nesbitt, S. J. (2016). Anomalously high variation in postnatal development is ancestral for dinosaurs but lost in birds. Proceedings of the National Academy of Sciences, 113(51), 14757-14762.
Goloboff, P. A., Mattoni, C. I., & Quinteros, A. S. (2006). Continuous characters analyzed as such. Cladistics, 22(6), 589-601.
Horner, J. R., & Goodwin, M. B. (2009). Extreme cranial ontogeny in the Upper Cretaceous dinosaur Pachycephalosaurus. PLoS One, 4(10), e7626.
Lee, A. H., & Werning, S. (2008). Sexual maturity in growing dinosaurs does not fit reptilian growth models. Proceedings of the National Academy of Sciences, 105(2), 582-587.
Nixon, K. C., & Carpenter, J. M. (1996). On consensus, collapsibility, and clade concordance. Cladistics, 12(4), 305-321.
Sumrall, C. D., Brochu, C. A., & Merck, J. W. (2001). Global lability, regional resolution, and majority-rule consensus bias. Paleobiology, 27(2), 254-261.
Wilkinson, M. (2003). Missing entries and multiple trees: instability, relationships, and support in parsimony analysis. Journal of Vertebrate Paleontology, 23(2), 311-323.

---

## Round 0.2 · accepted · Accept

Thank you for your very thorough attention to the comments from the reviewers. Upon re-review, they were unanimous (and I agree) that all requested changes have been addressed, and no further revision is warranted.

·

Basic reporting

The errors and inconsistencies in the first review have been corrected.

Experimental design

The experimental design, especially those of the a posteriori tests, has been substantially improved. My original criticisms have all been satisfied with modifications to the tests or thorough explanation.

Validity of the findings

I see no argument with the validity of the findings. The author addressed my concerns over testing for sexual dimorphism and geographic variation sufficiently.

Additional comments

You gave the manuscript substantial revision beyond what was requested. I appreciate your exploration of different ways to identify the most mature specimen and sexual dimorphism in the ontogram. Undoubtedly this is an important contribution to our knowledge of tyrannosaurs and I recommend it for publication in its revised form.

Reviewer 2 ·

Basic reporting

This revision of an already exceptional manuscript has only improved it. The clarity of the arguments have been strengthened, and the new version actually contains even more important descriptive and analytical information than the first.

The concerns of myself and the other previous reviewers have been appropriately addressed, either by incorporation into the present manuscript or by clear explanation as to why the suggestions cannot be implemented at the present time.

Experimental design

The modifications to the previous incarnation of the manuscript (for instance, the tree generation process) are clearly described and repeatable. The assumptions and recognition of the limitations of the procedures are made clear.

Validity of the findings

The author has presented strong evidence for his conclusions.

Additional comments

These points are trivial enough that they do not reach the level of me indicating "minor revisions"; nevertheless, they might mildly improve the manuscript.

I did find a minor typographical error on line 1133: "immatuirty"

The discussion of the dentary groove (starting on line 1150) should including the response paper of which the manuscript author was a contributor: doi: 10.1016/j.cretres.2016.02.007
* * *
I apologize for being slower to work my way through this version of the manuscript than other authors, but I wanted to be secure that the previous reviewers' comments had all been addressed and that the new material was fairly evaluated.

·

Basic reporting

All of my comments are in "General Comments" below.

Experimental design

All of my comments are in "General Comments" below.

Validity of the findings

All of my comments are in "General Comments" below.

Additional comments

Dr. Carr has done a very thorough job of addressing my concerns. His responses to my comments and those of the other reviewers universally improved my confidence in the results, deepened the discussion, and clarified the figures. In particular, I applaud the reanalysis of the data matrix in TNT and the use of a single most parsimonious ontogram that avoids the problems associated with using consensus ontograms. I am also very satisfied with his discussion of Ontogenetic Sequence Analysis (OSA). I have no further important comments or changes to recommend. This manuscript stands as a landmark work for ontogenetic studies in vertebrate paleontology, and I urge its acceptance for publication.